# Mapping global lake dynamics reveals the emerging roles of small lakes

Xuehui Pi [1,2,3,13], Qiuqi Luo [1,13], Lian Feng [1] ✉, Yang Xu [1,4], Jing Tang[5,6], Xiuyu Liang[1], Enze Ma[1], Ran Cheng[7], Rasmus Fensholt [4], Martin Brandt [4], Xiaobin Cai [8], Luke Gibson [1], Junguo Liu [1,9], Chunmiao Zheng [1,10], Weifeng Li[2,3,11] & Brett A. Bryan [12]

Lakes are important natural resources and carbon gas emitters and are undergoing rapid changes worldwide in response to climate change and human activities. A detailed global characterization of lakes and their long-term dynamics does not exist, which is however crucial for evaluating the associated impacts on water availability and carbon emissions. Here, we map 3.4 million lakes on a global scale, including their explicit maximum extents and probability-weighted area changes over the past four decades. From the beginning period (1984–1999) to the end (2010–2019), the lake area increased across all six continents analyzed, with a net change of +46,278 km$^2$, and 56% of the expansion was attributed to reservoirs. Interestingly, although small lakes (<1 km$^2$) accounted for just 15% of the global lake area, they dominated the variability in total lake size in half of the global inland lake regions. The identified lake area increase over time led to higher lacustrine carbon emissions, mostly attributed to small lakes. Our findings illustrate the emerging roles of small lakes in regulating not only local inland water variability, but also the global trends of surface water extent and carbon emissions.

Lakes play a major role in global hydrological and biogeochemical cycles[1–4] and underpin vital ecosystem functions and services[5,6]. However, rapid lake changes have been identified worldwide in response to changing climate and escalating human activities[7–10], threatening the ecosystem services provided by these lacustrine habitats. For example, lake desiccation has been observed in some populated regions due to unregulated water withdrawal, triggering water shortages, international conflicts, and other societal consequences[7,11,12]. Conversely, widespread glacier lake expansions have been detected due to climate warming-induced snowmelt and glacial melting[13,14].

A spatially explicit understanding of lake size changes is essential for evaluating the associated ecological, environmental, and societal impacts. In theory, determining lake dynamics using satellite images is straightforward, as remotely sensed imagery is available at high temporal frequencies with global coverage. However, terrestrial surface waters share similar optical features[15], and these similarities cause challenges when differentiating between lakes and rivers using satellite

[1]School of Environmental Science and Engineering, Southern University of Science and Technology, Shenzhen, China. [2]Department of Urban Planning and Design, The University of Hong Kong, Hong Kong SAR, China. [3]Urban Systems Institute, The University of Hong Kong, Hong Kong SAR, China. [4]Department of Geosciences and Natural Resource Management, University of Copenhagen, Copenhagen, Denmark. [5]Department of Biology, University of Copenhagen, Copenhagen, Denmark. [6]Department of Physical Geography and Ecosystem Science, Lund University, Lund, Sweden. [7]Guangdong Provincial Key Laboratory of Brain-inspired Intelligent Computation, Department of Computer Science and Engineering, Southern University of Science and Technology, Shenzhen, China. [8]Institute of Geodesy and Geophysics, Chinese Academy of Sciences, Wuhan, China. [9]School of Water Conservancy, North China University of Water Resources and Electric Power, 450046 Zhengzhou, China. [10]EIT Institute for Advanced Study, Ningbo, China. [11]Institute for Climate and Carbon Neutrality, The University of Hong Kong, Hong Kong SAR, China. [12]Centre for Integrative Ecology, School of Life and Environmental Sciences, Deakin University, Burwood, Victoria, Australia. [13]These authors contributed equally: Xuehui Pi, Qiuqi Luo ✉e-mail: fengl@sustech.edu.cn

signals. A typical example is the recent map of changes in global surface water[12], where the respective contributions of lakes and rivers to the total area or the detected changes remain uncertain. As lentic lakes and lotic river systems represent distinct hydrological and biochemical processes[16,17], further efforts are warranted to quantify their distinct roles in these hydrological changes.

Estimates of the global extent of lakes are available from several existing datasets[17–20], but these were largely generated using snapshots of historical imagery and did not consider seasonal and interannual fluctuations[12,18]. Previous studies on lake dynamics primarily focused on medium- to large-sized lakes with well-defined boundaries[7,14,21]. Small lakes (defined herein as lakes with an area <1 km²) are more variable than large lakes due to their high sensitivity to natural wet/dry transitions and human management activities[19,22]. Furthermore, small lakes have a disproportionally large contribution to global lacustrine systems in terms of their primary productivity[23], biodiversity[24,25], and carbon cycle[26]. However, these available global assessments of ecosystem parameters are usually based on uncertain estimates of the sizes and distribution of small lakes[3,26,27], and few studies have examined how freshwater biogeochemical cycles are influenced by lake variation because changes in lake sizes over time have never been characterized globally. Here, we fill this existing knowledge gap and map more than 3.4 million lakes and reservoirs with surface area >0.03 km² (hereafter simply lakes unless otherwise specified) at the global scale. We used deep learning to identify lakes smaller than the minimum mapping unit for all global lake datasets that are publicly available (0.1 km²). We then examined changes in the global lakes over four decades and discussed the associated implications for lacustrine carbon emissions.

## Results

### Determination of global lake extent

The dataset constructed in this study was named GLAKES and is based on the Global Surface Water Occurrence (GSWO) dataset and a deep-learning classification algorithm (see Methods and Supplementary Fig. 1). The GSWO dataset provides the probability of water presence, which was established using 30 m resolution Landsat satellite observations between 1984 and 2019. Deep learning allows for the disentanglement of lakes from rivers in the GSWO images, and the integration of high-resolution remote sensing images and deep learning makes it possible to detect lakes as small as 0.03 km² (corresponding to ~33 Landsat image pixels), which greatly improves the minimum mapping unit and mitigates the issues of mis-accounted small lakes in previous lake datasets that are publicly accessible. Validation showed high accuracy levels across different size groups in our dataset (Supplementary Figs. 2, 3, Supplementary Table 1). Our GLAKES dataset indicates a total lake area of $3.2 \times 10^6$ km² in all-time maximum (2.2% of the global land area), with 49.8% and 23.6% of the total number of lakes and total lake area located north of 60°N (Fig. 1), similar proportions to those reported in previous datasets[17,18]. We partitioned lakes into three size groups, small (<1 km²), medium (1–100 km²), and large (>100 km²); these size groups accounted for 94.39%, 5.56%, and 0.05% of the total number of lakes, respectively, and 15%, 26%, and 59% of the total lake area, respectively. Including lakes <0.1 km² in size (1.91 million lakes) resulted in a 30.2% larger bounded lake area in the small-lake group than that mapped by the recently published and widely used HydroLAKES dataset[17] (Supplementary Figs. 4, 5).

### Four decades of lake changes

We examined global lake dynamics across three periods (1980–1990s: 1984–1999), 2000s: 2000–2009), and 2010s: 2010–2019) (Fig. 2) by comparing the water probability-weighted area within lake boundaries as defined by our GLAKES dataset (see Methods). We gridded global lakes into 1° × 1° cells and excluded pixels with insufficient satellite

image coverage (particularly those located in eastern Russia and central Africa) in early periods in the comparison (Supplementary Fig. 6).

From the 1980–1990s to the 2000s, the global lake area showed a net increase of 39,784 km². Lakes and reservoirs fed by glaciers or permafrost, representing 30% of the worldwide lake area, accounted for 48% (19,104 km²) of the total increase (Fig. 2); associated expansion hotspots were found in previously well-documented regions, including Greenland, the Tibetan Plateau, and the Rocky Mountains[14]. The remaining 52% increase (20,681 km²) was attributed mostly (75%) to the expansion of reservoirs outside of glacial or permafrost regions (Fig. 2). In contrast, declining lake sizes were observed in the western USA, central Asia, northern China, and southern Australia. These can be associated with local drought events, anthropogenic water withdrawals, and/or other reasons;[7,11,28,29] most of these locations aligned closely with highly water-stressed regions reported in the previous studies[30]. Overall, 2.5-fold more grid cells contained increased lake sizes than the number of cells characterized by lake shrinkage.

From the 2000s to the 2010s, although the total lake area increased, the net area gain represented only 16.3% of that identified from the 1980–1990s to the 2000s. The substantial expansion of reservoirs (8802 km²) dominated the total areal increase from the 2000s to the 2010s, which was more than twice the amount of natural lake increases (3362 km²) in glacial or permafrost regions. Over half (53%) of the global grid cells exhibited inconsistent changes between the 1980–1990s to the 2000s and the 2000s to the 2010s. For example, the lake area has increased in southeastern Australia in the most recent decade, indicating the potential regional recovery of these lakes following the Millennium Drought in the 2000s[28,31]. Similar change patterns were also found in northwestern India, suggesting that this drought-prone region was likely to have suffered less from water stress in recent years[32]. In contrast, large lakes showed a net area decrease of 14,553 km² from the 2000s to the 2010s, 50% of which was attributed to the human-induced shrinkage of the Aral Sea in Central Asia[7].

Continuous lake expansions and shrinkages throughout the entire study period were found in 38% and 9% of the global inland lake regions (i.e., the 1° × 1° grid cells covered with GLAKES), respectively, leading to an overall net area increase of 46,278 km² (i.e., ~1.8 times the size of Lake Erie or approximately the size of Denmark) over the past four decades, and 56% of the increase was attributed to reservoirs.

### The outsized role of small lakes in global lake size variability

Small lakes showed higher long-term temporal variability than large and medium-sized lakes. The median values of the relative areal changes in small lakes were +2.9% from the 1980–1990s to the 2000s and +0.6% from the 2000s to the 2010s; these changes were significantly greater than those derived for medium and large lakes (matched-pair $t$-test, $P < 0.05$) (Fig. 3a). Such difference in temporal variability between small lakes and median/large lakes was also evident even when evaluating under equal relative pixel sizes, as revealed through a case experiment in Tibetan Plateau, which showed that lakes within the size range of 0.5–1.5 km² at the resolution of 30 m were found to exhibit a far larger range of the relative areal changes compared to lakes between 50–150 km² at the 300 m resolution (Supplementary Fig. 7). The sizes of 8.6% and 6.8% of small lakes more than doubled from the 1980–1990s to the 2000s and from the 2000s to the 2010s, respectively. As a result, small lakes supplied a disproportionately large contribution to global lake expansion, representing 46.2% of the net areal increase from the 1980–1990s to the 2010s (Fig. 3b). In contrast, extreme lake expansions occurred to a much lesser extent in the larger lake groups and were observed mainly in human-impounded reservoirs (Supplementary Fig. 8).

The outsized role of small lakes can be further revealed by analyzing their contributions to lake size variability. We define variability as the proportion of absolute areal changes in small lakes within a given 1° × 1° grid cell (see Methods). The changes in small lakes showed

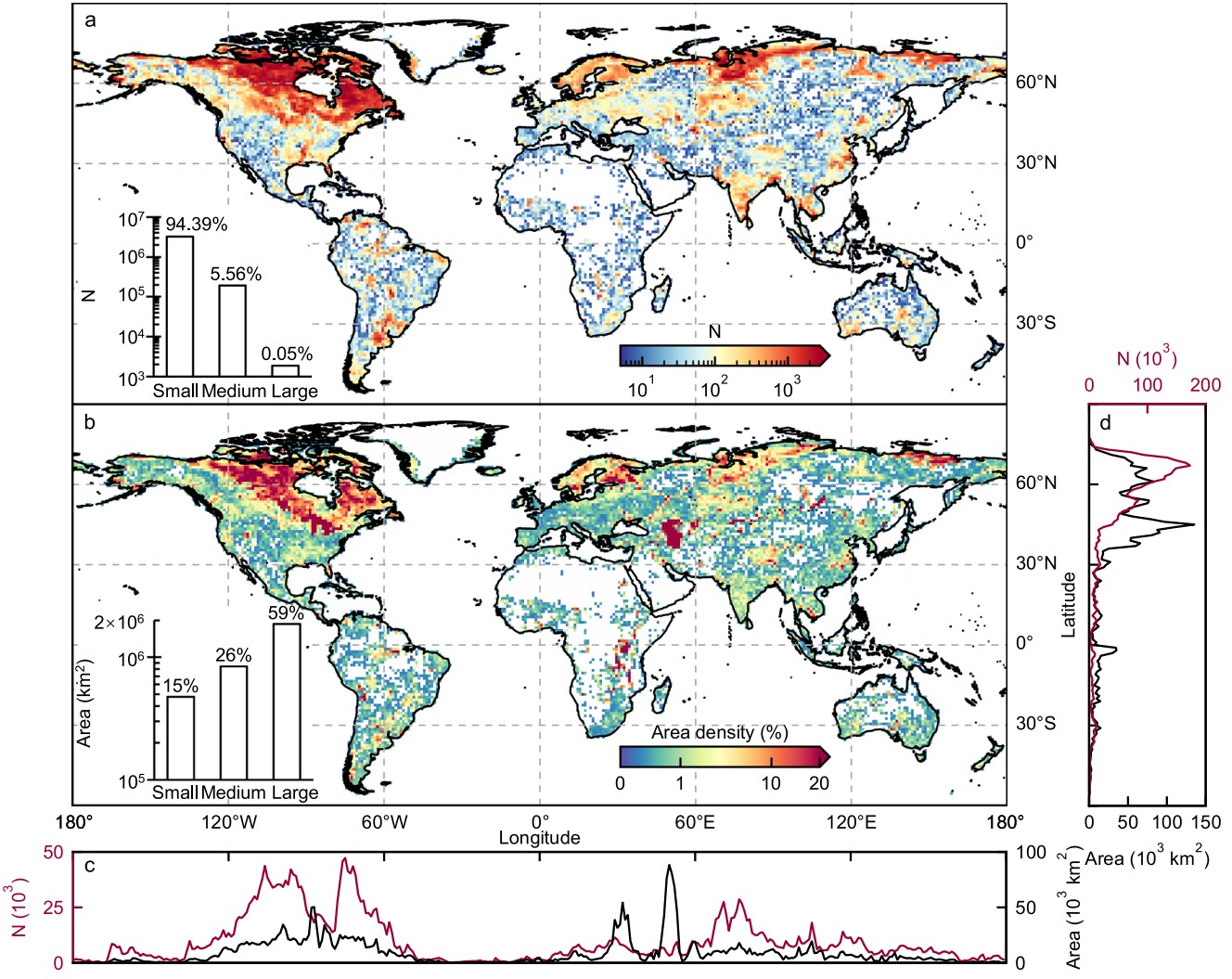

**Fig. 1 | Spatial distribution of global lakes.** Lakes with maximum surface area >0.03 km² were mapped, showing **a** lake count (total number of lakes) and **b** lake area density (total lake area/grid area) per 1° × 1° grid cell. The longitudinal and latitudinal lake profiles summarizing (by 1°) the lake count and lake area are shown on **c** and **d**. Statistics for small (<1 km²), medium (1–100 km²), and large (>100 km²) lakes are presented within each panel of **a** and **b**.

dominant contributions (>50%) in approximately half of the examined inland regions (49.9% of the grid cells from the 1980–1990s to the 2000s, and 50.1% from 2000s to 2010s) (Fig. 3c), and regions, where lake variability is dominated by small lakes, were spread across the entire globe in both low-populated regions and areas with high chances of human disturbance (Supplementary Fig. 9). Furthermore, decadal lake variations generally increased with regional population density, and the variation range was much higher for small lakes than for medium and large lakes, indicating the potential role of human activities in shaping small lakes.

### Updated estimates of lacustrine carbon emissions

Although lakes cover just a small fraction of the Earth's surface, they are important emitters of carbon gases[26,33,34]. A common methodology used to estimate global emissions involves upscaling local measurements to the global scale by multiplying the mean lake-surface fluxes by the global area. Both $CO_2$ and $CH_4$ fluxes are reported to be influenced by lake size, and the data describing global lake size thus become crucial when upscaling carbon emissions[3,26]. With the detailed mapping of global lakes from GLAKES, we can update the total estimations of $CO_2$ and $CH_4$ emissions from global lakes.

We estimated the $CO_2$ and $CH_4$ emissions from global lakes using our GLAKES dataset following the method proposed in a previous study[26] (see Methods). We reached a global total estimate of 226 Tg C yr⁻¹ for $CO_2$ (Fig. 4a), which is smaller than the previous estimate (571 Tg C yr⁻¹)[26]. The difference could be partly due to the use of over-estimated lake surface area by ref. 26, particularly for small lakes; small lakes are featured with high emission rates, and the total surface area from ref. 20 used by ref. 26 is 1.8 times the amount of GLAKES for all lakes, and 2.9 times greater for small lakes (Supplementary Fig. 4a). For similar reasons, our estimated $CH_4$ emission (1.4 Tg C yr⁻¹) was also smaller than that previously estimated[26]. Nevertheless, our estimating approach was from ref. 26, where their results were in good agreement with several other calculations when different methods or lake surface area datasets were used[3,4,26,27,35–40] (Supplementary Fig. 10). In this regard, our estimates based on more accurate lake boundaries documented in GLAKES dataset should be reasonable. Our calculations here further highlight the disproportionately large contributions of small lakes to global lake $CO_2$ and $CH_4$ emissions (25% and 37%), given their small share in the overall areal size of lakes (15%).

The mapping of lake variations during the past four decades enabled us to examine the changes in $CO_2$ and $CH_4$ emissions from global lakes throughout the study period (see Methods). Net increases in carbon emissions were found from the 1980–1990s to the 2010s, with +4.81 Tg C yr⁻¹ for $CO_2$ and +0.03 Tg C yr⁻¹ for $CH_4$, respectively (Fig. 4b). Small lakes contributed 45% and 59% to the net increases in lake $CO_2$

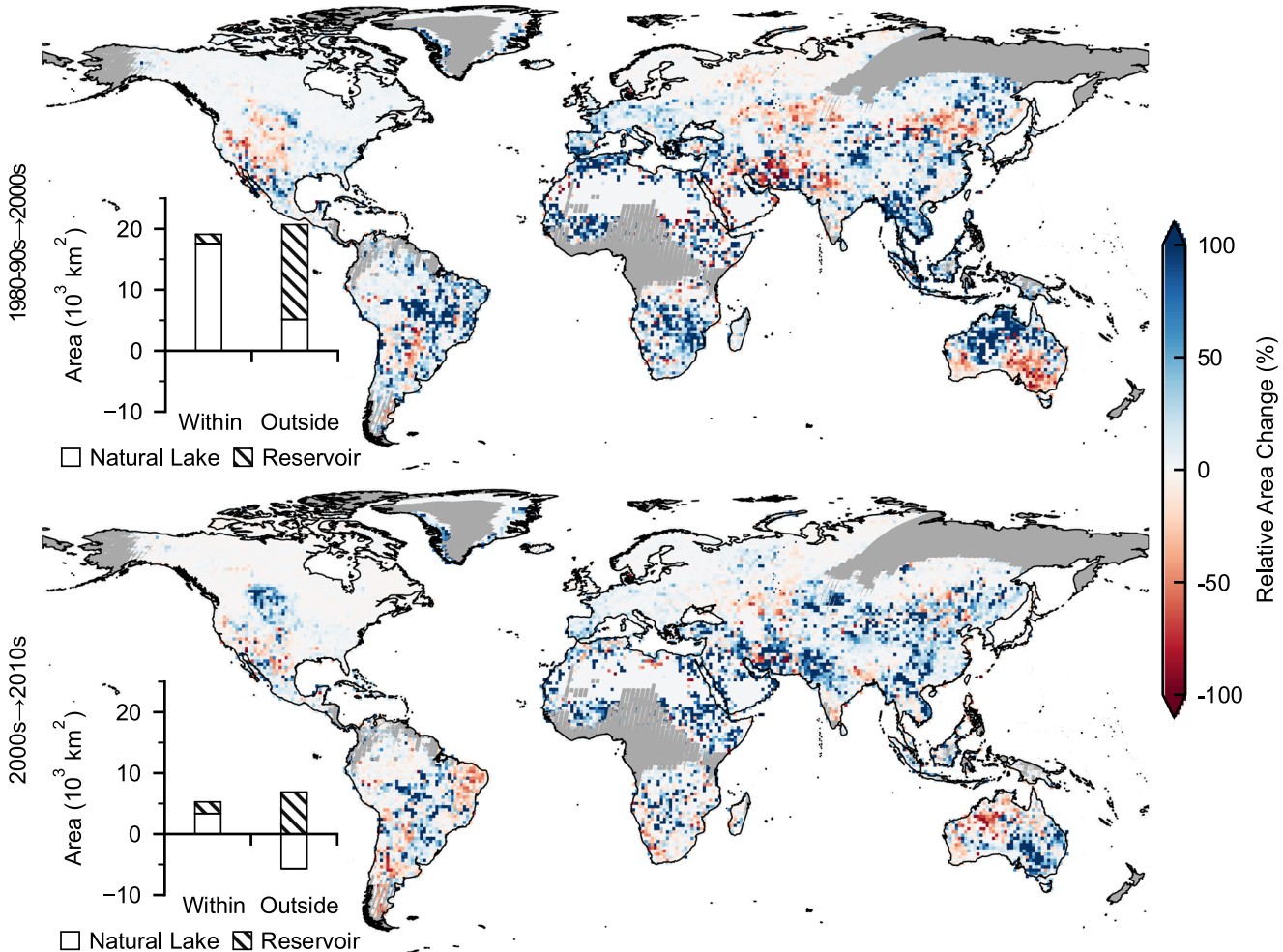

**Fig. 2 | Lake area changes across different periods (1980–1990s, 2000s, and 2010s).** Data were aggregated into 1° × 1° grid cells. The gray areas indicate regions with insufficient satellite coverage in the early periods; these regions were excluded from the analysis. Within each panel, the changes within and outside the glacial or permafrost regions are also presented, and the contributions of natural lakes and reservoirs are illustrated.

and $CH_4$ emissions over the entire study timeframe, respectively, and these contributions were similarly high across different periods.

## Discussion

We have developed a global lake dataset that comprises the maximum outlines of 3.4 million lakes over the past four decades. Overall, our GLAKES dataset shows marked improvements over previous global lake datasets, considering its advantages in global coverage (60°S–80°N), high spatial resolution (~30 m), long-term changes (four decades), spatiotemporal consistency (uniform mapping of global lakes instead of aggregation from different lake datasets), overall accuracy (overall accuracy >98.7% and MIoUs >88.7%), and the delineation of small lakes (lower limit as 0.03 km²). We have demonstrated an overall increase in inland lake areas worldwide over the past four decades. We corroborated previous findings regarding substantial lake expansions in glacial and permafrost regions[14], whereas we further revealed that such climate warming-induced changes were not a major contributor to global lake dynamics throughout the entire period. Instead, we demonstrated that human-regulated reservoirs contributed to more than half of the overall areal increases, highlighting the dominant role of human alterations on global water dynamics[41]. Our results also showed a net loss trend in lake area within endorheic basins (in endorheic basins, water does not flow into any sea); this result agrees with the findings obtained using GRACE-detected water storage measurements recorded between 2002 and 2016[10]. However,

our high-resolution mapping revealed the opposite trend when the desiccated Aral Sea was excluded; under this condition, we found continuously expanding small lakes in endorheic basins from the 1980–1990s to the 2010s (Supplementary Fig. 11).

Our study also provides an important update to previous lacustrine carbon emissions calculations using more accurate lake extents and offers detailed insights into changes in carbon emissions from global inland lakes over four decades. Nevertheless, we believed that the changes in $CO_2$ and $CH_4$ estimated herein represented conservative values, and the magnitudes of these changes would be higher when increased lacustrine eutrophication and expanded lakes of smaller size were incorporated into the gas exchange calculations[22,26,27,42,43].

Our detailed mapping of the dynamics of 3.4 million lakes can potentially be used to better characterize regional-to-global hydrological budgets, as the changes in evaporation and water storage induced by lake size variability have often been ignored in past studies[44,45]. In particular, the widespread expansion of the lake area can be synergistically analyzed with global increases in disastrous flood events[46], and the results reported herein provide critical information for assessing the possibly enhanced flood risks associated with lake dynamics. In addition, our dataset enabled a more thorough evaluation of the causes of the variations in surface water extent that may have previously been constrained by the completeness or quality of global lake mapping (especially for small lakes) to gain a more comprehensive perception of the impacts of climate change and anthropogenic

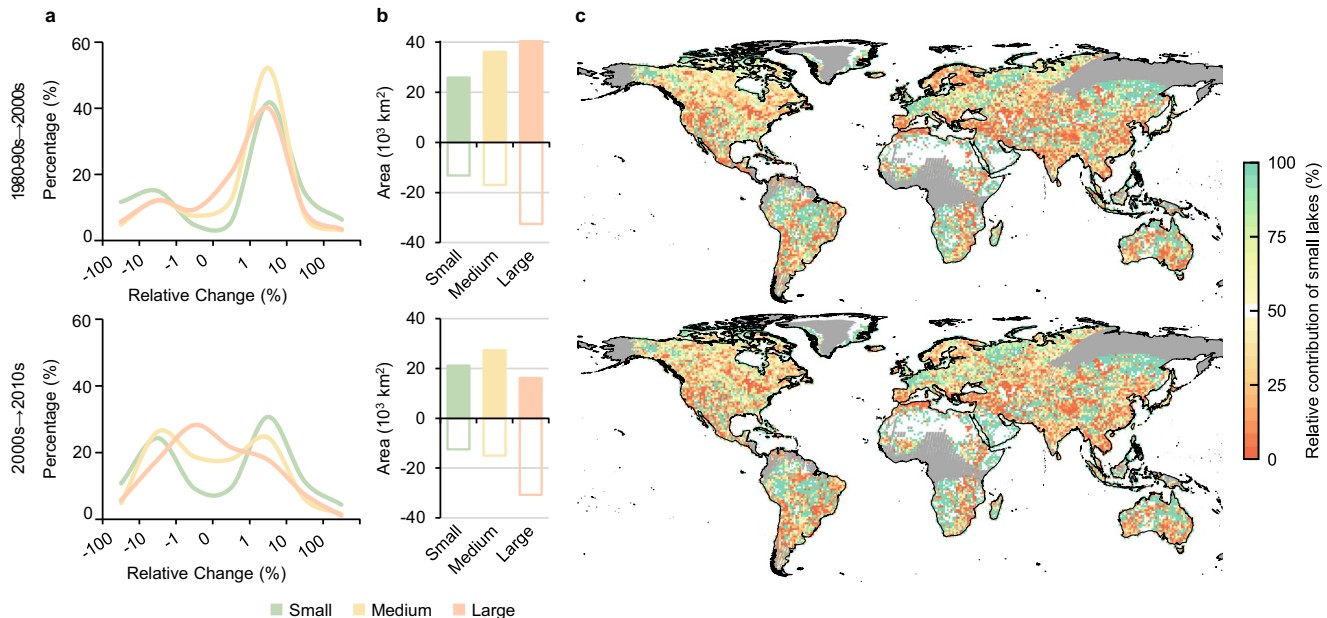

**Fig. 3 | Outsized role of small lakes. a** Histograms of relative areal changes derived for global lakes partitioned into different size groups between given time periods. **b** Absolute areal changes in lakes within different size groups between time periods, including both positive and negative changes. **c** Relative contributions of small lakes to lake size variability between different periods, estimated as the proportion of the absolute areal changes in small lakes within a given 1° × 1° grid cell (see Methods). The gray areas indicate regions with insufficient satellite coverage in the early periods.

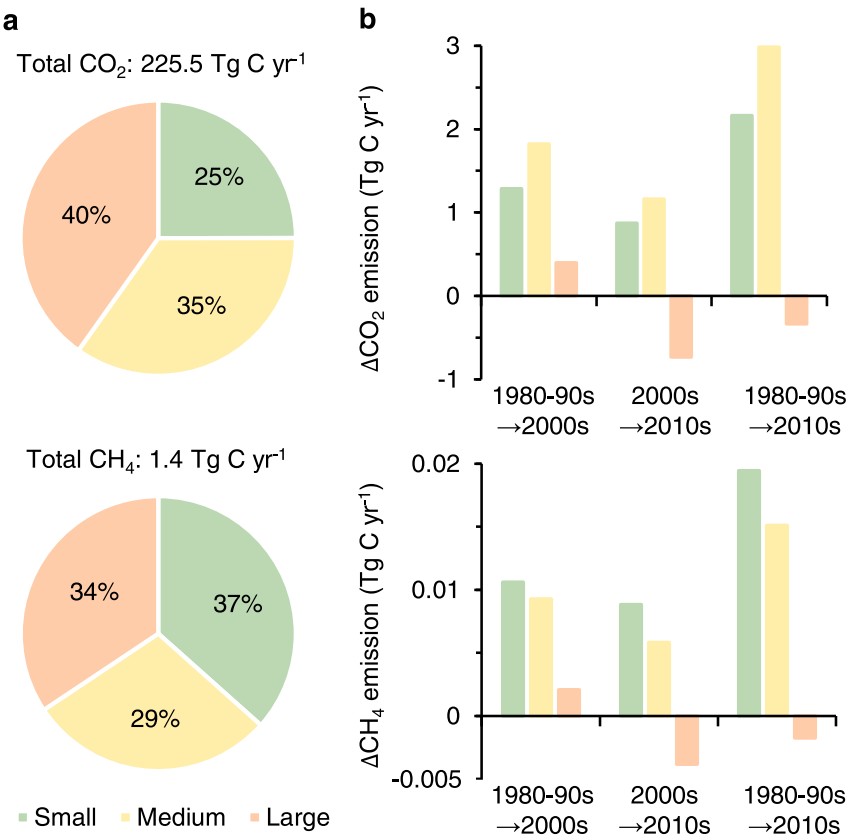

**Fig. 4 | Impacts of decadal lake changes on carbon emissions. a** Total CO₂ and CH₄ emissions from global lakes and the proportions derived from different sized lake groups. The gas emissions were estimated using the method proposed by ref. 26, by applying our GLAKES dataset. **b** Net changes in CO₂ and CH₄ emissions derived for lakes in different size groups across the three analyzed time periods.

activities. Furthermore, our established dataset is essential for quantifying various aspects of aquatic systems (e.g., freshwater species and nutrient/pollutant cycles) and identifying the potential transitions of ecosystem functions under climate change[47–49]. Our findings also underscore the urgent need for future management efforts focused on small lakes, given their crucial roles in regulating local hydrologic cycles and inland water-related carbon emissions.

## Methods

### Data sources

We used the Global Surface Water Occurrence (GSWO) dataset[12] as the source data to determine the areal extents of lakes. The GSWO dataset provides global (from 60°S to 80°N) documentation of the location and frequency of water occurrences over nearly four decades (1984–2019) and was generated using 30-m-resolution Landsat images. Extensive validation of the GSWO dataset was conducted at the global scale, over the whole study period, and among all Landsat sensors involved. The results demonstrated the high accuracy of the surface water delineation in the GSWO datasets (1 < % false water area detections and <5% missed water area) and, consequently, the ability to afford comparable, continuous, and consistent mapping spatially, temporally and across sensors[12]. In addition, we used the GSW monthly water history (MWH) data collection to calculate the occurrence of water in different periods over the past four decades. The smallest lake mapped in our study had a surface area of 0.03 km$^2$, corresponding to approximately 33 Landsat pixels. The GSW dataset was developed by the European Commission's Joint Research Centre and can be obtained from https://global-surface-water.appspot.com.

We used the Randolph Glacier Inventory 6.0 (RGI 6.0) glacier data[50] and ice sheet mass balance inter-comparison exercise (IMBIE) Rignot data[51] to determine the distributions of glacier-fed lakes. The RGI 6.0 data provide a global inventory of glacier polygons (see https://www.glims.org/RGI/rgi60_dl.html), and the IMBIE Rignot dataset includes an additional ice sheet polygon supplemental dataset representing Greenland (see http://imbie.org/imbie-2016/drainage-basins/). In addition, we used the permafrost distribution data provided by the National Snow and Ice Data Center of the National Aeronautics and Space Administration (NASA) to determine the locations of permafrost-fed lakes. This dataset provides the gridded permafrost distribution in percent area in the Northern Hemisphere[52]. The gridded permafrost data can be obtained through the NASA webpage (see https://neo.sci.gsfc.nasa.gov/view.php?datasetId=PermafrostNSIDC).

The HydroBASINS dataset[53] was used to identify endorheic lakes. HydroBASINS presents consistent and seamless watershed outlines at different scales (lev01–lev12) according to the Pfafstetter coding system, with a global spatial resolution of 15 arc-seconds. In this study, the lev12 watershed polygons were selected for analysis. In particular, HydroBASINS offers a variable called Endo to indicate whether each basin/subbasin is an endorheic basin. The HydroBASINS dataset was obtained from https://www.hydrosheds.org/page/hydrobasins.

The Georeferenced global Dam And Reservoir (GeoDAR) dataset[54] was used to distinguish reservoirs from natural lakes. The GeoDAR dataset, taking advantage of multi-source dam/reservoir inventories, provides global documentation of reservoirs with detailed attribute tables and well-georeferenced spatial locations. The GeoDAR dataset is available at https://doi.org/10.6084/m9.figshare.13670527.

We used the Gridded Population of the World (GPW) dataset[55,56] to investigate the relationship between the human population and lake area changes. The GPW dataset provides gridded population density estimations based on proportional allocation models utilizing population information gathered across global administrative units. Due to the long time span of our study, two GPW data versions were combined to establish population censuses reflecting different time periods. The GPW version-3 data (resolution: 2.5 arc-minutes) were used for 1990 and 1995, while the GPW version-4 data (resolution: 30 arc-seconds) were applied for 2000, 2005, 2010, 2015, and 2020. Then, the GPW data reflecting different years were averaged to represent the long-term population density status throughout the study period. The GPW data are accessible through the Socioeconomic Data and Applications Center (https://sedac.ciesin.columbia.edu).

### Mapping global lakes using deep learning

Deep learning has been widely used in many areas[57–60] and is proven to be a powerful and creative tool in detecting features of interest from satellite images[61–63]. A recent inspiring deep-learning application in remote sensing image processing was documented by Brandt et al.[64], who detected tree crowns by combining the U-Net model with sub-meter high-resolution satellite images. The U-Net model used in Brandt et al.[64] is a typical semantic segmentation technique that performs pixel-wise classification within an image for precise segmentation[65,66]. Compared to conventional classification tasks, U-Net yields not only the label category of a specific image but also its corresponding location. Upon the application of U-Net, Brandt et al.[64] managed to map more than 1.8 billion non-forest tree crowns (>3 m$^2$) in the West African Sahara and the Sahel, somewhat overturning the previous stereotype of tree scarcity in these dryland regions. Here, we modified the U-Net model developed by Brandt et al.[64] and transferred its application to global lake mapping. We expect a well-trained U-Net model to perform well when classifying lakes and rivers using GSWO images, as lakes and rivers are already highly distinguishable in visual examinations (Supplementary Fig. 3).

Here, lakes and rivers mainly indicate lentic and lotic water systems that are visible from space, including both permanent and seasonal waters. Lakes and rivers generally exhibit different features on GSWO images. Compared to lakes that usually have flat and oval outlines, rivers are typically long, meander and narrow in shape, which makes them distinguishable in most cases (see Supplementary Note 3). For seasonal water bodies, we implemented more carful examinations of those located around rivers and meanwhile span a large scale (such as floodplains), while keeping the small ephemeral water bodies as lakes. In addition, we also identify tidal flats surrounded by lakes and parts of wetlands as lakes because they are hard to separate from lakes via satellite observations[67]. Here we use parts because wetlands are usually covered with vegetation during the growing season, and thus cannot be captured by the GSWO images (or optical remote sensing images). GLAKES also contains constructed impounded water bodies (i.e., reservoirs) that are closely related to human activities. Notably, some agricultural fields are also included in our dataset, although the proportion may not be large, and further segregation is under process (see Supplementary Note 3). Last, we do not take into account lakes directly connected to the seas as the hydrological conditions and human interventions are intricate in those regions.

Building upon a fully convolutional neural network[68], the U-Net model is composed of various hierarchical convolution layers that are widely used in the semantic segmentation field for feature detection[60,66,69], vital for the extraction and segmentation of lakes from rivers. The convolution layer extracts features from an image in the previous layer and results in a less redundant output image called a feature map. Generally, the features learned by convolution layers transition from simple to more abstract ones as the level of the convolution layers increases[60,70,71], and these features are determined by the convolution kernels (i.e., an array of weights) that are learned automatically through backpropagation. Except for convolution layers, there are various structures that are also essential in the modified U-Net architecture of Brandt et al.[64], including activation function (enabling nonlinear classification), batch normalization (stabilizing and accelerating the training process), pooling (reducing data dimension and computation complexity), up-convolution (restoring the size of feature maps for precise localization) as well as

concatenation (combing the higher-level feature map with a lower-level one to better learn representation). In addition, the U-Net model adopts the overlap-tile strategy, which makes it possible to perform a seamless segmentation for large images without losing information about the divided border regions[66]. This makes U-Net particularly applicable to our goal of pixel-wise lake classification through GSWO images on a global scale. The specific structure of U-Net model is presented in Supplementary Fig. 1, Supplementary Table 2, and Supplementary Note 1.

The first step in deploying a deep-learning model is to prepare reliable labels for the training, validation, and test sets. In this study, lake labels were generated through an automatic extraction method followed by manual modification by combining several existing datasets, including the GSWO dataset, the Global River Widths from Landsat (GRWL) database[16], the OpenStreetMap Water Layer (OSMWL) dataset[72], and the HydroLAKES polygon dataset[17]. We used the GSWO map to mask land pixels and exclude low-confidence water pixels (some of which may be caused by the inherent classification errors of GSWO) with <5% occurrence (i.e., <5% of the Landsat observations were classified as water during the past four decades). Notably, in floodplains, the occurrence threshold was set to 30% instead of 5% to capture the 'core' portions of lakes and segment these portions from the larger-scale floodwaters (periodically occurring over a long time but lasting for less than one season (i.e., 25% occurrence) each year). We used the OSMWL and GRWL masking products to exclude ocean and river pixels. Since the GRWL dataset was developed using a limited set of Landsat images and was thus inadequate for representing the full coverage of rivers (this limitation is also applicable to the river data provided by OSMWL), we visually confirmed that all river masks used to prepare our labels were correct. Moreover, we gave higher priority to lakes that had already been detected in the HydroLAKES database by ensuring that the lake pixels within the HydroLAKES polygons would not be affected by the overlying river/ocean mask. Finally, extensive visual examinations were performed from one sample region to another, where some lakes in hydrologically complex regions were given more attention, such as those in the large river basins and floodplain regions. Manual corrections were performed mainly on the following two situations: (1) river residuals resulting from the absent coverage of the corresponding river masks and (2) river-connected lakes that required further division from river channels. Of all sample polygons, Case 1 polygons frequently occurred, which could take up ~10% of the total lake samples and thus require careful inspection. On the contrary, the percentage of Case 2 polygons was minor (far less than 1%). Ultimately, we would like to ensure that all extracted lake boundaries (i.e., the lake mask vectorization) matched well with the water/land interfaces isolated on the GSWO maps. Notably, the lower size limit of the samples was set as 0.03 km², according to our visual observations. The main objectives were to exclude small polygon residuals generated during this sample extraction procedure and to screen out small isolated agricultural fields.

We delineated a variety of globally distributed sample regions with varying sizes to create lake labels (Supplementary Fig. 2). Lakes in floodplains showed distinctive patterns compared to those in all other lake regions in the water-occurrence images; thus, we trained two separate deep-learning models (referred to as Floodplain Model and Normal Model, respectively) to extract these data better. In addition, visual explorations of the water-occurrence maps revealed several types of sample regions that presented different features (Supplementary Fig. 3). Specifically, we observed (1) relatively static lakes that exhibited high/moderate water occurrence (HO); (2) highly dynamic lakes with relatively low water occurrence (LO); (3) lakes spanning large spatial scales that were challenging to interpret using models due to the relatively large sizes of the lake objects

relative to the sizes of the modeled patches (LL); (4) lakes located alongside rivers that required more attention to be distinguished from rivers (AR); and (5) lakes within floodplains that often combined to form lake clusters (WF). Our sample regions were delineated and divided into these five different types based on visual observations. It should be noted that the region type was only representative of the major hydrological features of the lakes within the sample region bound, where lakes with distinctive patterns may also co-exist. Then, these five sample region types were proportionally allocated (stratified random sampling) to the training (60%), validation (20%), and test (20%) sets to ensure that the model's global representation of different lake patterns was balanced. In practice, not all five categories of sample regions were included in the Normal Model and Floodplain Model. Specifically, the Normal model consisted of types 1 (HO), 2 (LO), 3 (LL), and 4 (AR), since type 5 (WF) was not the target of the Normal Model. On the other hand, the Floodplain Model was made up of types 1, 3, 4, and 5. The reason why types 1, 3, and 4 were included for model interpretation was that the main patterns described by types 1, 3, and 4 were also observable within the regions defined by type 5, while type 2 was excluded because of the relatively high occurrence threshold (i.e., 30%) applied for the Floodplain Model. In summary, we delineated 754 sample regions for the Normal Model and 445 sample regions for the Floodplain Model; these region sets contained 90,512 and 71,170 lake labels, respectively. As seen in Supplementary Fig. 2a, the logarithmic sizes of these sample regions were approximated to a normal distribution, where the majority were on the order of magnitude of $10^2$–$10^3$ km², with a median area of $5.91 \times 10^2$ km² for the Normal Model and $5.69 \times 10^2$ km² for the Floodplain Model. The major principles considered when selecting samples included the selection of samples from different regions globally, the coverage of all typical hydrological conditions, and ensuring balance in the numbers of the five major region types.

In general, the size of sample regions was too large for the U-Net model to analyze the lake features within the region boundaries. Instead, a variety of patches (with a fixed size of 512 × 512 pixels) were randomly generated within each sample region, and the lake patterns within the patches were extracted and interpreted by the model. In addition, we applied the same local normalization method from Brandt et al.[64] for each patch (Supplementary Note 1).

According to our own research objectives, we made some essential modifications to the U-net model proposed by Brandt et al.[64]. First, we enlarged the patch sizes of the input images (i.e., training labels) from 256 × 256 pixels (based on submeter-spatial-resolution satellite data) to 512 × 512 pixels (based on the 30 m-spatial-resolution Landsat data) to enhance the capability of the model to capture the characteristics of large lakes. Additionally, the distance-weighted map between gaps used in the initial model for the segmentation of partially overlapped objects[64,66] was not considered in our loss function since we did not face this issue when using GSWO maps as the input images. In terms of the selection of hyperparameters, we kept the same type of loss function and optimizer as those used in the original model but made substantial changes to some other hyperparameters, such as batch size, epoch numbers, and iteration numbers (see all important hyperparameters documented in Supplementary Note 1), according to trial-and-error results. Likewise, we used the loss error for model selection (i.e., the model yielding the lowest loss error value using the validation labels was selected). In addition, we also introduced the mean intersection over union (MIoU) to assist model evaluation (which was not used in Brandt et al.[64]). MIoU is a widely used image segmentation performance indicator that fully considers true positives and false negatives[73]. The IoU of each class was calculated as the area of overlap divided by the area of union between the labels and predictions of that class. Then, the IoU from different classes was averaged to estimate MIoU.

After the selection for the final Normal and Floodplain Model, predictions were obtained for each 512 × 512-pixel patch in each GSWO map, resulting in a raw global lake classification map. Then, several postprocessing procedures were conducted to further enhance the classification accuracy. We first overlaid the OSMWL and GRWL masking products to exclude ocean and river pixels, similar to the procedure followed to prepare the labels. This step was necessary because our model performed well when distinguishing lakes from small rivers but exhibited some limitations with regard to oceans and large rivers, mainly induced by the relatively small patch size (i.e., 512 × 512 pixels) used for training. For coastal lakes, we further excluded lake polygons that directly intersected with the 10 m buffer of the global ocean boundaries defined by the OSMWL dataset; next, more detailed manual corrections were performed on near-coastal regions. For rivers, we minimized the residuals caused by the incomplete coverage of the GRWL and OSMWL river masks using the area ratio of the polygons representing the same lake before and after the application of the river masks (Supplementary Fig. 12). In general, if the area ratio was close to 1, only a small portion of the polygon was masked by the predefined river maps, indicating that the polygon was more likely to be a river-connected lake than a river. In contrast, if the ratio was close to 0, indicating that a large portion of the polygon was masked, this polygon could be considered to comprise the residuals of rivers (due to incomplete masking) at a higher confidence level. In practice, we excluded polygons with area ratios <0.8 (if these polygons were not within the boundaries of the HydroLAKES polygons) and conducted manual corrections on almost all large river basins to reduce such residual errors.

Except for these postprocessing procedures, the lakes from the Normal Model and the Floodplain Model were combined to generate our final version of global lake polygons (referred to as the GLAKES dataset). Specifically, given that both the Normal Model and Floodplain Model yielded lake predictions at global coverage, we applied predefined river buffer zones to determine whether the extracted lake polygons from the Normal Model or Floodplain Model should be used for our final GLAKES dataset. Specifically, for buffer zones flagged as "floodplain", lake polygons (within these buffer zones) extracted from the Floodplain Model were selected as a part of GLAKES dataset, while the corresponding outputs from the Normal Model were discarded. In contrast, for all remaining areas (including buffer zones flagged as "normal" and areas outside the river buffer zones), lake polygons from the Normal Model were included in our final dataset. The basic principle for the determination of the exact flag ("normal" or "floodplain") for each buffer zone is to measure the extent of seasonally flooded non-lake and non-river waters within the buffer zone. A 1 km buffer was applied to each vectorized polygon of global rivers documented in the GRWL Mask V01.01 product (https://zenodo.org/record/1297434#.YrvEzj5ByUk). Within each buffer, the area of seasonally flooded non-lake and non-river waters was calculated by summing the area of all GSWO pixels with occurrence <75% (to exclude the permanent and near-permanent water), except for those already being defined as rivers (by GRWL mask) and lakes (by rasterized HydroLAKES polygons). Finally, buffer zones where the ratio of their containing flooded area to the corresponding buffer area exceeded the flooding threshold were flagged as "floodplain". In this study, the flooding threshold was set as 0.1 through trial and error.

Similar to the generation of sample polygons, only predicted lakes with maximum surface water area >0.03 km$^2$ were included in our GLAKES dataset and were applied in further analysis. We partitioned the global lakes into three size groups (small: <1 km$^2$, medium: 1–100 km$^2$, large: >100 km$^2$) to estimate the numbers and area of lakes in different size groups (Fig. 1) and to further examine the varying change patterns in lake sizes and carbon emissions during the past decades (see below).

## Accuracy assessments and comparisons with previous global lake datasets

We used independent test labels to further evaluate our modified U-Net model. Notably, the Normal Model and the Floodplain Model were evaluated separately. We visually inspected how well the predicted patches matched the corresponding labels (Supplementary Fig. 3) and calculated error matrices to assess the accuracies of the predictions of different lake size groups at the pixel level. Overall, the mapped lake extents exhibited high accuracy levels in both the lake count and lake area for Normal Model and Floodplain Model (Supplementary Fig. 2b), and the predicted lake areas showed overall accuracies >98.7% and MIoUs >88.7% (Supplementary Table 1). Meanwhile, a slight and systematic underprediction of label area could be observed in our models (with Percent Bias (PBIAS) < 0), while the magnitude of PBIAS in terms of label count was smaller compared to that of label area.

In terms of the accuracy among different lake sizes, the omission errors (i.e., lakes classified as non-lake areas) were relatively high for small lakes (23.5% for Normal Model and 21.2% for Floodplain Model) compared to those obtained for medium and large lakes. It is noteworthy that the setting of the predefined cutting threshold (0.03 km$^2$) for lake samples should probably be responsible for such kind of issue (see Supplementary Note 3). Nevertheless, the commission errors (i.e., non-lake areas classified as lakes) were much smaller than the omission errors among all lake size groups, indicating that our mapped lake extents appeared to represent conservative estimations. Indeed, the lake change analysis was performed only within the boundaries defined by our GLAKES dataset (see below); therefore, the associated impacts of the classification errors (particularly the omission errors) should be limited. In addition, we further examined the model performance from the following perspectives: (1) polygon-based assessment at different size scales; (2) performance in small lakes with a finer division of size range; (3) performance among five different region types; and (4) the spatial distribution of model performance across the globe (see Supplementary Note 2, Supplementary Figs. 13, 14, and Supplementary Tables 3–5 for detailed information).

We further compared the areas and numbers of lakes in our lake database (i.e., GLAKES) with the corresponding values reported in several previously established global lake datasets[17–20] (Supplementary Fig. 4a). The Global Lakes and Wetlands Database (GLWD) developed by Lehner et al.[18] is a combination of several global or regional lake datasets (see GLWD documentation). The dataset is organized into three levels, focusing on large water body polygons, smaller water body shorelines, and the rasterized extents of potential wetlands. Based on the Shuttle Radar Topography Mission Water Body Data[74] (for most lakes between 56°S and 60°N), CanVec[75] (for the majority of North American lakes), as well as other lake datasets (see HydroLAKES documentation), the most widely used global lake database HydroLAKES[17] was developed, along with intensive automated and manual corrections. The HydroLAKES dataset contains ~1.42 million individual lake polygons with surface area >0.1 km$^2$. Downing et al.[19] focused on the scaling relationships and size distributions of lakes; these factors were tested in different regions and integrated into a global Pareto distribution model to extrapolate the global lake extent. By using the GeoCover™ program circa 2000 based on Landsat 7 images in combination with a corresponding water-extraction algorithm, Verpoorter et al.[20] developed the GLObal WAter BOdies database (GLOWABO), which contains 117 million lakes with the area of the smallest lake polygons set at 0.002 km$^2$. In this study, we made a more detailed spatial comparison using the latest version of the global lake dataset (i.e., HydroLAKES) at both the 1° × 1° grid scale (Supplementary Fig. 4b) and the pixel-level scale (Supplementary Fig. 5). Note that what we compared here was actually the lake area bounded by the lake polygons of each dataset (if provided). The explicit meaning of what the lake polygons represent (average/maximum/snapshot extent) may be different since they were all generated from different methods and with different objectives.

Comparing only lakes with area >0.1 km², the total numbers and area of lakes contained in GLAKES are similar to those of lakes contained in the HydroLAKES dataset (Supplementary Fig. 4a). However, the inclusion of 1.91 million extra smaller lakes (<0.1 km²) in our dataset resulted in a lake area of $1.05 \times 10^5$ km² (30.2%) larger than that indicated by the small-lake group in the HydroLAKES dataset[17]. Moreover, we found a substantial number of missing lakes in eastern Canada and Scandinavia in our dataset compared to HydroLAKES, as well as lake overestimations with varying degrees in other regions, such as Siberia and major river floodplains (Supplementary Figs. 4b, 5). These discrepancies could be raised for many reasons, either to be responsible for the inherent limitation of the GLAKES or otherwise HydroLAKES. One example is the inability of GLAKES (or GSWO) to capture lakes that are seasonally ice-covered throughout the year and heavily vegetated in the remaining month, which is typical for some small and shallow lakes in places such as the Canada Shield. The large values of GLAKES could also be partially explained by the inclusion of some agricultural fields (used to be lakes) or accidentally large floodplains. In contrast, for HydroLAKES the constraint of its composing dataset (e.g., MODIS MOD44W water mask and SRTM Water Body Data) in detecting small lakes may be the possible reason for the lake underestimation in some regions. Overall, both GLAKES and HydroLAKES have their own strengths and limitations in terms of lake coverage, but what distinguishes GLAKES is its global consistency (not mosaic from different datasets), higher resolution (better characterizes water/land interface), the reflection of multidecadal lake extent (not snapshot on short time period) as well as the inclusion of smaller lakes (<0.1 km²). This is significant for the long-term monitoring of the lake surface water area dynamics. As a comparison, the Global Lake area, Climate, and Population (GLCP) dataset provides annual time series lake surface area records from 1995 to 2015 for all HydroLAKES polygons[76]. Nevertheless, GLCP faces the same limitations as HydroLAKES in terms of the lake size limit and spatial consistency. Besides, since the HydroLAKES polygons did not represent the maximum water extent, a fixed buffer zone around lakes was generated in GLCP for area estimation, which might result in fallaciously inclusion of water coverage that did not belong to the target lakes or missed detection of water area due to the insufficient coverage of the buffer outlines. On the other hand, large discrepancies were found between the estimates obtained using GLAKES and several other previously reported estimates[18-20] (Supplementary Fig. 4a), especially for small lakes (the relative differences reached >50%). The overestimation of Downing et al.[19] is likely due to the reason that the statistics derived for lakes <0.1 km² were not determined from explicit lake mapping but from extrapolated values. In contrast, the overestimation of the GLOWABO dataset[20] probably resulted from the inclusion of non-lake polygons such as rivers, given that the disentanglement of lakes from rivers was never mentioned in their documentation. As for Lehner and Döll[18], similarly, their estimation of small lakes may be constrained by the underlying data sources composing GLWD.

## Analyzing lake size changes over four decades

We estimated the water probability-weighted area, compared the water probability-weighted area among different periods (i.e., 1980–1990 s, 2000s, and 2010s), and investigated the significance of small lakes on global lake dynamics. The probability-weighted lake area ($Area_{pw}^{lake}$) was calculated as follows: $Area_{pw}^{lake} = \sum (Area^{pixel} * WO^{pixel})$, where the $Area^{pixel}$ is the area of each pixel (i.e., 900 m²) constrained by the lake boundary defined in our GLAKES dataset and $WO^{pixel}$ is the corresponding water occurrence (WO) in each pixel. For each period, a $WO^{pixel}$ value was calculated using the same method as that outlined by Pekel et al.[12] by normalizing the number of water presence ($N_w$) incidences against the number of valid observations ($N_{vo}$) within a period. In practice, both $N_w$ and $N_{vo}$ can be derived using the GSW MWH data collection. In the MWH dataset, each pixel was assigned to one of the

three values (0: no data, 1: non-water pixel, and 2: water pixel); for each pixel within a given time period, we estimated $N_w$ as the number of images corresponding to a value of 2 and set $N_{vo}$ as the number of images corresponding to pixel values >0.

We aggregated the probability-weighted areas into $1° \times 1°$ grid cells and calculated their relative changes between different periods (Fig. 2). We further examined the relative contribution of small lakes to lake variability within each grid cell (Fig. 3). The relative contribution was defined as the proportion of absolute area changes identified in the small-lake group (i.e., $|\Delta A_{small}|$) to the total areal changes (i.e., $|\Delta A_{small}| + |\Delta A_{large+medium}|$) identified within a grid cell (i.e., $|\Delta A_{small}| / (|\Delta A_{small}| + |\Delta A_{large+medium}|)$). Notably, grid cells with insufficient satellite coverage in each period (fewer than 30, 20, and 20 valid observations in the 1980–1990s, the 2000s, and the 2010s, respectively) were excluded from the cross-period comparison (i.e., grid cells in eastern Russia and central Africa) (Supplementary Fig. 6).

## Lake changes under different geographic conditions

To investigate the associations between lake areal changes and population density, we averaged the GPW population density data within each $1° \times 1°$ grid cell. We then compared the gridded lake area changes among different population classes; we conducted two types of comparisons: (1) all lakes and (2) only small lakes (Supplementary Fig. 9).

We identified glacier-fed lakes as lakes that spatially intersected with the 1 km buffers surrounding the glacier polygons obtained from the RGI 6.0 and IMBIE Rignot datasets, following the same method as ref. 14. It should be noted that the main focus of this method is lakes experiencing recent detachment from glaciers within a few decades or large supraglacial lakes that are highly distinguishable on long-term satellite observations. Likewise, a spatial intersection approach was applied to identify lakes that received water supply from permafrost (i.e., intersection with selected $0.1° \times 0.1°$ grids whose inside permafrost coverage was >10%, determined by using permafrost distribution data sourced from the National Snow and Ice Data Center) (Supplementary Fig. 6f). Similarly, we used the GeoDAR dataset to recognize the location of global reservoirs and conducted the same intersection approach to distinguish reservoirs from natural lakes in our GLAKES dataset. In total, we extracted 24,514 reservoir polygons in our dataset, accounting for 16.9% of the global lake area. The numbers of small, medium, and large reservoirs are 17,301, 6813, and 400, respectively.

To determine endorheic and exoreic lakes, we used the information provided in the HydroBASINS dataset. Basins with the 'Endo' variable attribute >0 were considered endorheic basins; thus, all lakes falling into these regions were considered endorheic lakes.

## Estimation of lacustrine carbon emissions and changes

Following the method outlined by Holgerson and Raymond[26], we classified our lake dataset into seven logarithmic size classes and used the estimates of the size-dependent mean flux estimates[26] directly to calculate the $CO_2/CH_4$ emissions. We first multiplied the mean $CO_2/CH_4$ flux by the total area of lakes (in all-time maximum) classified to estimate the total gas emissions for each size class and subsequently calculated the total emissions for global lakes. We repeated this step by using the upper/lower bounds of the flux values to estimate the ranges of global lacustrine $CO_2$ and $CH_4$ emissions. We also estimated the carbon gases emitted from global lakes in different periods (1980–1990s–2010s) as well as the corresponding emission changes over time. Similar to the procedures described above, we combined the size-dependent $CO_2/CH_4$ fluxes with the probability-weighted lake area to estimate global lake carbon emissions for each period.

## Uncertainty and limitations

Several uncertainties and limitations should be acknowledged in this study, both during the process of lake mapping along with related change analysis of lake area and carbon emissions. In lake mapping,

these could be further categorized into the following major sources: lake definition, auxiliary datasets, U-Net model, and postprocessing. The temporal change of probability-weighted lake area among different time periods, otherwise, may be influenced by seasonal lake dynamics. As for the estimation of global carbon emissions as well as their long-term changes, the accuracy of our results was closely related to the representativeness of the average emission rates used for global upscaling, the impacts of lake dynamics at shorter timescales, and the quantification of emissions through different pathways (for $CH_4$). Please see Supplementary Note 3 for detailed information.

## Data availability
The entire GLAKES dataset and labels used to train the U-Net model are available under the accession code: https://doi.org/10.5281/zenodo.7016548.

## Code availability
The code associated with the training and validation of the U-Net model, and the prediction of global lakes by using the trained model are also accessible through the same link documented in data availability.

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

## Acknowledgements

L.F. acknowledges the National Natural Science Foundation of China (No. 41971304), and J.L. is supported by the Strategic Priority Research Program of the Chinese Academy of Sciences and the Henan Provincial Key Laboratory of Hydrosphere and Watershed Water Security (No. XDA20060402). L.F. also acknowledges the Shenzhen Science and Technology Innovation Committee (No. JCYJ20190809155205559), the Stable Support Plan Program of the Shenzhen Natural Science Fund (No. 20200925155151006), and the Shenzhen Science and Technology Program (No. KCXFZ20201221173007020). R.F. is supported by the research grant DeReEco (34306) from Villum Fonden. J.T. is financially supported by Swedish FORMAS mobility grant (2016-01580) and acknowledges support from Lund University strategic research area Modelling the Regional and Global Earth System, MERGE.

## Author contributions

X.P. and Q.L.: methodology, data processing and analyses, and writing; L.F.: conceptualization, methodology, funding acquisition, supervision, and writing. Y.X. and E.M.: involved in the data processing and analysis. J.T., X.L., R.C., R.F., M.B., X.C., L.G., J.L., C.Z., W.L., and B.B. participated in interpreting the results and refining the manuscript.

## Competing interests

The authors declare no competing interests.
