## [Peer Review File · Nature Communications]

Mapping global lake dynamics reveals the emerging roles of small lakesReviewer #1 (Remarks to the Author):

General comments

I quite liked this paper and find it well suited for Nat. Comms. It has broad implications (geographic and biogeochemical) and the approach taken well justified in my view. It takes a multi-decadal perspective to show that, over the past 40 years or so, the surface area occupied by very small lakes has increased and that of large lakes (actually mostly just the Aral sea) has decreased. The manuscript is divided into two quite distinct sections, one purely hydrological (lake mapping) and the other exploring the consequences of the changes in lake surface area on lake greenhouse gas emissions. In both cases, I thought the manuscript did quite a good job. Nevertheless, I feel the robustness of some of the assertions made need to be explored more fully.

Lake mapping

My main concern in this section is the stability of the detection/classification algorithm through time. It is difficult to imagine that the quality of the satellite images haven't changed over the 40 years. I did not look up the GSWO data set in details on which the present analysis is based but it would seem necessary to me to show that the temporal trend in lake coverage is not due to the deep learning algorithm performing differently in the different decades examined. I would therefore recommend parsing the validation separately in the different decades. Incidentally, some details of the specific deep learning approach used here (the U-Net model as applied by Brandt et al.) should be given and not simply referenced.

Incidentally, some points are made that I feel should be even more underlined forcefully. For example, their lake global coverage of is very close to that estimated by Downing et al. (2006) if we only consider the overlapping size classes. It is also very similar to Feng et al. (2015) estimate. However, it is much lower than the impossibly high GLOWABO numbers.

Greenhouse gas emission

There is no doubt that small lakes emit per unit area much more than larger lakes and the manuscript makes an important that the areal expansion of small lakes disproportionately affects the global lake emission numbers. However, I felt there was too much reliance on a single study (Holgerson and Raymond) and would have like some sensitivity analysis done on that. The authors actually acknowledge that (line 622) but there are many published relationships between GHG emissions and lake size (e.g. Rasilo et al. 2014) and a comparison of results with different such equations would enhance the robustness of their conclusions. Similarly, the occurrence of several negative CO₂ fluxes if a single equation is applied to all lakes is problematic. While there are cases of negative CO₂ fluxes in lakes, they are not particularly related to lake size but rather to eutrophication. Nevertheless, I appreciate that they ultimately used the average flux within binned log classes.

Reference not cited in their manuscript

1. Rasilo, T., Prairie, Y. T. & Giorgio, P. A. Large-scale patterns in summer diffusive CH₄ fluxes across boreal lakes, and contribution to diffusive C emissions. *Global Change Biol* 21, 1124–1139 (2014).

Reviewer #2 (Remarks to the Author):

Review of NCOMMS-21-48587

Summary

This manuscript presents an interesting spatio-temporal analysis of a new high quality lake dataset, with insights on changes in global lake area over the past 4 decades. The results generally align with previous global scale analyses of both surface water and greenhouse gas emissions, reaffirming the well-established disproportionate roles of small water bodies in surface water hydrology and biogeochemistry. This analysis goes further than similar previous efforts to distinguish patterns with more nuance - natural lakes vs reservoirs, separating patterns within and outside glacier and permafrost regions. Although I don't have the expertise to evaluate the method used to distinguish glacier and permafrost affected lakes, I found this is an interesting aspect of the study. Additionally, datasets on population density and type of drainage basin were used to provide further insights on different decadal patterns.

There is room for improvement in terms of clarity and accuracy. Some of the interpretations in comparison to other datasets are not always supported with appropriate evidence – in particular the minimum lake size in existing datasets is smaller than in this new dataset (contradicting some statements), which also seems to affect the comparison of ghg emission estimates. The paper would also be much stronger with a clearer operational definition of lakes to help explain the assumptions and applicability of the classification model, and enable appropriate re-use of the data.

A potential important advancement from this paper is the creation of a new spatially explicit high resolution global lake dataset – although the 2014 GLOWABO dataset from Verpoorter et al is higher resolution, it is not easily findable or readily accessible, and also not multi-temporal. The data availability statement says the data will be provided, but there are not enough details provided to assess whether it will be shared with adherence to FAIR principles (<https://www.go-fair.org/fair-principles/>).

My comments are provided below:

Main suggestions for improvement

1. A clearer description of how "lakes" are operationally defined would improve this manuscript and provide a helpful foundation for future users of the GLAKES dataset. Although it is established in the introduction that lakes and rivers have distinct hydrology and biogeochemical processes, the criteria for how this distinction is made based on surface water extent/shape/geometry could be specified in more detail. The roles that other features such as wetlands play in surface water extent & variability is somewhat overlooked (eg. ~line 52-54) – is the underlying assumption that all surface water can be classified as lake vs river, or is the distinction lake vs. non-lake? Despite such complications, the analysis presented for upscaling GHG emissions is a good example for why it is valuable to distinguish lentic and lotic waterbodies in surface water datasets - is it possible to elaborate on other appropriate potential future uses of GLAKES given the assumptions used for the model and to classify the training data (~lines 240-244)?

2. The novelty of GLAKES in comparison to previous datasets seems a little overstated. Specifically, compared to GLOWABO (Verpoorter et al 2014), which is described as having a minimum lake size of 0.002 km², with the smallest objects based on filtering anything smaller than 9 pixels. This contradicts statements that GLAKES is the first global lake dataset to include lakes smaller than 0.1 km² without statistical extrapolation (lines 72-73; lines 527-530; lines 545-547).

3. Statements about comparisons to other lake GHG estimates should be reconsidered (lines 187- 201). Making these comparisons is obviously complicated because of the differences in underlying data on surface water extent datasets and different categorizations used for size classes (as mentioned), however as written there appear to be some misleading/confusing statements.

3a. The abstract concludes GLAKES leads to higher GHG emissions (line 34) however the comparison to Holgerson and Raymond 2016 (line 190) shows smaller estimates than

previous, which seems contradictory. Extended data figure 9 also shows emission estimates from the present study are smaller than most other estimates. Is the conclusion about higher emissions from lake area increases about changes over time between the periods analyzed (not differences with previous datasets)? If so this could be stated more clearly.

3b. The lake surface area estimates from Holgerson and Raymond include more than 2 additional logarithmic size classes smaller than the minimum size in GLAKES - 0.0001 to 0.001 km² (from statistical extrapolation), and 0.001-0.01 km² (based on Verpoorter et al 2014 for >0.002 km² and estimated for 0.001-0.002 km²). While it is certainly possible that the surface area extents for these smaller classes are overestimated, they are not included in GLAKES at all and therefore would be an obvious source of the difference (~line 191). The 2 other more recent studies included in Extended Fig 9 (DeSontro et al 2018 and Li et al 2020) both use GLOWABO without extrapolation beyond the lower limit of 0.002 km², however neither appear to be in better agreement (line 194-195) with the GLAKES emission estimates compared to Holgerson and Raymond 2016.

3c. The CO₂ estimate of 194 Tg C is compared to combined number for both CO₂ and CH₄ in Holgerson and Raymond (should be only 571 Tg from CO₂ not 583 on line 190).

4. The paper would be improved with at least a brief description of the U-Net model to explain why it is applicable for distinguishing lakes from rivers/other surface water features, especially for readers less familiar with such classification models. Is it possible to be more specific about what kind of features make lakes and rivers highly distinguishable in visual examinations (line 401-402), and explain why that might justify using the U-Net model for this kind of classification problem?

5. I think the identification/ description of the 5 categories of sample regions (lines 430-436) is a strength of the analysis – the discussion could be improved by describing how well the model performs in each of those categories. Is it possible to show an example of each of the 5 types of sample region, or at least label the region types shown in Extended data fig 2? What is the size range of the sample regions used to create the training data?

6. Data availability statement does not provide enough detail to evaluate how dataset will be archived, made available or how to access. Should be revised with adherence to FAIR principles (<https://www.go-fair.org/fair-principles/>), e.g. publishing the dataset in a repository that issues a unique identifier.

7. It would help the reader if the time scale of variability (decadal?) was stated more prominently (e.g. line 1480-149, caption for Fig 4), especially since seasonal and interannual fluctuations are mentioned in the introduction. Consider rephrasing “dynamics” to “changes” or “trends” in surface water extent, given the well documented but complicated relationships between GHG emissions and changes in water level possible on much shorter timescales (eg <https://doi.org/10.5194/bg-9-2459-2012>, <https://doi.org/10.1672/07-98.1>, <https://doi.org/10.1002/2015JG003283>, <https://doi.org/10.3390/atmos10050269>). The manuscript would also be stronger if the discussion addressed how the results may be affected by changes/dynamics at those shorter timescales, for example frozen lakes becoming only seasonally ice-covered.

Other/minor suggestions

1. Line 30 – maybe specify “all six continents analyzed” since Antarctica is excluded
2. Line 64 – should “water productivity” be “primary productivity” ?
3. Line 82 - How was 0.03 km² threshold determined for minimum size? Are lakes smaller than 0.03 km² not present in GSW, not captured by the model, or excluded for other reasons? It seems somewhat overstated to say improved resolution “solves” issues associated with mis-accounted small lakes in previous datasets since lakes/surface water can be mis-classified for reasons other than spatial resolution (e.g. optical complexity, forest canopy, mixed pixels, shadows).
4. Line 88 – Isn't the GSWO dataset limited to 80 deg N? Consider specifying max northern extent, and/or include latitude range where dataset is described in methods.
5. It would help the reader to explain why P1 is two decades and P2 and P3 are each

one decade. In Figure 2, it is somewhat confusing what dates/date ranges are being compared in top and bottom panels - It may be more clear to give beginning and end year of each range.

6. Line 143 & elsewhere – clarify if “global inland regions” are used interchangeably with the 1 deg grid cells?
7. Line 144 – maybe specifying “net” area increase if that is what is meant here
8. Line 147 – consider modifying sub-heading to specify what patterns small lakes have an outsized role in
9. Line 163 – unclear whether the “small lake-dominated regions” referenced here is about areas with just many small lakes or areas where small lakes dominate the variability. (As written I interpret as the former, but the referenced figure and rest of the paragraph seem to be about the latter.)
10. Line 166 – unclear what ‘dynamic amplitude’ means here
11. Extended data Figure 8 – consider adding a reference line at $y = 0$
12. Were the normal and floodplain models both applied globally, or was there a criteria/dataset used for distinguishing normal and floodplain areas?
13. Extended data figure 9a – error bar on Marotta et al 2008 appears misaligned
14. Typo in caption for Fig 4? Line 189 says the emissions estimates were calculated following methods in ref 23 (Holgerson and Raymond 2016) but caption for Fig 4 says method is from ref 22 (Oertli et al 2002, about species area relationships)
15. Line 233-235 should also discuss or at least reference DelSontro et al 2018 (ref 24) which addresses eutrophication/productivity effects on emissions.
16. Line 383 – is there evidence that these lake area changes are actually caused by human interventions without other natural variability? consider rephrasing “impacts of human interventions” to something like relationship with or co-occurrence with human populations
17. Line 444 – how was “coverage of all possible hydrological conditions” determined?
18. Line 465 – should “flood” be floodplain?
19. Lines 478-484 – can this part be explained more clearly? (or examples shown in extended figures?)
20. Line 539 – found the wording here confusing, whether “missing” lakes in eastern Canada and Scandinavia were commission or omission errors in GLAKES
21. Line 603 – what is the source of the air temperature data?
22. Line 616-618 – consider including newer references for work on reservoir ghg emissions eg. <https://doi.org/10.1093/biosci/biw117>, <https://doi.org/10.1029/2019JG005600>

Reviewer #3 (Remarks to the Author):

Attachment on the following page

Review Report

Article's reference: NCOMMS-21-48587

Title: Mapping global lake dynamics reveals the emerging roles of small lakes

Authors: Xuehui Pi, Qiuqi Luo, Yang Xu, Jing Tang, Xiuyu Liang, Enze Ma, Ran Cheng, Rasmus Fensholt, Martin Brandt, Xiaobin Cai, Luke Gibson, Junguo Liu, Chunmiao Zheng, Weifeng Li, Brett A. Bryan, Lian Feng

Date accepted to review: 2022/02/17

Date review submitted: 2022/03/07

Summary

This study introduces a new dataset named GLAKES, which depicts the surface extent dynamics of 3.4 million lakes at the global scale from the 1980s to present day. The authors delineated individual lake polygons by adapting a deep-learning model to conduct a supervised classification of existing grids of long-term (1984-2019) surface water occurrence. The deep learning model outputs were enhanced by removing delineated water bodies that substantially overlap with river channels and seawater. With this new dataset of lake polygons, changes in global lake density and area were then computed between three time periods (1980-90s, 2000s, and 2010s). The authors estimate that lakes expanded across all continents over the past four decades, mostly due to artificial reservoir building, and that small lakes account for most of the variability in global lake area. Leveraging these estimates, this study also determines that global carbon emissions from natural lakes increased over the same time periods, most of which attributed to small lakes.

Outstanding features

I consider the following to be the outstanding features of this research work:

- Leverages state-of-the-art data sources and models to create a near-comprehensive dataset of individual lakes, delineating lakes down to 0.03 km² in surface area (compared to 0.1 km² for the most widely used lake polygon dataset at present, HydroLAKES).
- Provides the first spatially-explicit estimate, to the reviewer's knowledge, of global long-term lake extent dynamics.
- Updates estimates of global carbon emissions from natural lakes, predicting a net increase in lacustrine emissions over time.
- Confirms the outsized role of small lakes in global lacustrine carbon emissions.

Key points and results

I consider the following to be the key points and results of this research work:

- Identifies 3.4 million individual lakes, totalling $3.2 \times 10^6 \text{ km}^2$ (2.2% of the global land area).
- While displaying high overall accuracy, the lake delineation model significantly underestimates the extent of small lakes (omission rates of 19.3% and 23.7% for lakes inside and outside floodplains, respectively) and overestimates the extent of large lakes (commission rates of 22.3% and 11.6%, respectively).
- The great majority of lakes are small ($< 1 \text{ km}^2$), but most of the global lake area is attributed to large lakes ($> 100 \text{ km}^2$).
- The estimates of lake number and area resulting from this study align closely with HydroLAKES (Messenger et al. 2016), another free-to-access dataset, confirming in the reviewer's opinion that two other existing studies (Verpoorter et al. 2014 and Downing et al. 2010) yielded substantial overestimates.
- Global lake area increased from the 1980s-90s to the 2000s and from the 2000s to 2010s, most of this expansion stemming from the expansion of glacier- and permafrost-fed lakes as well as by artificial reservoir building.
- Small lakes ($< 1 \text{ km}^2$) showed the highest temporal variability in extent compared to larger lakes.
- Estimates that natural lakes emit 194 Tg C yr^{-1} of CO_2 and 7.2 Tg C yr^{-1} of CH_4 . These new figures are smaller than previous estimates because most previous models relied on Verpoorter et al.'s surface water extent dataset, which overestimated global lake surface area.
- Due to the increase in global lake area, carbon emissions from natural lakes increased by $3.02 \text{ Tg C yr}^{-1}$ for CO_2 and $+0.31 \text{ Tg C yr}^{-1}$ for CH_4 , most of these increases stemming from small lakes.

Originality and significance

Will the work be of significance to the field and related fields? How does it compare to the established literature? If the work is not original, please provide relevant references.

This study is a significant and valuable contribution to the literature, and the conclusions are original. It will likely become a baseline dataset for many subsequent studies, thus enabling the advancement of our understanding of the role of lake in regional and global hydrological and biogeochemical cycles, as well as the impact of human activities of lake ecosystems and the services they provide globally. I congratulate the authors for their substantial effort and resulting contribution.

This work represents a substantial step forward compared to the established literature, foremost because it quantifies the spatio-temporal dynamics of lake surfaces over the past four decades. Whereas other studies have quantified inland water dynamics as a whole (e.g., Pekel et al. 2016⁴

which was used by this study, the coarser dataset by Klein et al. 2017⁵, Pickens et al. 2020⁶, and most recently Pickens et al. 2022⁷), what sets this study apart is that it aims to focus exclusively on the dynamics of lakes as individual entities (delineating the shoreline of individual lakes rather than continuous grids of surface water cover). This matters because lacustrine ecosystems differ fundamentally from other inland waters like rivers, seasonally inundated floodplains and other types of wetlands — in their hydrology, biogeochemistry, biodiversity, and their contribution to people and society.

In terms of a static dataset of lake polygons, this study is also an advancement compared to the two most established studies/datasets on the topic: HydroLAKES⁸ and GLOWABO⁹ (I do not include the GLWD¹⁰ here as it can fairly be considered as a product from a previous generation).

It is an advancement over HydroLAKES for three main reasons:

1. It has a higher resolution and thus provides a more comprehensive accounting of small lakes.
2. The lake polygon dataset provides a more temporally integrated view of lake extent globally because it is derived from Earth observation data over decade. By contrast, the bulk of HydroLAKES polygons, geographically (< 60N), were delineated from a short satellite mission (SRTM) over 11 days in February 2000.
3. It is likely more spatially consistent than HydroLAKES. HydroLAKES results from the compilation and harmonization from over five original data sources at different temporal and spatial resolutions while all polygons in GLAKES were delineated with the same processes using a common data source (despite differences among Landsat sensors over time).

It is also an advancement over GLOWABO, despite GLOWABO including lakes down to 0.002 km² (9 x 30-m pixels), for three main reasons:

- GLOWABO was never publicly released. Therefore, its application was limited and it underwent little external validation.
- GLOWABO polygons were also extracted from Landsat imagery but only from the year 2000 ± 3 year, thus representing a snapshot in time rather than a long-term picture of lake extent.
- The summary statistics provided by GLOWABO suggest that it substantially overestimates global lake cover, at least by a factor of two. This is probably due to the lack of discrimination between lakes, rivers, and wetlands in that dataset (at least no mention of such a distinction was made in Verpoorter et al. 2014⁹).

Validity

Does the work support the conclusions and claims, or is additional evidence needed?

Overall, this study relies on high-quality data sources and implements proven methods implemented in previous studies (for the deep-learning model¹ and for the carbon emission upscaling²). I commend the authors for their effort. However, several points deserve additional justification and clarification.

1. I am unclear about the nature of the labels used in training the deep learning model/supervised classification. The use of robust and reproduceable labels is obviously foundational to the validity of the model outputs, particularly given that the authors do not use independent data sources to validate it. From my understanding, the authors created the labels by masking land in the GSWO (<30% and <5% of water cover frequency out of the valid observations during the past four decades inside and outside floodplains, respectively) and masking ocean and river pixels (using the OSMWL and GRWL datasets, respectively), while retaining pixels overlapping with HydroLAKES polygons. Following this first step, the authors conducted “extensive visual examinations and necessary manual postprocessing corrections were performed to ensure that all extracted lake boundaries (i.e., the lake mask vectorization) matched well with the water/land interfaces isolated on the GSWO maps“ (P18L420). The extent of manual postprocessing corrections is not entirely clear in this description: was every lake polygon/label checked? Of those, how many were manually corrected? And based on what criterion? For instance, in mapping tree crowns, Brandt et al.³ manually delineated individual tree crowns based on the following criteria: “two conditions had to be fulfilled for a crown to be marked during the manual labelling process: (a) the NDVI value had to be clearly higher than the surrounding (only trees have green leaves in the dry season), and (b) a shadow had to be seen”. In the case of lakes, similar challenges exist, what level of permanence did the authors consider to qualify as characterizing water/land interfaces? In areas of extremely dense lake coverage where differentiating between rivers and lakes is arduous (e.g., across the Canadian Shield), and where the GRWL tends not to include river channels, how did the authors delineate individual lakes (vs. clusters of lakes)?

For the sake of reproducibility, I encourage authors to provide intermediate products of this analysis, including the initial mask pre- manual corrections and the final labels/polygons used in training the dataset.

2. Related to my previous point, what the lake polygons represent hydrologically should be more explicitly defined. Importantly, is the intent that the polygons represent the average or maximum lake extents? Do the polygons represent permanent lakes or are seasonal lakes also included (and what maximum degree of seasonality is included)? Because lake extent dynamics are only analyzed within the delineated polygons, I assume that the polygons represent their maximum extent. If the polygons are an all-time maximum, I recommend that

the overall statistics of the area of lakes and comparisons with other datasets (e.g., Extended Data Fig. 3) be based on the probability-weighted area rather than on the maximum extent. For example, I was really surprised to see >150 more lakes with a surface area >100km² in GLAKES than in HydroLAKES as most of these lakes are rather well-known. Only when I inspected the polygons did I realized that many of those large lakes are >100 km² because it is apparently their maximum extent which was represented.

3. The omission and commission rates are quite high for small and large lakes, respectively. However, the reasons for and implications of this limitation are not discussed. I am not familiar with the inner workings of the deep learning model employed in this study, but would it be possible to train two separate models that would be catered for differently-sized lakes? At least, a quantitative estimate of the uncertainties in the predictions would strengthen this study.
4. Multiple studies have previously highlighted the outsized role of small lakes for several global processes and this study adds a salient piece to the puzzle. Nonetheless, I suggest that the findings from this study regarding the role of small lakes need to be caveated in two main ways:
 - a. It makes sense that small lakes display higher relative temporal variability (smaller volume to area ratio, greater sensitivity to catchment, etc.). However, it is important to highlight potential biases in quantifying the amplitude of this variability that are due to scale. Because the size of individual pixels is large compared to the total area of small lakes, the default/random variability in extent of small lakes is higher. In a lake spanning 0.05 km², a single 900-m² pixel going from wet to dry leads to a ~2% change. I wonder how similar the variability in extent would be between small and large lakes if it were evaluated with equal relative pixel sizes (e.g., 5 m for a 0.05 km² lake and 5 km for a 50 km² lake).
 - b. The outsized role of small lakes in driving carbon emissions may be largely driven by the fact that reservoirs were excluded from the calculations. My guess is that the inclusion of emissions from large reservoirs may change this finding. I suggest that this be mentioned. Moreover, all sections discussing the carbon emission estimates should use the term “natural lakes” rather than just “lakes” (as the latter refers to both lakes and reservoirs by default in the manuscript; P4L70). The reporting of the findings would thus be clearest to readers who may not read the entire piece (i.e., most readers), avoiding that this work be miscited. This is not currently clear in the abstract for example.

Data & methodology, appropriate use of statistics and treatment of uncertainties

Are there any flaws in the data analysis, interpretation and conclusions? Do these prohibit publication or require revision? Is the methodology sound? Does the work meet the expected standards in your field? Is there enough detail provided in the methods for the work to be reproduced?

There are no major flaws in the data analysis, interpretation and conclusions that would prohibit publication, though see my previous and subsequent remarks which I believe warrant revisions. This work meets standards in the field in terms of data and methodology, but its explanation and treatment of uncertainties could be further strengthened.

Data sources

To my knowledge, the data sources used in this study are some of the best available datasets for this application. The only exception to this statement is the OpenStreetMap Water Later (OSMWL) dataset, whose provenance is largely undocumented and quality yet to be demonstrated. Nonetheless, I am confident that using (or producing) a better dataset would not fundamentally change the conclusions of this study.

Explanation of methodology and treatment of uncertainties

- The main text currently contains no discussion on sources of uncertainty, I highly encourage that a substantial paragraph or section be dedicated to the main sources and extents of uncertainty affecting this study. For instance, discussions on the difficulty of disentangling lakes from temporarily inundated floodplains or agricultural fields would be needed.
- Overall, the deep learning model deserves to be explained more clearly and at greater length. Additional explanations should be provided as to how the deep learning model functions and a table of hyperparameters should be provided in the supplementary material. Additional information on why some decisions were taken is needed (see my specific comments further on), which could be complemented by information on the sensitivity of model outputs to these decisions.
- An assessment of the spatial distribution of uncertainties would be valuable. In which regions does the model perform better or worse? This could be implemented through spatial cross-validation.
- Currently, the labels used for assessing the model performance are only partly independent, as they were created through the same semi-automatic process using the same data source (GSWO). Adding a truly independent validation, based on a different sensor and/or higher resolution imagery (e.g., using Sentinel data as was most recently done by Pickens et al.⁷), would strengthen the evaluation of the model performance.
- The evaluation of model performance (Extended Data Table 1) is only provided at the pixel level whereas this study also produced a polygon dataset. To better grasp uncertainties, I encourage the authors to provide a polygon-based performance assessment

(omission and commission at the lake entity level as well as measures of fit and bias between test polygons and output polygons).

Dataset formatting

I opened and visualized the dataset with no difficulty. It fits the description provided in the manuscript. The documentation is clear. Although it may not be in its final form, I suggest that the authors also provide a license for the data, and that the database be available as a Shapefile and/or geopackage to enable a greater range of users to access these data.

Conclusions

Do you find that the conclusions and data interpretation are robust, valid and reliable?

Provided that the findings on lacustrine emissions be more explicitly re-framed as that of natural lakes, I believe that the conclusions and data interpretation are robust, valid and reliable.

Presentation

The manuscript is written clearly overall, with minor edits needed (detailed in the following section).

The figures are very informative and aesthetically pleasing. Some minor edits are needed for the legends and captions to be clear, complete and accurate.

Miscellaneous comments and suggested improvements

Please find below suggestions, many minor, for strengthening this research work during revisions:

- P2L29 “explicit extents and changes”: this relates back to my previous comment on the meaning of the GLAKES polygons. “explicit” extent is not clear, is the dataset about average or maximum extent?
- P2L30: “Lake area increased across all six continents”, an explicit mention of the dates used in the study here would be useful.
- P2L32 “global lake *areas*” and “variabilities”. I believe that both of these should be singular.
- P2L35: “Our findings illustrate the emerging roles of small lakes in regulating local inland water variabilities and greenhouse gas emissions.” The results suggest that small lakes don’t only regulate *local* inland water (extent?), but also the *global* dynamics of surface water extent and greenhouse gas emissions.
- To be specific, the authors could refer to carbon emissions rather than greenhouse gas emissions, as N₂O was not studied here (despite its role as a GHG¹¹).
- P3L39: “...underpin vital ecosystem function and services” for the sake of thoroughness, please provide a citation.
- P3L41: Woolway, R. I., Kraemer, B. M., Lenters, J. D., Merchant, C. J., O’Reilly, C. M., & Sharma, S. (2020). Global lake responses to climate change. *Nature Reviews Earth &*

Environment, 1(8), 388-403. Would be more adequate (global scale) for this statement than O’Beirne

The following study would also be relevant to this statement:

Grant, L., Vanderkelen, I., Gudmundsson, L., Tan, Z., Perroud, M., Stepanenko, V. M., Debolskiy, A. V., Droppers, B., Janssen, A. B. G., Woolway, R. I., Choulga, M., Balsamo, G., Kirillin, G., Schewe, J., Zhao, F., del Valle, I. V., Golub, M., Pierson, D., Marcé, R., ... Thiery, W. (2021). Attribution of global lake systems change to anthropogenic forcing. *Nature Geoscience* 2021, 1–6. <https://doi.org/10.1038/s41561-021-00833-x>

- P3L57: “*Estimates of the global extent of lakes are available*” would be a more exact description of these datasets.
- P4L65: “However, these available global assessments...” it is not entirely explicit in this paragraph what “these assessments” refer to. Please clarify.
- P4L71: “We used deep learning to identify lakes smaller than the minimum mapping unit for all previous global lake datasets (0.1 km²).” Verpoorter et al.⁹ describe mapping lakes down to 0.002 km² (nine Landsat pixels).
- P4L81: “Deep learning makes it possible to detect lakes as small as 0.03 km² (corresponding to approximately 33 Landsat image pixels)...”. This may be due to my misunderstanding of what is involved in the deep learning algorithm, but I am not clear about how deep learning itself enables detecting small lakes. This statement deserves additional explanation/justification.
- Figure 1: This is a nicely done and informative figure. Good job! For panel *b*, given that a degree square has a substantially different surface area depending on latitude, I recommend that the lake extents be expressed as limnicity (% land area covered by lakes) rather than absolute area (km²).
- P6L103: For ease of reading, I suggest simply using the actual years (1980-90s, 2000s, and 2010s) throughout the manuscript rather than P1, P2 and P3.
- P10L189: It is not clear from the methods whether this estimate is determined with the total area of the polygons (that would mean the maximum water extent based on my observation of the dataset, and would thus potentially represent an overestimate) or the probability-weighted area over a given period.
- Figure 4. I suggest using the same color set for small, medium and large lakes here as in panels a and b of Figure 3.
- P12L226: Very interesting finding!
- P12L235: The following citations are quite relevant here for discussion:
 - Keller, P. S., Marcé, R., Obrador, B. et al. Global carbon budget of reservoirs is overturned by the quantification of drawdown areas. *Nature Geoscience*. 14, 402–408 (2021). <https://doi.org/10.1038/s41561-021-00734-z>

- Johnson, M. S., Matthews, E., Bastviken, D., Deemer, B., Du, J., & Genovese, V. (2021). Spatiotemporal methane emission from global reservoirs. *Journal of Geophysical Research: Biogeosciences*, 126, e2021JG006305. <https://doi.org/10.1029/2021JG006305>
- Bridget R. Deemer, John A. Harrison, Siyue Li, Jake J. Beaulieu, Tonya DelSontro, Nathan Barros, José F. Bezerra-Neto, Stephen M. Powers, Marco A. dos Santos, J. Arie Vonk, Greenhouse Gas Emissions from Reservoir Water Surfaces: A New Global Synthesis, *BioScience*, Volume 66, Issue 11, 1 November 2016, Pages 949–964, <https://doi.org/10.1093/biosci/biw117>
- P12L235: “Our detailed mapping of the dynamics of 3.4 million lakes can potentially be used to better characterize regional-to-global hydrological budgets...” A brief mention could also be made of the possibilities for a more thorough assessment of the causes of surface water extent variations given that the present study is (understandably, given its scope) cursory in its assessment of the effects of climatic and anthropogenic influences.
- P15L349: Here it is worth pointing out as an assurance to the reader that Pekel et al. demonstrated remarkable continuity in the accuracy of the GSWO among sensors and, consequently, through time, which is paramount to the validity of this analysis.
- P15L353: How was the lower limit of 33 pixels determined? Was a sensitivity analysis conducted in terms of model performance with higher and lower size limits? How does performance change towards that lower limit of 0.03 km² (versus for all small lakes together)?
- P17L407: It seems that the reference for this dataset is incorrect. Ref 49 points to Yamazaki, D. et al. A high-accuracy map of global terrain elevations. *Geophysical Research Letters* 44, 5844-5853 (2017).
However, this reference does not include any reference to OSMWL. I believe that the authors may be referring to Yamazaki, D., Ikeshima, D., Sosa, J., Bates, P. D., Allen, G. H., & Pavelsky, T. M. (2019). MERIT Hydro: a high-resolution global hydrography map based on latest topography dataset. *Water Resources Research*, 55, 5053– 5073. <https://doi.org/10.1029/2019WR024873>
And the following dataset: http://hydro.iis.utokyo.ac.jp/~yamadai/OSM_water/index.html
- P17L409: How were the thresholds of 5% and 30% for non-floodplain and floodplain regions determined?
- P18L426: I believe that the correct word would be “compared to” rather than “with” in “floodplains showed distinctive patterns with all other lake regions...”
- P18L428: It makes sense that additional region types were identified, yet their relationship to the normal and floodplain region types (and the multiple uses of the term region or region type) is unclear and there is no reference elsewhere of these “region sub-types” in the manuscript (in the figures or in the performance reporting). How are model uncertainties distributed among those regions?

- P18L438: were the regions allocated by formal stratified random sampling? What is the size distribution of labels (Extended Data Fig. 1 suggests that they are of different sizes)?
- P19L446: “For each sample region, a variety of patches were randomly generated and used for model training, and the same local normalization method was also utilized for each patch.” I am unclear about the meaning of this sentence. I suggest that patches, regions (including region types and sub-types), and labels be more clearly defined and differentiated. Normalization was also not mentioned beforehand. A workflow diagram could help the readers to grasp this analysis, as it is crucial to the quality of the model.
- P19L461: please define MIoU in the manuscript itself.
- P21L498: It seems from Extended Data Fig. 1 that the model systematically underpredicts label area. I suggest reporting mean percent bias for the normal and floodplain models and addressing this pattern in the text.
- P22L518: The GLWD is a minor component of HydroLAKES. The vast majority of lake polygons from 56°S to 60°N are from the Shuttle Radar Topography Mission (SRTM) Water Body Data (SWBD), and all lakes in Canada (62% of all lakes in the database) are from CanVec. This clause may be corrected or removed.
- P22L527: “the statistics for lakes smaller than 0.1 km² (if any were available) were represented by values extrapolated from larger lakes” I mentioned it in a previous comment, but I was under the impression that GLOWABO provided statistics for lakes at least down to 0.01 km² if not 0.002 km².
- P22L538: “Moreover, we found a substantial number of missing lakes in eastern Canada and Scandinavia in the HydroLAKES dataset and lake overestimations with varying degrees in other regions (Extended Data Figs. 3b & 4); these errors could be due to uncharacterized seasonal or interannual dynamics and other unsourced uncertainties from the inherited datasets.”
 - In its current form, the first clause of this sentence is not entirely clear. Figure 3b suggests that HydroLAKES overestimates lake density and area compared to GLAKES in eastern Canada and Scandinavia (although the current sentence structure suggests the opposite) but underestimates lake prevalence in several other regions compared to GLAKES (foremost in Siberia and along major river floodplains e.g., Mississippi, Amazon, and Ganges-Brahmaputra).
 - Regarding the second clause of the sentence, I recommend caution in characterizing the discrepancies between HydroLAKES and GLAKES as necessarily errors.
 - In Canada, many lakes that are present in HydroLAKES but absent from GLAKES do exist. Canadian data in HydroLAKES was sourced from CanVec, which itself was built by digitizing topographic maps that, in my experience, are rather reliable, particularly in southern regions. I expect that GLAKES is unable to detect a lot of smaller and shallower lakes because their water surface is frozen/snow-covered for 4-9 months/year and heavily

vegetated for the other months of the year (see example maps below, which fall within the frame of Figure 4 and show HydroLAKES outlines and GSWO). Many lakes in this region are transitioning to wetlands, so some may also be dry for part of the year.

- In Siberia above 60N, the underestimation of lake prevalence by HydroLAKES is real. This underestimation stems from the fact that polygons in this region were generated from the MODerate resolution Imaging Spectroradiometer (MODIS) MOD44W Collection 5 water mask, which has a resolution of only 250-m. A considerable proportion of surface water bodies between 10 and 25 ha (≤ 4 pixels) were not detected in this region due to the coarse pixel size of the MODIS instrument (each pixel is ~ 6.25 ha in area)¹².
- The overall higher number of small lakes in GLAKES compared to HydroLAKES South of 60N may also stem from two other reasons:
 - a size discontinuity in the SWBD that was used to produce HydroLAKES in this region. The minimum size threshold used by technicians for digitizing a waterbody was a length of 600 m (and a width of 90 m). The largest lake missing due to this constrain is theoretically a round lake of 570 m in diameter spanning ~ 25 ha, and the proportion of omitted lakes increases with decreasing lake area.
 - The inclusion of additional seasonal waterbodies, including some that cannot really qualify as lakes, particularly in river floodplains (e.g., lake_id 69 is $> 2,000$ km² but appears to result from a flood), and of flooded fields for agriculture (lake_id 410210 are human-flooded fields).
- A broader point with these examples is that, while GLAKES may not be always right (and lakes found in HydroLAKES may thus not be errors), is that a strength of GLAKES is its global coherence/consistency (as I pointed out in previous sections), compared to datasets that result from aggregating multiple datasets. This is worth highlighting.

- P23L553: The area of each pixel is 900 m²
- P23L563: In Pekel et al. 2016, changes in water occurrence between epochs is matched by month (see quote from the original publication below) to avoid artefacts stemming from unequal detection among months and satellite coverage. Is the current computation approach immune from this issue?

"Change in water occurrence intensity between two epochs (16 March 1984 to 31 December 1999, and 1 January 2000 to 10 October 2015) was also produced (Extended Data Fig. 6a). This is derived from homologous pairs of months (that is, the same months

contain valid observations in both epochs). The occurrence difference between epochs was computed for each pair and differences between all homologous pairs of months were then averaged to create the surface water occurrence change intensity map. Areas where there are no pairs of homologue months could not be mapped. The averaging of the monthly processing mitigates variations in data distribution over time (that is, both seasonal variation in the distribution of valid observations, temporal depth and frequency of observations through the archive) and provides a consistent estimation of the water occurrence change. This map shows where surface water occurrence increased, decreased or remained invariant between the two epochs."

- P24L579: Here and elsewhere in the manuscript, it would be worth caveating that lakes identified with this method is mostly focused on "lakes that recently (probably within the last few decades") detached from glaciers due to glacial retreat as well as larger supraglacial lakes that are persistent enough to be visible on multi-year mosaics.", as explained in Shugar et al.¹³
- P24L583: "the percent area of permafrost *within lake polygon boundaries* was 10%"?
- P24L587: GEODAR authors report including 23,680 dam points (often, multiple dams can be associated with a single reservoir) and 20214 reservoir polygons. How was the intersection conducted to extract 24,514 reservoirs in GLAKES?
- P25L595: Please provide a table summarizing the equations used.
- P25L603: What was the data source used to determine air temperature?
- P25L610: Does this imply that lakes were not classified among the logarithmic size classes for computing CH₄ emissions? Please clarify.
- P26L628: By combining, do you mean that you computed the product of the probability-weighted lake areas and lake-specific CO₂/CH₄ areal fluxes, then summed across lakes?
- Extended Data Fig. 1. Label area unit should be km², not km³
- Extended Data Fig. 4. In light of my previous comment on possible issues with GLAKES in Canada, I recommend altering the language of the title of this figure to a more neutral tone about comparing.
- Extended Data Fig. 8. Please indicate the meaning of the box plot components in the legend.
- Extended Data Figure 9. A few suggestions below the comparison to be more informative:
 - o For the different estimates to be more comparable, I recommend either writing the minimum size of lakes included in each study or maybe showing these as 2D plots showing the total lake area/number determined in each study on one axis and the estimated emission on the other axis. Alternatively, different bar plots could be colored depending on which baseline lake dataset was used.
 - o I am not an expert in biogeochemistry, but do all these studies include the same types of fluxes (diffusive and/or ebullitive? See Deemer et al.¹¹)?

- Finally, the error bars represent different intervals (e.g., Monte-Carlo approach for Holgerson et al. to produce a 95% interval, vs. a min-max fluxes for this study) so this should be explained in the caption.

Please indicate any particular part of the manuscript, data, or analyses that you feel is outside the scope of your expertise, or that you were unable to assess fully.

Deep learning algorithms and biogeochemistry are not part of my primary area of expertise

I enjoyed reading this piece and hope that my comments and suggestions will be helpful to the authors. Best of luck for continuing to improve this valuable study.

References for peer-review report

1. Brandt, M. *et al.* An unexpectedly large count of trees in the West African Sahara and Sahel. *Nature* **587**, 78–82 (2020).
2. Holgerson, M. A. & Raymond, P. A. Large contribution to inland water CO₂ and CH₄ emissions from very small ponds. *Nat. Geosci.* **2016** *93* **9**, 222–226 (2016).
3. Brandt, M. *et al.* An unexpectedly large count of trees in the West African Sahara and Sahel. *Nature* 1–5 (2020) doi:10.1038/s41586-020-2824-5.
4. Pekel, J.-F., Cottam, A., Gorelick, N. & Belward, A. S. High-resolution mapping of global surface water and its long-term changes. *Nature* **540**, 418–422 (2016).
5. Klein, I., Gessner, U., Dietz, A. J. & Kuenzer, C. Global WaterPack – A 250 m resolution dataset revealing the daily dynamics of global inland water bodies. *Remote Sens. Environ.* **198**, 345–362 (2017).
6. Pickens, A. H. *et al.* Mapping and sampling to characterize global inland water dynamics from 1999 to 2018 with full Landsat time-series. *Remote Sens. Environ.* **243**, 111792 (2020).
7. Pickens, A. H. *et al.* Global seasonal dynamics of inland open water and ice. *Remote Sens. Environ.* **272**, 112963 (2022).
8. Messenger, M. L., Lehner, B., Grill, G., Nedeva, I. & Schmitt, O. Estimating the volume and age of water stored in global lakes using a geo-statistical approach. *Nat. Commun.* **7**, 13603 (2016).
9. Verpoorter, C., Kutser, T., Seekell, D. A. & Tranvik, L. J. A global inventory of lakes based on high-resolution satellite imagery. *Geophys. Res. Lett.* **41**, 6396–6402 (2014).
10. Lehner, B. & Döll, P. Development and validation of a global database of lakes, reservoirs and wetlands. *J. Hydrol.* **296**, 1–22 (2004).
11. Deemer, B. R. *et al.* Greenhouse Gas Emissions from Reservoir Water Surfaces: A New Global Synthesis. *Bioscience* **66**, 949–964 (2016).
12. Carroll, M. L., Townshend, J. R., DiMiceli, C. M., Noojipady, P. & Sohlberg, R. A. A new global raster water mask at 250 m resolution. *Int. J. Digit. Earth* **2**, 291–308 (2009).
13. Shugar, D. H. *et al.* Rapid worldwide growth of glacial lakes since 1990. *Nat. Clim. Chang.* **2020** *1010* **10**, 939–945 (2020).

Dear Editor and reviewers:

Thank you for your letter and for the reviewers' comments on our manuscript (manuscript number: NCOMMS-21-48587). These comments are very valuable and helpful for revising and improving our manuscript. We have addressed these comments carefully and have made revisions according to reviewers' suggestions one by one.

Response to the reviewer 1:

General comments:

Point: I quite liked this paper and find it well suited for Nat. Comms. It has broad implications (geographic and biogeochemical) and the approach taken well justified in my view. It takes a multi-decadal perspective to show that, over the past 40 years or so, the surface area occupied by very small lakes has increased and that of large lakes (actually mostly just the Aral sea) has decreased. The manuscript is divided into two quite distinct sections, one purely hydrological (lake mapping) and the other exploring the consequences of the changes in lake surface area on lake greenhouse gas emissions. In both cases, I thought the manuscript did quite a good job. Nevertheless, I feel the robustness of some of the assertions made need to be explored more fully.

Response: Thanks for your encouraging comments. We have made itemized revisions to address your concerns.

Specific comments:

Point 1: My main concern in this section is the stability of the detection/classification algorithm through time. It is difficult to imagine that the quality of the satellite images haven't changed over the 40 years. I did not look up the GSWO data set in details on which the present analysis is based but it would seem necessary to me to show that the temporal trend in lake coverage is not due to the deep learning algorithm performing differently in the different decades examined. I would therefore recommend parsing the validation separately in the different decades. Incidentally, some details of the specific deep learning approach used here (the U-Net model as applied by Brandt et al.) should be given and not simply referenced.

Response 1: Thanks for your insightful suggestion. For the first point, the deep-learning classification algorithm was performed on the GSWO dataset that spanning the whole study period (i.e.,1984-2020) to generate our GLAKES dataset, which was then used for the calculation of the probability-weighted area of each lake for the three periods (i.e., 1980-90s, 2000s, and 2010s). Therefore, there is no need to concern about the discrepancy in the performance of the U-Net model along time since we only used one

set of global lake boundaries to calculate the surface area changes among different periods.

For the second point, according to the documentation of Pekel et al. (2016), extensive validation of the GSWO dataset has been conducted at global scale, over the whole study period and among all involving Landsat sensors. The results demonstrated the high accuracy of the surface water delineation in the GSWO datasets (1<% false water area detections and <5% missed water area) and, consequently, the ability to afford comparable, continuous and consistent mapping spatially, temporally and across sensors. The above explanation has been added to the description section of the GSWO dataset in our revised manuscript (lines 383 to 388), as an assurance to readers that GSWO achieved high accuracy throughout different time periods, to prove the validity of our temporal change-related analysis.

As for the U-Net model, we have added more details for your information: “Deep learning has been widely used in many areas (Krizhevsky, Sutskever and Hinton 2012, Hinton et al. 2012, Sutskever, Vinyals and Le 2014, LeCun, Bengio and Hinton 2015), and are proven to be a powerful and creative tool in detecting features of interests from satellite images (Ma et al. 2017, Weiss, Jacob and Duveiller 2020, Reichstein et al. 2019). A recent inspiring deep learning application in remote sensing image processing was documented by Brandt et al. (Brandt et al. 2020), who detected tree crowns by combining the U-Net model with submeter high-resolution satellite images. The U-Net model used in Brandt et al. (Brandt et al. 2020) is a typical semantic segmentation technique that performs pixel-wise classification within an image for precise segmentation (Yu et al. 2018, Ronneberger, Fischer and Brox 2015). Compared to the conventional classification tasks, U-Net yields not only the label category of a specific image, but also its corresponding location. Upon the application of U-Net, Brandt et al. (Brandt et al. 2020) managed to map more than 1.8 billion non-forest tree crowns (>3 m²) in the West African Sahara and the Sahel, somewhat overturning the previous stereotype of trees scarcity in these dryland regions. Here, we modified the U-Net model developed by Brandt et al. (Brandt et al. 2020) and transferred its application to global lake mapping. We expect a well-trained U-Net model to perform well when classifying lakes and rivers using GSWO images, as lakes and rivers are already highly distinguishable in visual examinations (Extended Data Fig. 3).

Building upon a fully convolutional neural network (Long, Shelhamer and Darrell 2015), the U-Net model composes of various and hierarchical convolution layers that are widely used in semantic segmentation field for feature detection (LeCun et al. 2015, Ronneberger et al. 2015, Liu 2018), vital for the extraction and segmentation of lakes from rivers. The convolution layer extracts features from an image in the previous layer and results in a less redundant output image called feature map. Generally, the features

learned by convolution layers transition from simple to more abstract ones as the level of the convolution layers increase (Zeiler and Fergus 2014, LeCun et al. 2015, Ribeiro, Lazzaretti and Lopes 2018), and these features are determined by the convolution kernels (i.e., an array of weights) that are learned automatically through backpropagation. Except for convolution layers, there are various structures that are also essential in the modified U-Net architecture of Brandt et al. (Brandt et al. 2020), including activation function (enabling nonlinear classification), batch normalization (stabilizing and accelerating the training process), pooling (reducing data dimension and computation complexity), up-convolution (restoring the size of feature maps for precise localization) as well as concatenation (combing the higher-level feature map with a lower-level one to better learn representation). In addition, the U-Net model adopts the overlap-tile strategy, which makes it possible to perform a seamless segmentation for extremely large images without losing information about the divided border regions (Ronneberger et al. 2015). This makes U-Net particularly applicable to our goal of pixel-wise lake classification through the GSWO images at a global scale.

Specifically, the U-Net model comprises two major parts: a contracting path for feature interpretation and a near symmetric expanding path for location identification, leading to a u-shaped architecture that enables pixel-to-pixel classification (Ronneberger et al. 2015). In the contracting path, the input feature map undergoes four repeated blocks for downsampling, consisting of two 3x3 convolution layers (each accomplished with a rectified linear unit (ReLU) activation function), a batch normalization layer and a 2x2 max-pooling layer. Notably, the feature channels double after each downsampling process and then half after each upsampling process in the expanding path. Likewise, the expanding path consists of four comparable blocks for upsampling. The difference is that once a 2x2 up-convolution and batch normalization are conducted on the feature map, a concatenation will be performed with its cropped feature map from the corresponding contracting path, and together they go through two 3x3 convolutions activated by ReLU. At last, a 1x1 convolution layer is used to produce the final classification map.”

Point 2: Incidentally, some points are made that I feel should be even more underlined forcefully. For example, their lake global coverage of is very close to that estimated by Downing et al. (2006) if we only consider the overlapping size classes. It is also very similar to Feng et al. (2015) estimate. However, it is much lower than the impossibly high GLOWABO numbers.

Response 2: Thanks for this comment. According to our statistics on Extended Data Fig. 4, if we consider all lakes contained in the two datasets, the lake estimated by Downing

et al. (2006) is 87.7 times than the GLAKES dataset in number and 32.9% larger in area. It's true that the area and numbers of these two datasets will be much closer (52.4% higher in numbers and 4.5% lower in the area) if we compare only lakes with areas >0.1 km². As a comparison, the lakes documented in HydroLAKES dataset are 5.2% lower in numbers and 5.1% lower in area than GLAKES. Therefore, our results are very close to both the dataset of Downing et al. (2006) and HydroLAKES in the area if only the overlapping size classes are included. Nevertheless, our results show closer proximity to HydroLAKES dataset in numbers and both in area and numbers if all lakes are examined.

In terms of the estimate of Feng et al. (2015), we are sorry that we did not find such an article. Could you please provide the link? We will then make a comparison with their study.

As for the GLOWABO dataset, the reasons why the numbers of lakes are extremely high may partly be because the author did not discriminate the rivers from lakes (reminded by reviewer 3). In such a case, the dataset may contain different types of water bodies instead of only pure lakes, which may result in exceptionally high values of area and numbers compared with other global lake datasets.

Point 3: There is no doubt that small lakes emit per unit area much more than larger lakes and the manuscript makes an important point that the areal expansion of small lakes disproportionately affects the global lake emission numbers. However, I felt there was too much reliance on a single study (Holgerson and Raymond) and would have liked some sensitivity analysis done on that. The authors actually acknowledge that (line 622) but there are many published relationships between GHG emissions and lake size (e.g. Rasilo et al. 2014) and a comparison of results with different such equations would enhance the robustness of their conclusions. Similarly, the occurrence of several negative CO₂ fluxes if a single equation is applied to all lakes is problematic. While there are cases of negative CO₂ fluxes in lakes, they are not particularly related to lake size but rather to eutrophication. Nevertheless, I appreciate that they ultimately used the average flux within binned log classes.

Response 3: Thanks for this valuable comment. We agree that the empirical equations used in Holgerson and Raymond (2016) in applying all lakes may be problematic. Further, we think this may also be applicable to global CH₄ emissions. Therefore, we kept the method for calculating the CH₄ emissions the same as that of calculating CO₂ emissions and updated relevant results (i.e., using the average flux within binned logarithmic size classes). Considering only the emissions from lakes > 0.1km² (which were the common size range for lakes in GALKES and GLOWABO dataset), we can clearly find out that our former version of the global CH₄ estimate is evidently

overestimated compared to that of Holgerson and Raymond (2016), given that the lake area used in our study is smaller than that of Holgerson and Raymond (2016), but we eventually yield a far larger emission value (5.3 vs 2.0 Tg C yr⁻¹), even though the equations used to calculate the CH₄ emissions are from their study. On the contrary, the magnitude of global CO₂ emission and updated global CH₄ emission values are reasonable compared to that of Holgerson and Raymond (2016), the discrepancy between which mainly resulted from the deviations in lake area used for calculations.

In terms of the different methods to calculate global lake carbon emissions, we read many relevant papers. We find that researcher has paid much more attention to the CH₄ emissions than CO₂ and the relevant papers on the former prevail against the latter one, especially in recent years. Nevertheless, among all these articles, some were based on regional scale which are not suitable for global upscaling (Rasilo, Prairie and Del Giorgio 2015, Ran et al. 2021, Borges et al. 2022, Sánchez-Carrillo et al. 2022), some did not provide relevant equations or average flux within binned logarithmic size classes for direct estimation of diffusive emission (Tranvik et al. 2009, Bastviken et al. 2011, Raymond et al. 2013, Rosentreter et al. 2021, Zheng et al. 2022), some offered equations with several key variables hard to obtain for global lakes (Deemer and Holgerson 2021, Zheng et al. 2022). These situations limited our selection for estimating global lake carbon emissions. In particular, DelSontro, Beaulieu and Downing (2018) first introduced binned water productivity classes in the field of carbon emission estimation for global lakes, which are reported to be better than methods utilizing lake size only. However, we tried their approach but couldn't reproduce the same results as documented in their article for unknown reasons. We tried to contact them but didn't receive the exact answers yet.

Besides, we realize that it's inappropriate to directly compare our result with previous estimations that contained 2 additional logarithmic size classes, as mentioned by reviewer 2. Therefore, we changed the overall logic as follows. To begin with, we discovered that our estimation was smaller than Holgerson and Raymond (2016) and discussed the reason (the use of overestimated lake surface area by ref (Holgerson and Raymond 2016), particularly for small lakes). Then, instead of direct comparisons, we compared the original result yielded by Holgerson and Raymond (2016) with previous studies, which indicated a good agreement with several other calculations when different methods or lake surface area datasets were used (Extended Data Fig. 9). Then we declared that since our estimating approach was from Holgerson's study (with different global lake datasets for upscaling), our estimates on current lakes documented in GLAKS dataset should be reasonable, although uncertainty still existed (see limitation and uncertainty section).

Point 4: Reference not cited in their manuscript: 1.Rasilo, T., Prairie, Y. T. & Giorgio, P. A. Large-scale patterns in summer diffusive CH₄ fluxes across boreal lakes, and contribution to diffusive C emissions. *Global Change Biol* 21, 1124 – 1139 (2014).

Response 4: Thanks for this suggestion. We have added this reference to the following sentence in the newly added “uncertainty and limitation” section in Supplementary Note 3: “Besides, incorporation of other relevant drivers such as the water body type, water depth, water productivity, sediments as well as ecoclimate zone would also enhance the representativeness of the average emission rates for more accurate global estimates (Rasilo et al. 2015, Wik et al. 2016, DelSontro et al. 2018, Deemer and Holgerson 2021)”

Response to the reviewer 2:

General comments:

Point: This manuscript presents an interesting spatio-temporal analysis of a new high quality lake dataset, with insights on changes in global lake area over the past 4 decades. The results generally align with previous global scale analyses of both surface water and greenhouse gas emissions, reaffirming the well-established disproportionate roles of small water bodies in surface water hydrology and biogeochemistry. This analysis goes further than similar previous efforts to distinguish patterns with more nuance - natural lakes vs reservoirs, separating patterns within and outside glacier and permafrost regions. Although I don't have the expertise to evaluate the method used to distinguish glacier and permafrost affected lakes, I found this is an interesting aspect of the study. Additionally, datasets on population density and type of drainage basin were used to provide further insights on different decadal patterns. There is room for improvement in terms of clarity and accuracy. Some of the interpretations in comparison to other datasets are not always supported with appropriate evidence – in particular the minimum lake size in existing datasets is smaller than in this new dataset (contradicting some statements), which also seems to affect the comparison of ghg emission estimates. The paper would also be much stronger with a clearer operational definition of lakes to help explain the assumptions and applicability of the classification model, and enable appropriate re-use of the data.

A potential important advancements from this paper is the creation of a new spatially explicit high resolution global lake dataset – although the 2014 GLOWABO dataset from Verpoorter et al is higher resolution, it is not easily findable or readily accessible, and also not multi-temporal. The data availability statement says the data will be provided, but there are not enough details provided to assess whether it will be shared with adherence to FAIR principles (<https://www.go-fair.org/fair-principles/>).

Response: Thanks for your insightful comments. We have taken these comments seriously and made revisions accordingly.

Main suggestions for improvement:

Point 1: A clearer description of how “lakes” are operationally defined would improve this manuscript and provide a helpful foundation for future users of the GLAKES dataset. Although it is established in the introduction that lakes and rivers have distinct hydrology and biogeochemical processes, the criteria for how this distinction is made based on surface water extent/shape/geometry could be specified in more detail. The roles that other features such as wetlands play in surface water extent & variability is somewhat overlooked (eg. ~line 52-54) – is the underlying assumption that all surface water can be classified as lake vs river, or is the distinction lake vs. non-lake? Despite such complications, the analysis presented for upscaling GHG emissions is a good example for why it is valuable to distinguish lentic and lotic waterbodies in surface water datasets - is it possible to elaborate on other appropriate potential future uses of GLAKES given the assumptions used for the model and to classify the training data (~lines 240-244)?

Response 1: Thanks for this valuable comment. We have made a clear definition of “lakes” and “rivers” along with their distinctive features in our revised manuscript. In addition, since our U-Net models were based on binary classification, the classification results would either be lake or non-lake. However, in our definition “lakes” actually contain various types of water bodies, including permanent waters, small ephemeral water bodies, “core portion” of water bodies in floodplains, tidal flat surrounded by lakes, and parts of wetlands as you mentioned, as well as human-transformed water bodies (reservoirs and some large agricultural fields (not much)). Please see the following newly added paragraph for detailed information (lines 448 to 464).

“Here, lakes and rivers mainly indicate lentic and lotic water systems that are visible from space, including both permanent and seasonal waters. Lakes and rivers generally exhibit different features on GSWO images. Compared to lakes that usually have flat and oval outlines, rivers are typically long, meander and narrow in shape, which makes them distinguishable in most cases (see Supplementary Note 3 to explore exceptions). For seasonal water bodies, we tend to address those located around rivers and meanwhile span a large scale (such as floodplains), while keeping the small ephemeral water bodies as lakes. In addition, we also identify tidal flats surrounded by lakes and parts of wetlands as lakes because they are hard to separate from lakes via satellite observations (Tootchi, Jost and Ducharme 2019). Here we use “parts of” because wetlands are usually covered with vegetation during the growing season, and are thus beyond the capture of the GSWO images (or optical remote sensing images). GLAKES

also contains constructed impounded water bodies (i.e., reservoirs) that are closely related to human activities. Notably, some agricultural fields are also included in our dataset, although the proportion may not be large, and further segregation is under process (see Supplementary Note 3). Last, we do not take into account lakes directly connected to the seas as the hydrological conditions, and human interventions are intricate in those regions.”

Thank you for your comment on the other appropriate potential future uses of GLAKES. We think one of the most important applications is to characterize regional-to-global hydrological budgets, as the changes in evaporation and water storage induced by lake size variability were often ignored in past studies. This point has been discussed in the manuscript (lines 246 to 252).

Point 2: The novelty of GLAKES in comparison to previous datasets seems a little overstated. Specifically, compared to GLOWABO (Verpoorter et al 2014), which is described as having a minimum lake size of 0.002 km², with the smallest objects based on filtering anything smaller than 9 pixels. This contradicts statements that GLAKES is the first global lake dataset to include lakes smaller than 0.1 km² without statistical extrapolation (lines 72-73; lines 527-530; lines 545-547).

Response 2: Thanks for your suggestions. We’re aware that the GLOWABO dataset has extracted lakes as small as 0.002 km² in size. However, the GLOWABO dataset is not publicly available, and the total lake area and number documented seem peculiarly high, which raises our concern about their data quality. Nevertheless, our description of “the first global lake dataset to include lakes smaller than 0.1 km² without statistical extrapolation” is inappropriate here. Actually, we are “the first global lake dataset to include lakes smaller than 0.1 km² that is publicly available without statistical extrapolation”. Revisions have been made accordingly on places where similar descriptions occurred.

In lines 73-75 of the revised manuscript, the statement was written as “We used deep learning to identify lakes smaller than the minimum mapping unit for global lake datasets that are publicly available (0.1 km²).”.

In lines 82-85 of the revised manuscript, we changed the clause as “Deep learning makes it possible to detect lakes as small as 0.03 km² (corresponding to approximately 33 Landsat image pixels), which greatly improves the minimum mapping unit and mitigates the issues of mis-accounted small lakes in previous lake datasets that are accessible in public.”.

In lines 527-530 of the original manuscript, the sentence “Notably, in all these previous datasets, the statistics for lakes smaller than 0.1 km² (if any were available) were represented by values extrapolated from larger lakes based on the assumption of power-law distributions for global lake numbers and areas” was deleted.

In lines 708-714 of the revised manuscript, changes have been made to relevant expression as “The overestimation of Downing et al. (2006) is likely due to the reason that the statistics derived for lakes <0.1 km² were not determined from explicit lake mapping but from extrapolated values. In contrast, the overestimation of the GLOWABO dataset (Verpoorter et al. 2014) probably resulted from the inclusion of non-lake polygons such as rivers, given that the disentanglement of lakes from rivers was never mentioned in their documentation. As for Lehner and Döll (2004), similarly, their estimation of small lakes may be constrained by the underlying data sources composing GLWD.”.

Point 3: Statements about comparisons to other lake GHG estimates should be reconsidered (lines 187- 201). Making these comparisons is obviously complicated because of the differences in underlying data on surface water extent datasets and different categorizations used for size classes (as mentioned), however as written there appear to be some misleading/confusing statements.

3a. The abstract concludes GLAKES leads to higher GHG emissions (line 34) however the comparison to Holgerson and Raymond 2016 (line 190) shows smaller estimates than previous, which seems contradictory. Extended data figure 9 also shows emission estimates from the present study are smaller than most other estimates. Is the conclusion about higher emissions from lake area increases about changes over time between the periods analyzed (not differences with previous datasets)? If so this could be stated more clearly.

3b. The lake surface area estimates from Holgerson and Raymond include more than 2 additional logarithmic size classes smaller than the minimum size in GLAKES - 0.0001 to 0.001 km² (from statistical extrapolation), and 0.001-0.01 km² (based on Verpoorter et al 2014 for >0.002 km² and estimated for 0.001-0.002 km²). While it is certainly possible that the surface area extents for these smaller classes are overestimated, they are not included in GLAKES at all and therefore would be an obvious source of the difference (~line 191). The 2 other more recent studies included in Extended Fig 9 (DelSontro et al 2018 and Li et al 2020) both use GLOWABO without extrapolation beyond the lower limit of 0.002 km², however neither appear to be in better agreement (line 194-195) with the GLAKES emission estimates compared to Holgerson and Ramyond 2016.

3c. The CO₂ estimate of 194 Tg C is compared to combined number for both CO₂ and CH₄ in Holgerson and Raymond (should be only 571 Tg from CO₂ not 583 on line 190).

Response 3: Thanks for this valuable comment. As for 3a, what this line wants to express is that the increase in lake area among different time periods results in the increase in lacustrine carbon emission among the corresponding periods. Sorry for this misleading description, we have changed it to “The identified lake area increase over time led to higher lacustrine carbon emissions, mostly attributed to small lakes.”

As for 3b, we realize that it’s inappropriate to directly compare our result with previous estimations that contained 2 additional logarithmic size classes. Therefore, we have changed the overall logic as follows. To begin with, we discovered that our estimation was smaller than Holgerson and Raymond (2016) and discussed the possible reason (the use of overestimated lake surface area by ref (Holgerson and Raymond 2016), particularly for small lakes). Then, instead of directly comparing, we compared the original result yielded by Holgerson and Raymond (2016) with previous studies, which indicated a good agreement with several other calculations when different methods or lake surface area datasets were used (Extended Data Fig. 9). Then we declared that since our estimating approach was from Holgerson’s study (with different global lake datasets for upscaling), our estimates on current lakes documented in GLAKS dataset should be reasonable, although uncertainty still existed (see limitation and uncertainty section).

In terms of 3c, thanks for pointing out this mistake. The CO₂ emissions value has been corrected as 571 Tg C yr⁻¹.

Point 4: The paper would be improved with at least a brief description of the U-Net model to explain why it is applicable for distinguishing lakes from rivers/other surface water features, especially for readers less familiar with such classification models. Is it possible to be more specific about what kind of features make lakes and rivers highly distinguishable in visual examinations (line 401-402), and explain why that might justify using the U-Net model for this kind of classification problem?

Response 4: Thanks for this valuable suggestion. As we mentioned in response 1: “compared to lakes that usually have flat and oval outlines, rivers are typically long, meander and narrow in shape, which makes them distinguishable in most cases.”. In addition, we have added more details of the U-Net model and illustrated why it’s suitable for such kind of classification problem.

“Deep learning has been widely used in many areas (Krizhevsky et al. 2012, Hinton et al. 2012, Sutskever et al. 2014, LeCun et al. 2015), and are proven to be a powerful and creative tool in detecting features of interest from satellite images (Ma et al. 2017, Weiss et al. 2020, Reichstein et al. 2019). A recent inspiring deep learning application in remote sensing image processing was documented by Brandt et al. (Brandt et al. 2020), who detected tree crowns by combining the U-Net model with submeter high-resolution

satellite images. The U-Net model used in Brandt et al. (Brandt et al. 2020) is a typical semantic segmentation technique that performs pixel-wise classification within an image for precise segmentation (Yu et al. 2018, Ronneberger et al. 2015). Compared to the conventional classification tasks, U-Net yields not only the label category of a specific image, but also its corresponding location. Upon the application of U-Net, Brandt et al. (Brandt et al. 2020) managed to map more than 1.8 billion non-forest tree crowns (>3 m²) in the West African Sahara and the Sahel, somewhat overturning the previous stereotype of trees scarcity in these dryland regions. Here, we modified the U-Net model developed by Brandt et al. (Brandt et al. 2020) and transferred its application to global lake mapping. We expect a well-trained U-Net model to perform well when classifying lakes and rivers using GSWO images, as lakes and rivers are already highly distinguishable in visual examinations (Extended Data Fig. 3).

Building upon a fully convolutional neural network (Long et al. 2015), the U-Net model composes of various and hierarchical convolution layers that are widely used in semantic segmentation field for feature detection (LeCun et al. 2015, Ronneberger et al. 2015, Liu 2018), vital for the extraction and segmentation of lakes from rivers. The convolution layer extracts features from an image in the previous layer and results in a less redundant output image called feature map. Generally, the features learned by convolution layers transition from simple to more abstract ones as the level of the convolution layers increase (Zeiler and Fergus 2014, LeCun et al. 2015, Ribeiro et al. 2018), and these features are determined by the convolution kernels (i.e., an array of weights) that are learned automatically through backpropagation. Except for convolution layers, there are various structures that are also essential in the modified U-Net architecture of Brandt et al. (Brandt et al. 2020), including activation function (enabling nonlinear classification), batch normalization (stabilizing and accelerating the training process), pooling (reducing data dimension and computation complexity), up-convolution (restoring the size of feature maps for precise localization) as well as concatenation (combing the higher-level feature map with a lower-level one to better learn representation). In addition, the U-Net model adopts the overlap-tile strategy, which makes it possible to perform a seamless segmentation for extremely large images without losing information about the divided border regions (Ronneberger et al. 2015). This makes U-Net particularly applicable to our goal of pixel-wise lake classification through the GSWO images at a global scale.

Specifically, the U-Net model comprises two major parts: a contracting path for feature interpretation and a near symmetric expanding path for location identification, leading to a u-shaped architecture that enables pixel-to-pixel classification (Ronneberger et al. 2015). In the contracting path, the input feature map undergoes four repeated blocks for downsampling, consisting of two 3x3 convolution layers (each accomplished with a

rectified linear unit (ReLU) activation function), a batch normalization layer and a 2x2 max-pooling layer. Notably, the feature channels double after each downsampling process and then half after each upsampling process in the expanding path. Likewise, the expanding path consists of four comparable blocks for upsampling. The difference is that once a 2x2 up-convolution and batch normalization are conducted on the feature map, a concatenation will be performed with its cropped feature map from the corresponding contracting path, and together they go through two 3x3 convolutions activated by ReLU. At last, a 1x1 convolution layer is used to produce the final classification map.”

Point 5: I think the identification/ description of the 5 categories of sample regions (lines 430-436) is a strength of the analysis – the discussion could be improved by describing how well the model performs in each of those categories. Is it possible to show an example of each of the 5 types of sample region, or at least label the region types shown in Extended data fig 2? What is the size range of the sample regions used to create the training data?

Response 5: Thanks for this valuable suggestion. Examples illustrating the 5 categories of sample regions are presented in the revision (also see below). In addition, we re-calculated the error matrix listed in Table S4 for each region type individually to assess the model’s performance, the table of which is also presented in the revision (also see below). First of all, it should be noted that the region type was only the representativeness of the major hydrological features of the lakes within the region bound, where lakes with distinctive features may also co-exist. In practice, not all five categories of sample regions were included in the Normal Model and Floodplain Model (see Table S4). Specifically, the Normal model consisted of type 1 (“lakes with middle/high occurrence”, HO), 2 (“lakes with low occurrence”, LO), 3 (“large lakes”, LL) and 4 (“lakes alongside river”, AR), since the type 5 (“lakes within floodplains”, WF) was not the target of the Normal Model. On the other hand, the Floodplain Model was made up of types 1, 3, 4 and 5. The reason why types 1, 3 and 4 were included for model interpretation was that the main patterns described by types 1, 3 and 4 were also observable within the regions defined by type 5, while type 2 was excluded because of the relatively high occurrence threshold (i.e., 30%) applied for the Floodplain Model.

Overall, the accuracy of the former three categories (HO, LO, and LL) was higher than that of the remaining two categories (AR and WF), probably owing to the relatively intricate hydrological conditions of the last two types of regions. Here the largest omission errors (19.6%) of the Normal Model originated from AR, which probably resulted from the missed detection of oxbow lakes that were hardly distinguishable from rivers. There was additional and comparative omission lying in the Floodplain Model,

i.e., WF (12.8%), where the occurrence patterns were extremely complicated, and the exact floodplain extents were hard to depict. It is noteworthy that the deviation of the scatter points representing the region type AR was not evident for either model in Extended Data Fig. 2b, although exhibiting large omission errors. This is because their containing lake area was generally small and thus, their scatter points were hidden in the lower-left region of the scatter plot with dense scatter concentration. Similarly, commission errors were relatively low in almost all region types, especially compared to omission errors, confirming the model's conservative strategy in delineation for all region types analyzed.

In addition, the size distribution of the sample regions is appended in the Extended Data Fig. 2a (see below). Overall, the logarithmic sizes of these regions were approximated to a normal distribution, both in terms of the Normal model and Floodplain Model, while the total number of the former was ~1.7 times of the latter. In addition, the size of most sample regions was at the order of magnitude of $10^2 \sim 10^3 \text{ km}^2$, with a median area of $5.91 \times 10^2 \text{ km}^2$ for the Normal Model and $5.69 \times 10^2 \text{ km}^2$ for the Floodplain Model.

Extended Data Fig. 3 | Image pairs revealing the performance of the deep-learning algorithm in predicting lake extents using the test set. The right panels show the predicted lake extents; the correct classification (Label + Prediction), omission errors (Label only), and commission errors (Prediction only) are color-coded. The left panels are the input images sourced from the GSWO dataset and are independent of the labels used for algorithm training and validation. The lower right annotations represent the abbreviations for the five region types: “lakes with middle/high occurrence” (HO), “lakes with low occurrence” (LO), “large lakes” (LL), “lakes alongside rivers” (AR) and “lakes within floodplains” (WF). For specific accuracy statistics, please refer to Extended Data Fig. 2 and Extended Data Table 1.

Extended Data Fig. 2 | Development and validation of the deep-learning algorithm for predicting lake extents. (a) Spatial distribution of the sample regions selected for training, validation, and testing, along with size range of the sample regions.

Table S4 | Accuracy assessments of our developed deep-learning algorithm for different region types.

Error matrix for the GLAKES dataset estimated using independent test labels for the Normal Model and Floodplain Model, listing the accuracy levels derived for different region types. Note that only pixel-based results are presented here as the polygon-based results are largely biased by the prevalence of small lake polygons in almost all region types, and thus cannot reflect the true deviations among different region types.

	Type Index	Type Name	Omission (%)	Commission (%)
Normal	1	High Occurrence (HO)	12.4	0.3
	2	Low Occurrence (LO)	8.5	1.6
	3	Large Lakes (LL)	3.3	0.2
	4	Alongside Rivers (AR)	19.6	5.0
	-	All	5.4	0.5
Floodplain	1	High Occurrence (HO)	8.0	1.9
	3	Large Lakes (LL)	4.2	0.3
	4	Alongside Rivers (AR)	12.6	1.9
	5	Within Floodplains (WF)	12.8	2.7
	-	All	9.6	1.9

Point 6: Data availability statement does not provide enough detail to evaluate how dataset will be archived, made available or how to access. Should be revised with adherence to FAIR

principles (<https://www.go-fair.org/fair-principles/>), e.g. publishing the dataset in a repository that issues a unique identifier.

Response 6: Thanks for this valuable suggestion. We have read the FAIR principles carefully and revised our dataset accordingly to make it more adherent to the FAIR principles. The relevant data can be temporally accessed through the following link https://drive.google.com/drive/folders/12wQKJ_ME_E_lJHT3mt2WC5vEcMNM-hpII?usp=sharing. The final dataset will be uploaded to the zenodo platform (<https://zenodo.org/>) upon publication of this manuscript.

Point 7: It would help the reader if the time scale of variability (decadal?) was stated more prominently (e.g. line 1480149, caption for Fig 4), especially since seasonal and interannual fluctuations are mentioned in the introduction. Consider rephrasing “dynamics” to “changes” or “trends” in surface water extent, given the well documented but complicated relationships between GHG emissions and changes in water level possible on much shorter timescales (eg <https://doi.org/10.5194/bg-9-2459-2012>, <https://doi.org/10.1672/07-98.1>, <https://doi.org/10.1002/2015JG003283>, <https://doi.org/10.3390/atmos10050269>). The manuscript would also be stronger if the discussion addressed how the results may be affected by changes/dynamics at those shorter timescales, for example frozen lakes becoming only seasonally ice-covered.

Response 7:

Thanks for this valuable suggestion. First of all, the time scale of variability here indicates decadal or long-term variability. We have specified the exact time scale of temporal changes wherever necessary as well. In addition, “dynamics” have been rephrased to “changes” or “trends” when it relates to carbon emissions and mainly indicates long-term interannual variations. As for the impacts of changes at shorter timescales on our current results, we added two paragraphs below for discussion (one for long-term lake area changes and another for decadal carbon emission changes).

“In temporal change analysis, owing to the constraint of the valid observations of Landsat images, this study mainly focused on the changes of lake extent at decadal scale. However, lake dynamics at shorter timescales could also be evident. Pickens *et al.* discovered that only 23% of the total area of open surface water was permanent without ice cover within 2019, while permanent water covered by seasonal ice/snow constituted 41% of the total area, and the remaining 36% was made up of seasonal waters regardless of ice coverage (Pickens et al. 2022). Furthermore, such seasonal patterns of water/land/ice transition may witness substantial changes during the whole study period owing to the impact of climate change, including intensifying reductions in ice cover duration and varied changes in wetting/drying trends (Magnuson et al. 2000, Sharma

et al. 2019, Greve et al. 2014, Roderick et al. 2014, Woolway et al. 2020). These changes at a shorter timescale may impose divergent impacts on our decadal change analysis of lake surface area. The impacts of seasonal water/land transition along with its trend were minimal as they have already been incorporated into the occurrence map and thus the calculation of probability-weighted lake area each period. However, the negligence of lake ice coverage (in the GSW MWH dataset, ice was flagged as invalid observations) might lead to a conservative extraction of the lake outlines as well as an underestimation of the water occurrence value. Given that the extension of the ice-free season was reported to exhibit an increasing trend (Woolway et al. 2020, Wang et al. 2022), the underestimation of probability-weighted lake area might be less severe in the more recent period, indicating that there might be a slight overestimation of the calculated lake area changes over different periods in places covered with ice.

In addition, in calculating the changes over three periods, we kept the average flux values constant, considering only the long-term carbon emission changes that were brought by the lake area variations over different time periods. Nevertheless, the transfer of carbon gases from the aquatic environment to the atmosphere is a highly dynamic process, which could also be modulated by lake dynamics (such as ice phenology and water/land transitions) at shorter timescales (Chamberlain et al. 2016, Deemer et al. 2016, Holgerson and Raymond 2016, Wik et al. 2016, DelSontro et al. 2018, Keller et al. 2021). It has been reported that CO₂ and CH₄ accumulate under the ice, and subsequently vent a substantial amount to the atmosphere during the spring melt, during which CH₄ oxidation may co-occur, although this is probably not applicable to oligotrophic lakes or completely frozen lakes (Bastviken et al. 2004, Kortelainen et al. 2006, Michmerhuizen, Striegl and McDonald 1996, Utsumi et al. 1998). Considering the trend of global warming over the study period, the ice-free seasons for most lakes extended and some permanently frozen lakes became seasonally ice-covered, leading to a further boost in global carbon emissions (Natchimuthu, Panneer Selvam and Bastviken 2014, Sharma et al. 2019, Wik et al. 2016). Besides, the seasonal drying and wetting of lakes was also an important carbon emission source. Studies have revealed complex relationships between water level and aquatic carbon emissions and identified dry aquatic sediments as significant carbon gas hot pots (Chamberlain et al. 2016, Keller et al. 2021, Marcé et al. 2019, Tangen and Bansal 2019), which we also did not account for. Given the varied wetting/drying trends across different regions globally (Greve et al. 2014, Roderick et al. 2014, Woolway et al. 2020), the overall impact on long-term carbon emission changes is hard to quantify, and more data are required for systematic evaluations.”

Other/minor suggestions:

Point 1: Line 30 – maybe specify “all six continents analysed” since Antarctica is excluded.

Response 1: Thanks for this comment. The statement was corrected as “From the beginning period (1984-1999) to the end (2010-2019), lake area increased across all six continents analyzed, with a net change of +46,278 km², and 56% of the expansion was attributed to reservoirs.”.

Point 2: Line 64 – should “water productivity” be “primary productivity” ?

Response 2: Yes, the “water productivity” should be “primary productivity”.

Point 3: Line 82 - How was 0.03 km² threshold determined for minimum size? Are lakes smaller than 0.03 km² not present in GSW, not captured by the model, or excluded for other reasons? It seems somewhat overstated to say improved resolution “solves” issues associated with mis-accounted small lakes in previous datasets since lakes/surface water can be mis-classified for reasons other than spatial resolution (e.g. optical complexity, forest canopy, mixed pixels, shadows).

Response 3: Thanks for this comment. First of all, lakes smaller than 0.03 km² are observable in GSW owing to the high spatial resolution (30m). As for whether those lakes are able to be extracted by the model, it largely depends on the size threshold of the samples used to train the models. In practice, we first predefined the cutting threshold of samples as 0.03 km² according to our visual observations. The main objectives were to exclude small polygon residuals generated during this sample extraction procedure as well as to screen out small isolated agricultural fields. Then the lake samples with sizes above this threshold were trained by the models to extract their underlying features and yield predictions. The results would be impacted by the value of the cutting threshold during this process, as we observed a vast amount of tiny lakes with size <0.03 km² unable to be detected by models. Of course, there were indeed some lakes below the threshold that were extracted by our models, which were then further screened out during our post-processing procedure so that our remaining lake polygons were all >0.03 km². Therefore, actually, all lake samples were set to be >0.03 km².

Nevertheless, it's acknowledged that the setting of a pre-defined fixed cutting threshold for lake samples was probably the major reason for the relatively high omission errors (23.5% for Normal Model and 21.2% for Floodplain Model) for small lakes (especially those around 0.03km²). This could be induced by the influence of the negative samples. Specifically, in the process of sample preparation, we applied a filter to screen out all lakes with a size <0.03km². In fact, the difference between lakes just exceeding the size

threshold and those approximating the thresholds (e.g., 0.031km² and 0.029 km²) was probably minor. Therefore, the setting of a fixed cutting threshold for samples may somewhat “confuse” the model, in a way that some lakes were interpreted as true lakes (because they were marked by our labels) while the others with similar features may be identified by the model as non-lakes (due to the lack of overlaying labels), thus leading to a certain extent of missed detections of small lakes around the size of the cutting threshold (0.03 km²).

We further examined the error matrix of small lakes with a finer division of size range (see below). Here the reason why we could obtain results with a size <0.03 km² (below our cutting threshold of samples) was because the U-Net model learned features at the patch level (512*512 pixels), where some lakes with area >0.03 km² across multiple patches would be split into smaller pieces that might below 0.03 km² and interpreted by our models. As seen in the table, the omission and commission errors generally decreased as the lake size increased. The errors were extremely high for lakes with a size <0.03 km² in both models, with omission errors of >50% and commission errors of >15%. The accuracy was much higher when considering lakes with a size >0.1 km² (the lower size limit for most global lake datasets), where the corresponding omission and commission values dropped to below 20% and 5%, respectively. Nevertheless, although lakes with size ranges of 0.03-0.05 km² still faced high omission issues, the commission errors declined suddenly from >15% to ~5%. In practice, we treasured commission errors more than omission errors so as to ensure that the detected portion of our GLAKES polygons was generally “true” and thus could be placed with more confidence in further analysis. Therefore, we kept the size threshold as 0.03km² to include more lakes in our dataset without much compromise on misclassification.

Table S3 | A further investigation of the accuracy of our developed deep-learning algorithm in subdivided small lake groups.

Error matrix for the GLAKES dataset estimated using independent test labels for the Normal Model and Floodplain Model, listing the accuracy levels derived for subdivided size groups of small lakes. Both pixel-based and polygon-based results are included for assessments. Likewise, the polygon-based omission and commission values presented below represent the average of the corresponding values of all lake polygons being assessed.

Size	Pixel-based		Polygon-based		
	Omission (%)	Commission (%)	Omission (%)	Commission (%)	
Normal	<0.01 km ²	85.3	76.6	91.4	88.1
	0.01-0.03 km ²	52.6	17.4	54.9	22.3
	0.03-0.05 km ²	50.5	3.6	51.5	3.7
	0.05-0.1 km ²	34.5	3.1	35.0	3.0
	0.1-1 km ²	18.9	2.3	22.2	2.5
Floodplain	<0.01 km ²	81.4	73.0	88.6	86.1
	0.01-0.03 km ²	51.4	21.1	53.7	25.2
	0.03-0.05 km ²	51.2	6.7	52.8	6.8
	0.05-0.1 km ²	31.3	6.4	32.0	6.4
	0.1-1 km ²	17.0	4.7	19.5	4.8

Of course, we acknowledged that a better solution might be the inclusion of samples with size <0.03 km² in model training, followed by a result-oriented determination of cutting threshold for lake predictions (i.e., finding out a threshold where the accuracy for lakes below the certain size threshold was unacceptable, if possible). Nevertheless, despite being troubled by the relatively high omission errors for small lakes, our GLAKES dataset showed marked improvements over previous global lake datasets, considering its advantages in global coverage (60°S-80°N), high spatial resolution (~30m), long-term changes (four decades), spatiotemporal consistency (uniform mapping of global lakes instead of aggregation from different lake datasets), overall accuracy (overall accuracy >98.7% and MIoUs >88.7%), and the delineation of small lakes (lower limit as 0.03 km²). Besides, the lake change analysis was performed only within the boundaries defined by our GLAKES dataset (see below); therefore, the associated impacts of the classification errors (particularly the omission errors) should be limited. Of course, there was still room for improvement in our GLAKES dataset. An example was the above-mentioned inclusion of lakes <0.03km² in model training, along with a statistics-based decision of the cutting threshold for predicted lake polygons, which may be considered in our next version of GLAKES in the future.

In addition, we changed the statement “solves the issues of mis-accounted small lakes in previous lake datasets” to “mitigates the issues of mis-accounted small lakes in previous lake datasets that are accessible in public”.

Point 4: Line 88 – Isn’t the GSWO dataset limited to 80 deg N? Consider specifying max northern extent, and/or include latitude range where dataset is described in methods.

Response 4: Thanks for this suggestion, we have added the latitude range to the corresponding methodology section. The description of the GSWO dataset is now

presented as “The GSWO dataset provides global (from 60°S to 80°N) documentation of the location and frequency of water occurrences over nearly four decades (1984-2019) and was generated using 30-m-resolution Landsat images (Pekel et al. 2016).”.

Point 5: It would help the reader to explain why P1 is two decades and P2 and P3 are each one decade. In Figure 2, it is somewhat confusing what dates/date ranges are being compared in top and bottom panels - It may be more clear to give beginning and end year of each range.

Response 5: Thanks for this suggestion. Actually, the exact time period of P1 is 1984-1999, which is altogether 16 years. The reason why P1 covers 16 years while the remaining periods span only 10 years is mainly due to the number of valid observations from Landsat imagery. As we can see, the coverage of Landsat-5 is limited at its first stage and gradually increases over time (Pekel et al. 2016). For example, the Landsat images of America and western Europe are available as early as 1984-1985, while the first year of imaging would be 1987-1991 for many places in Asia, let alone places like Siberia and New Zealand where the first observation year is later than 1995 (Pekel et al. 2016, Wulder et al. 2016). Since the launch of Landsat-7 in 1999, the number of valid observations has almost reached twice with extended global coverage (Pekel et al. 2016, Wulder et al. 2016). Therefore, in our study, the first time period is set as 16 years (1984-1999) instead of 10 to include more regions where data are not available at the first stage. Then P2 (2000-2009) and P3 (2010-2019) cover 10 years respectively, the length of which is enough for near-global cross period comparisons (Extended Data Fig. 6).

To make the period range more clear, we have specified the beginning and end year at its first occurrence, which is presented below: “We examined global lake dynamics across three periods (1980-90s: 1984-1999, 2000s: 2000-2009, and 2010s: 2010-2019) (Fig. 2) by comparing the water probability-weighted area within lake boundaries as defined by our GLAKES dataset (see Methods).”.

Point 6: Line 143 & elsewhere – clarify if “global inland regions” are used interchangeably with the 1 deg grid cells.

Response 6: Yes, “global inland regions” and “1°×1° grid cells” are interchangeable. Actually, they both indicated grid cells that spatially intersected with our GLAKES polygons, as we mentioned, “we examined global lake changes across three periods by comparing the area within lake boundaries as defined by our GLAKES dataset)”. We have clarified this in the revised manuscript.

Point 7: Line 144 – maybe specifying “net” area increase if that is what is meant here

Response 7: Thanks for this suggestion, we have corrected it as indicated.

Point 8: Line 147 – consider modifying sub-heading to specify what patterns small lakes have an outsized role in.

Response 8: Thanks for this suggestion. We have changed it to “The outsized role of small lakes in global lake size variability.”

Point 9: Line 163 – unclear whether the “small lake-dominated regions” referenced here is about areas with just many small lakes or areas where small lakes dominate the variability. (As written I interpret as the former, but the referenced figure and rest of the paragraph seem to be about the latter.)

Response 9: Sorry for the confusion, we have merged this sentence with the previous one to make it clearer (lines 162 to 167): “The changes in small lakes showed dominant contributions (>50%) in approximately half of the examined inland regions (49.9% of the grid cells from 1980-90s to 2000s, and 50.1% from 2000s to 2010s) (Fig. 3c); such small lake-dominated regions were spread across the entire globe in both low-populated regions and areas with high chances of human disturbance (Extended Data Fig. 8).”

Point10: Line 166 – unclear what ‘dynamic amplitude’ means here

Response 10: The ‘dynamic amplitude’ meant the magnitude or range of the relative area changes for lakes in each population density group, as demonstrated in (Extended Data Fig. 8). To avoid confusion, the statement has been rephrased as: “Furthermore, decadal lake area variations generally increased with regional population density, and the variation range was much higher for small lakes than for medium and large lakes, indicating the potential role of human activities in shaping small lakes.”

Point 11: Extended data Figure 8 – consider adding a reference line at $y = 0$

Response 11: Thanks for this suggestion, the reference line is added as suggested.

Point 12: Were the normal and floodplain models both applied globally, or was there a criteria/dataset used for distinguishing normal and floodplain areas?

Response 12: Thanks for this comment. It should be noted that lakes from Normal Model and Floodplain Model were combined to generate our final version of global lake polygons (the “GLAKES” dataset). In practice, since both the Normal Model and

Floodplain Model yielded lake predictions at global coverage, we applied pre-defined river buffer zones to determine whether the extracted lake polygons from the Normal Model or Floodplain Model should be used for our final GLAKES dataset. Specifically, for buffer zones flagged as “floodplain”, lake polygons (within these buffer zones) extracted from the Floodplain Model were selected as a part of GLAKES dataset, while the corresponding outputs from the Normal Model were discarded. On the contrary, for all remaining areas (including buffer zones flagged as “normal” and areas outside the river buffer zones), lake polygons from the Normal Model were included in our final dataset. The basic principle for the determination of the exact flag (“normal” or “floodplain”) for each buffer zone is to measure the extent of seasonally flooded non-lake and non-river waters within the buffer zone. To begin with, a 1km buffer was applied to each vectorized polygon of global rivers documented in the GRWL Mask V01.01 product (<https://zenodo.org/record/1297434#.YrvEzj5ByUk>). Within each buffer, the area of seasonally flooded non-lake and non-river waters was calculated by summing the area of all GSWO pixels with occurrence <75% (to exclude the permanent and near-permanent water), except for those already being defined as rivers (by GRWL mask) and lakes (by rasterized HydroLAKES polygons). Finally, buffer zones where the ratio of their containing flooded area to the corresponding buffer area exceeded the flooding threshold were flagged as “floodplain”. In this study, the flooding threshold was set as 0.1 through trial and error. We have added the above explanation in our revised manuscript to inform readers how the two models were combined together to obtain our GLAKES dataset (see lines 601 to 623).

Point 13: Extended data figure 9a – error bar on Marotta et al 2008 appears misaligned.

Response 13: Thanks for this comment. According to Marotta et al. (2008), the carbon emissions of global lakes were estimated using two different piston velocity, which yielded 440 Tg C yr⁻¹ and 860 Tg C yr⁻¹, respectively. The former estimation was regarded as the final result since it was put in the abstract, while the less conservative one only appeared in the final discussion. Therefore, we deemed the value 440 Tg C yr⁻¹ as the final estimation, while the less conservative estimation served as the upper bound, which was the reason why the error bar appeared like this.

Point 14: Typo in caption for Fig 4? Line 189 says the emissions estimates were calculated following methods in ref 23 (Holgerson and Raymond 2016) but caption for Fig 4 says method is from ref 22 (Oertli et al 2002, about species area relationships)

Response 14: Sorry for this type of mistake. The reference should be Holgerson and Raymond (2016).

Point 15: Line 233-235 should also discuss or at least reference DelSontro et al 2018 (ref 24) which addresses eutrophication/productivity effects on emissions.

Response 15: Thanks for this suggestion. We have added this important reference to the revised statement “Besides, incorporation of other relevant drivers such as the water body type, water depth, water productivity, sediments as well as ecoclimate zone would also enhance the representativeness of the average emission rates for more accurate global estimates (Rasilo et al. 2015, Wik et al. 2016, DelSontro et al. 2018, Deemer and Holgerson 2021).”

Point 16: Line 383 – is there evidence that these lake area changes are actually caused by human interventions without other natural variability? consider rephrasing “impacts of human interventions” to something like relationship with or co-occurrence with human populations.

Response 16: Thanks for this valuable comment. It may be assertive to declare that these lake area changes are solely caused by human interventions based on our current analysis. Natural factors may also take profound roles in modulating such variations, which requires further analysis. Based on your suggestion, this sentence has been changed to “We used the Gridded Population of the World (GPW) dataset to investigate the relationship between human population and lake area changes.”

Point 17: Line 444 – how was “coverage of all possible hydrological conditions” determined?

Response 17: Thanks for this comment. Actually, the assertion “all possible hydrological conditions” is not accurate. As we mentioned, we defined 5 types of samples that presented different features, i.e., “1) relatively static lakes that exhibited high/moderate water occurrence (HO); 2) highly dynamic lakes with relatively low water occurrence (LO); 3) lakes spanning large spatial scales that were challenging to interpret using models due to the relatively large sizes of the lake objects relative to the sizes of the modeled patches (LL); 4) lakes located alongside rivers that required more attention to be distinguished from rivers (AR); and 5) lakes within floodplains that often combined to form lake clusters (WF).” Based on visual explorations, we think those scenarios generally represent the features of the most common lakes, although it should be admitted that the hydrological conditions are very complicated in some regions, and thus the “coverage of all possible hydrological conditions” is impracticable. Therefore, the words have been changed to “the coverage of all typical hydrological conditions”.

Point 18: Line 465 – should “flood” be floodplain?

Response 18: Yes, “flood” should be “floodplain”.

Point 19: Lines 478-484 – can this part be explained more clearly? (or examples shown in extended figures?)

Response 19: Thanks for this suggestion. We have added examples in Extended Data Fig. 11 to better illustrate the procedure regarding the removal of river residuals (see below). Figure a and c represent situations where the target GLAKES polygons (those overlaid by river mask datasets) are covered by the HydroLAKES polygons while those in figure b and d are not. In addition, the area ratio for target polygons is ≥ 0.8 in figures a and b, while the corresponding values are < 0.8 in figures c and d.

Extended Data Fig. 11 | Post-processing of river residual removal and the corresponding results. (a) Target GLAKES polygons covered by HydroLAKES polygons with an area ratio ≥ 0.8 ; (b) Target GLAKES polygons not covered by HydroLAKES polygons with an area ratio ≥ 0.8 ; (c) Target GLAKES polygons covered by HydroLAKES polygons with an area ratio < 0.8 ; (d) Target GLAKES polygons not covered by HydroLAKES polygons with an area ratio < 0.8 . For (a)-(d), the left figures represent the GLAKES polygons before residuals removal and how they spatially overlay with river masks (GEWL/OSMWL), while the right figures indicate the results after the residual removal post-processing procedure by utilizing the area ratio before/after river masks and the spatial relationship with HydroLAKES.

Point 20: Line 539 – found the wording here confusing, whether “missing” lakes in eastern Canada and Scandinavia were commission or omission errors in GLAKES.

Response 20: Thanks for this comment. The underestimation or overestimation of lakes could either originate from the inherent constraint of both GLAKES and HydroLAKES datasets. Now the relevant sentences have been rewritten as: “Moreover, we found a substantial number of missing lakes in eastern Canada and Scandinavia in our dataset compared to HydroLAKES, as well as lake overestimations with varying degrees in other regions, such as Siberia and major river floodplains (Extended Data Figs. 4b & 5). These discrepancies could be raised for many reasons, either to be responsible for the inherent limitation of the GLAKES or otherwise HydroLAKES. One example is the inability of GLAKES (or GSWO) to capture lakes that are seasonally ice-covered throughout a year and heavily vegetated in the remaining month, which is typical for some small and shallow lakes in places like Canada Shield. The large values of GLAKES could also be partially explained by the inclusion of some agricultural fields (used to be lakes) or accidentally large floodplains. On the contrary, for HydroLAKES, the constraint of its composing dataset (e.g., MODIS MOD44W water mask and SRTM Water Body Data) in detecting small lakes may be the possible reason for the lake underestimation in some regions. Overall, both GLAKES and HydroLAKES have their own strengths and limitations in terms of lake coverage, but what distinguishes GLAKES is its global consistency (not mosaic from different datasets), higher resolution (better characterizes water/land interface), the reflection of multidecadal lake extent (not snapshot on short time period) as well as the inclusion of smaller lakes (<0.1 km²).”

Point 21: Line 603 – what is the source of the air temperature data?

Response 21: The air temperature data for each lake was retrieved on the basis of the ERA5-Land monthly averaged air temperature at 2m. Nevertheless, as reviewer 1 suggested, we have changed the approach for calculating CH₄ emissions. Now the method for calculating CH₄ emissions is the same as that for CO₂ emissions. Therefore, the air temperature data is no longer needed anymore. Detailed illustration is presented below.

In our previous submission, the CH₄ emissions were not computed by directly multiplying the size-dependent mean flux estimates from Holgerson and Raymond (2016) with the total lake area of each logarithmic size class. We actually utilized multiple equations from Holgerson and Raymond (2016) to estimate flux and carbon emission for each lake, and sum them all to obtain the global lake emissions. This process, however, we thought might have some issues. As reviewer 1 said, the occurrence of negative carbon fluxes might reflect that the empirical equations used in Holgerson and

Raymond (2016) in applying all lakes were problematic. This could both result from the fact that these lakes might be more closely related to water productivity (or other factors) instead of lake size, or the empirical equations in calculating carbon emissions had large uncertainties for lakes outside the geographic range (i.e., 30°N-70°N) of the in-situ samples used to construct the empirical equations.

Reviewer 1 recommended the use of the average flux within binned logarithmic size classes. Although this comment was mainly for CO₂ emissions, we thought it was also applicable to CH₄ emissions, though uncertainty still existed (see Supplementary Note 3). Overall, we kept the method for calculating the CH₄ emissions the same as that of calculating CO₂ emissions and updated relevant results. Considering only the emissions from lakes > 0.1km² (which were the common size range for lakes in GALKES and GLOWABO dataset), we could clearly find out that our former version of the global CH₄ estimate was evidently overestimated compared to that of Holgerson and Raymond (2016), given that the lake area used in our study was smaller than that of Holgerson and Raymond (2016), but we eventually yielded a far larger emission value (5.3 vs 2.0 Tg C yr⁻¹), even though the equations used to calculate the CH₄ emissions were from their study. In contrast, the magnitude of global CO₂ emission and updated global CH₄ emission values were reasonable compared to that of Holgerson and Raymond (2016), the discrepancy between which mainly resulted from the deviations in the lake area used for calculations.

Point 22: Line 616-618 – consider including newer references for work on reservoir ghg emissions eg. <https://doi.org/10.1093/biosci/biw117>, <https://doi.org/10.1029/2019JG005600>.

Response 22: Thanks for this suggestion. As for the estimation of carbon emission section, after a review of Holgerson and Raymond (2016), we discovered that they only excluded the in-situ samples of CH₄/CO₂ fluxes of reservoirs but combined the natural lake area with reservoir area in global upscale (i.e., they used GLOWABO dataset which did not distinguish between natural lakes and reservoirs). This indicated that they actually regarded reservoirs as natural lakes in terms of carbon emission estimation, like several other studies (Raymond et al. 2013, DelSontro et al. 2018, Li et al. 2020). Therefore, we decided to add the reservoir area into the total lake area and update our new results, while we informed readers that the waterbody type (natural lakes vs reservoirs) matters in carbon emission estimates for global lakes in Supplementary Note 3: “Nevertheless, previous studies had revealed that methane emissions in different water body type might be driven by different processes, which would inevitably impact the final estimation results (Deemer et al. 2016, Hayes et al. 2017, Deemer and Holgerson 2021)”. After incorporating the reservoir area into the total lake area, the contribution

of small lakes to total emission decreased, given the relatively large size of reservoirs. Nevertheless, the roles of small lakes in modulating lacustrine carbon emissions were still evident, considering the small proportion of lake surface area. Please refer to the section “New estimates of lacustrine carbon emissions” for more detailed updated results (lines 171 to 217).

As for the listed two references, they were very informative and have been added to the above paragraph.

Response to the reviewer 3:

General Comments:

Summary

This study introduces a new dataset named GLAKES, which depicts the surface extent dynamics of 3.4 million lakes at the global scale from the 1980s to present day. The authors delineated individual lake polygons by adapting a deep-learning model to conduct a supervised classification of existing grids of long-term (1984-2019) surface water occurrence. The deep learning model outputs were enhanced by removing delineated water bodies that substantially overlap with river channels and seawater. With this new dataset of lake polygons, changes in global lake density and area were then computed between three time periods (1980-90s, 2000s, and 2010s). The authors estimate that lakes expanded across all continents over the past four decades, mostly due to artificial reservoir building, and that small lakes account for most of the variability in global lake area. Leveraging these estimates, this study also determines that global carbon emissions from natural lakes increased over the same time periods, most of which attributed to small lakes.

Outstanding features

I consider the following to be the outstanding features of this research work:

- Leverages state-of-the-art data sources and models to create a near-comprehensive dataset of individual lakes, delineating lakes down to 0.03 km² in surface area (compared to 0.1 km² for the most widely used lake polygon dataset at present, HydroLAKES).
- Provides the first spatially-explicit estimate, to the reviewer’s knowledge, of global longterm lake extent dynamics.
- Updates estimates of global carbon emissions from natural lakes, predicting a net increase in lacustrine emissions over time.
- Confirms the outsized role of small lakes in global lacustrine carbon emissions.

Key points and results

I consider the following to be the key points and results of this research work:

- Identifies 3.4 million individual lakes, totalling 3.2 x 10⁶ km² (2.2% of the global land area).
- While displaying high overall accuracy, the lake delineation model significantly underestimates the extent of small lakes (omission rates of 19.3% and 23.7% for lakes inside and outside floodplains, respectively) and overestimates the extent of large lakes (commission

rates of by 22.3% and 11.6%, respectively).

- The great majority of lakes are small (< 1 km²), but most of the global lake area is attributed to large lakes (> 100 km²).
- The estimates of lake number and area resulting from this study align closely with HydroLAKES (Messenger et al. 2016), another free-to-access dataset, confirming in the reviewer's opinion that two other existing studies (Verpoorter et al. 2014 and Downing et al. 2010) yielded substantial overestimates.
- Global lake area increased from the 1980s-90s to the 2000s and from the 2000s to 2010s, most of this expansion stemming from the expansion of glacier- and permafrost-fed lakes as well as by artificial reservoir building.
- Small lakes (< 1 km²) showed the highest temporal variability in extent compared to larger lakes.
- Estimates that natural lakes emit 194 Tg C yr⁻¹ of CO₂ and 7.2 Tg C yr⁻¹ of CH₄. These new figures are smaller than previous estimates because most previous models relied on Verpoorter et al.'s surface water extent dataset, which overestimated global lake surface area.
- Due to the increase in global lake area, carbon emissions from natural lakes increased by 3.02 Tg C yr⁻¹ for CO₂ and +0.31 Tg C yr⁻¹ for CH₄, most of these increases stemming from small lakes.

Originality and significance

Will the work be of significance to the field and related fields? How does it compare to the established literature? If the work is not original, please provide relevant references.

This study is a significant and valuable contribution to the literature, and the conclusions are original. It will likely become a baseline dataset for many subsequent studies, thus enabling the advancement of our understanding of the role of lake in regional and global hydrological and biogeochemical cycles, as well as the impact of human activities of lake ecosystems and the services they provide globally. I congratulate the authors for their substantial effort and resulting contribution.

This work represents a substantial step forward compared to the established literature, foremost because it quantifies the spatio-temporal dynamics of lake surfaces over the past four decades. Whereas other studies have quantified inland water dynamics as a whole (e.g., Pekel et al. 2016, which was used by this study, the coarser dataset by Klein et al. 2017, Pickens et al. 2020, and most recently Pickens et al. 2022), what sets this study apart is that it aims to focus exclusively on the dynamics of lakes as individual entities (delineating the shoreline of individual lakes rather than continuous grids of surface water cover). This matters because lacustrine ecosystems differ fundamentally from other inland waters like rivers, seasonally inundated floodplains and other types of wetlands — in their hydrology, biogeochemistry, biodiversity, and their contribution to people and society. In terms of a static dataset of lake polygons, this study is also an advancement compared to the two most established studies/datasets on the topic: HydroLAKES and GLOWABO (I do not include the GLWD here as it can fairly be considered as a product from a previous generation).

It is an advancement over HydroLAKES for three main reasons:

1. It has a higher resolution and thus provides a more comprehensive accounting of small lakes.
2. The lake polygon dataset provides a more temporally integrated view of lake extent

globally because it is derived from Earth observation data over decade. By contrast, the bulk of HydroLAKES polygons, geographically (< 60N), were delineated from a short satellite mission (SRTM) over 11 days in February 2000.

3. It is likely more spatially consistent than HydroLAKES. HydroLAKES results from the compilation and harmonization from over five original data sources at different temporal and spatial resolutions while all polygons in GLAKES were delineated with the same processes using a common data source (despite differences among Landsat sensors over time).

It is also an advancement over GLOWABO, despite GLOWABO including lakes down to 0.002 km² (9 x 30-m pixels), for three main reasons:

- GLOWABO was never publicly released. Therefore, its application was limited and it underwent little external validation.
- GLOWABO polygons were also extracted from Landsat imagery but only from the year 2000 ± 3 year, thus representing a snapshot in time rather than a long-term picture of lake extent.
- The summary statistics provided by GLOWABO suggest that it substantially overestimates global lake cover, at least by a factor of two. This is probably due to the lack of discrimination between lakes, rivers, and wetlands in that dataset (at least no mention of such a distinction was made in Verpoorter et al. 2014).

Response: Thanks for your meticulous and exhaustive comments. We're encouraged by your dedication and enthusiasm for the scientific scope. We have read these comments carefully and made revisions accordingly, hoping that we have addressed all your concerns.

Main suggestions for improvement:

Point 1: I am unclear about the nature of the labels used in training the deep learning model/supervised classification. The use of robust and reproduceable labels is obviously foundational to the validity of the model outputs, particularly given that the authors do not use independent data sources to validate it. From my understanding, the authors created the labels by masking land in the GSWO (<30% and <5% of water cover frequency out of the valid observations during the past four decades inside and outside floodplains, respectively) and masking ocean and river pixels (using the OSMWL and GRWL datasets, respectively), while retaining pixels overlapping with HydroLAKES polygons. Following this first step, the authors conducted “extensive visual examinations and necessary manual postprocessing corrections were performed to ensure that all extracted lake boundaries (i.e., the lake mask vectorization) matched well with the water/land interfaces isolated on the GSWO maps“ (P18L420). The extent of manual postprocessing corrections is not entirely clear in this description: was every lake polygon/label checked? Of those, how many were manually corrected? And based on what criterion? For instance, in mapping tree crowns, Brandt et al.3 manually delineated individual tree crowns based on the following criteria: “two conditions

had to be fulfilled for a crown to be marked during the manual labelling process: (a) the NDVI value had to be clearly higher than the surrounding (only trees have green leaves in the dry season), and (b) a shadow had to be seen”. In the case of lakes, similar challenges exist, what level of permanence did the authors consider to qualify as characterizing water/land interfaces? In areas of extremely dense lake coverage where differentiating between rivers and lakes is arduous (e.g., across the Canadian Shield), and where the GRWL tends not to include river channels, how did the authors delineate individual lakes (vs. clusters of lakes)? For the sake of reproducibility, I encourage authors to provide intermediate products of this analysis, including the initial mask pre- manual corrections and the final labels/polygons used in training the dataset.

Response 1: Thanks for this comment. In the manual postprocessing process, the lake sample polygons were visually checked from one sample region to another till all were finished, where some lakes in hydrologically complex regions were paid more attention, such as those in the large river basins and floodplain zones. As for the criteria of manual revision, we mainly performed it on the following two cases: (1) river residuals resulting from the absent coverage of the corresponding river masks and (2) river-connected lakes that required further division from river channels. Of all sample polygons, Case 1 polygons frequently occurred, which could take up ~10% of the total lake samples and thus require careful inspection. On the contrary, the percentage of Case 2 polygons was minor (maybe far less than 1%).

In terms of characterizing water/land interfaces, it was mainly determined by the occurrence values of the underlying GSWO map. Pixels within lake outlines have occurrence values of > 5% (and 30% for Floodplain Model), while pixels outside lake outlines exhibited very low occurrence values (< 5% for Normal Model and 30% for Floodplain Model). Actually, this has been considered in our automatic sample extraction approach such that most lake samples fulfilled this requirement, and we usually did not revise their boundaries. Exceptions were those directly connected to rivers as specified above that required further manual editions.

The Canadian Shield was an interesting region, as you mentioned, from two aspects. On the one hand, some lakes formed a similar shape to rivers (long and narrow). On most occasions, we would deem them as lakes if they did not extend to a very far region and were not covered by river masks. For those extremely long polygons, there were possibilities that they would be divided into several pieces on model predictions, from their thin and narrow areas. On the other hand, some lakes were connected to form a cluster. Taken together, we tended to consider them as individual lakes since they hydrologically linked to each other for a long period, and more importantly, segregation between them was hard and lacked uniform criteria. Similarly, some of them were likely to break into pieces from their thin and narrow places in model predictions.

Lastly, relevant intermediate products will be provided upon publication of this manuscript.

Point 2: Related to my previous point, what the lake polygons represent hydrologically should be more explicitly defined. Importantly, is the intent that the polygons represent the average or maximum lake extents? Do the polygons represent permanent lakes or are seasonal lakes also included (and what maximum degree of seasonality is included)? Because lake extent dynamics are only analyzed within the delineated polygons, I assume that the polygons represent their maximum extent. If the polygons are an all-time maximum, I recommend that the overall statistics of the area of lakes and comparisons with other datasets (e.g., Extended Data Fig. 3) be based on the probability-weighted area rather than on the maximum extent. For example, I was really surprised to see >150 more lakes with a surface area >100km² in GLAKES than in HydroLAKES as most of these lakes are rather well-known. Only when I inspected the polygons did I realized that many of those large lakes are >100 km² because it is apparently their maximum extent which was represented.

Response 2: Sorry for any confusion. As we described in the method section, the lake polygons extracted from the U-Net models were supposed to reflect the maximum water extent (specifically, occurrence >5% for major lakes and >30% for lakes in floodplain ideally) over the period of 1984-2019 with both permanent lakes and seasonal lakes included.

In terms of whether to use the probability-weighted area or maximum extent of GLAKES for comparison, we thought both had their own issues. The use of maximum extent, as you mentioned, was likely to result in an overestimation of the lake area, especially for large lakes. However, the use of probability-weighted areas would also lead to an underestimation, especially for seasonal lakes. This's because all global lake datasets being compared also constituted a large proportion of seasonal water pixels. If the probability-weighted area were applied for GLAKES, an occurrence-based weighting factor (0-1) should be multiplied by the unit area of each pixel (30m*30m) to obtain the actual pixel area, which, however, was not done in all the other global lake datasets. In the other datasets, the actual area for seasonal water pixels was counted as the same as the permanent water pixels, regardless of the large discrepancy in occurring frequency. In sum, since all the five global lake datasets were generated from different methods and with different objectives, it's tough to apply the completely uniform criteria for comparison. What we compared here was actually the total lake area bounded by the lake polygons of each dataset (if provided). We have underscored this point in the revised manuscript to avoid misinterpretation (lines 679 to 682).

Point 3: The omission and commission rates are quite high for small and large lakes, respectively. However, the reasons for and implications of this limitation are not discussed. I am not familiar with the inner workings of the deep learning model employed in this study, but would it be possible to train two separate models that would be catered for differently-sized lakes? At least, a quantitative estimate of the uncertainties in the predictions would strengthen this study.

Response 3: First of all, we're sorry to discover that some mistakes were made in the Extended Data Table 1, the new version of which is presented below. Compared to the former version, the commission errors significantly declined among almost all size classes of the two models, while the change of omission errors was minor. In this situation, the commission errors of large lakes were no longer high (~0.1%). Nevertheless, the omission errors of small lakes were still evident, which may probably be induced by the influence of the negative samples. Specifically, in sample preparation, we applied a filter to screen out all lakes with a size of $<0.03\text{km}^2$. In fact, the difference between lakes just exceeding the size threshold and those approximating the thresholds (e.g., 0.031km^2 and 0.029 km^2) was probably minor. Therefore, the setting of a fixed cutting threshold for samples may somewhat "confuse" the model, in a way that some lakes were interpreted as true lakes (because they were marked by our labels) while the others with similar features may be identified by the model as non-lakes (due to the lack of overlaying labels), thus leading to a certain extent of missed detection of small lakes around the size of the cutting threshold (0.03 km^2).

We further examined the error matrix of small lakes with a finer division of size range (see below). Note that the reason why we could obtain results with size $<0.03\text{km}^2$ (below our cutting threshold of samples) was because the U-Net model learns features at patch level, where some lakes with area $>0.03\text{km}^2$ across multiple patches would be split into smaller pieces that might be below 0.03km^2 and interpreted by our models. As seen in the table, the omission and commission errors generally decreased as the lake size increased. The errors were extremely high for lakes with sizes $<0.03\text{ km}^2$ in both models, with omission error of $>50\%$ and commission error of $>15\%$. The accuracy was much higher when considering lakes with sizes $>0.1\text{ km}^2$ (the bottom size threshold for most global lake datasets), where the corresponding omission and commission values dropped to below 20% and 5%, respectively. Nevertheless, although lakes with the size range of $0.03\text{-}0.05\text{ km}^2$ still faced high omission issues, the commission error declined suddenly from $>15\%$ to $\sim 5\%$. In practice, we treasured commission error more than omission error to ensure that the detected portion of our GLAKES polygons was generally "true" and thus could be placed with more confidence in further analysis. Therefore, we kept the size threshold as 0.03km^2 to include more lakes in our dataset without much compromise on misclassification.

As for the second question, we thought the use of two separate models to train lakes of different sizes was not necessary. On the one hand, the lake size has already been considered in our model by specifying an individual region type to capture the characteristics of large lakes, and the accuracy of large lakes presented in the updated Extended Data Table 1 was generally satisfying. On the other hand, as we stated above, the high omission values of small lakes mainly resulted from the setting of a fixed cutting threshold, which could not be well improved by just utilizing two separate models. A better solution may be the inclusion of samples with sizes $<0.03 \text{ km}^2$ in model training, followed by a result-oriented determination of the cutting threshold for lake predictions (i.e., finding out a threshold where the accuracy for lakes below the certain size threshold was unacceptable, if possible). Nevertheless, despite being troubled by the relatively high omission errors for small lakes, our GLAKES dataset showed marked improvements over previous global lake datasets, considering its advantages in global coverage (60°S - 80°N), high spatial resolution ($\sim 30\text{m}$), long-term changes (four decades), spatiotemporal consistency (uniform mapping of global lakes instead of aggregation from different lake datasets), overall accuracy (overall accuracy $>98.7\%$ and MIoUs $>88.7\%$), and the delineation of small lakes (lower limit as 0.03 km^2). Besides, the lake change analysis was performed only within the boundaries defined by our GLAKES dataset (see below); therefore, the associated impacts of the classification errors (particularly the omission errors) should be limited. Of course, there was still room for improvement in our GLAKES dataset. An example was the above-mentioned inclusion of lakes $<0.03\text{km}^2$ in model training, along with a statistics-based decision of the cutting threshold for predicted lake polygons, which may be considered in our next version of GLAKES in the future.

Extended Data Table 1 | Accuracy assessments of our developed deep-learning algorithm at different lake-size classes.

Error matrix for the GLAKES dataset estimated using independent test labels for the Normal Model and Floodplain Model, listing the accuracy levels derived for different lake size groups.

	Size	Omission (%)	Commission (%)	MIoU (%)	Overall Accuracy (%)
Normal	Small (0.03-1 km ²)	23.5	2.5	94.0	99.3
	Medium (1-100 km ²)	4.2	0.4		
	Large (>100 km ²)	1.1	0		
	All	5.4	0.5		
Floodplain	Small (0.03-1 km ²)	21.2	5.0	88.7	98.7
	Medium (1-100 km ²)	7.3	1.7		
	Large (>100 km ²)	9.4	0.2		
	All	9.6	1.9		

Table S3 | A further investigation of the accuracy of our developed deep-learning algorithm in subdivided small lake groups.

Error matrix for the GLAKES dataset estimated using independent test labels for the Normal Model and Floodplain Model, listing the accuracy levels derived for subdivided size groups of small lakes. Both pixel-based and polygon-based results are included for assessments. Likewise, the polygon-based omission and commission values presented below represent the average of the corresponding values of all lake polygons being assessed.

Size	Pixel-based		Polygon-based		
	Omission (%)	Commission (%)	Omission (%)	Commission (%)	
Normal	<0.01 km ²	85.3	76.6	91.4	88.1
	0.01-0.03 km ²	52.6	17.4	54.9	22.3
	0.03-0.05 km ²	50.5	3.6	51.5	3.7
	0.05-0.1 km ²	34.5	3.1	35.0	3.0
	0.1-1 km ²	18.9	2.3	22.2	2.5
Floodplain	<0.01 km ²	81.4	73.0	88.6	86.1
	0.01-0.03 km ²	51.4	21.1	53.7	25.2
	0.03-0.05 km ²	51.2	6.7	52.8	6.8
	0.05-0.1 km ²	31.3	6.4	32.0	6.4
	0.1-1 km ²	17.0	4.7	19.5	4.8

Point 4: Multiple studies have previously highlighted the outsized role of small lakes for several global processes and this study adds a salient piece to the puzzle. Nonetheless, I suggest that the findings from this study regarding the role of small lakes need to be caveated in two main ways:

a. It makes sense that small lakes display higher relative temporal variability (smaller volume to area ratio, greater sensitivity to catchment, etc.). However, it is important to highlight potential biases in quantifying the amplitude of this variability that are due to scale. Because

the size of individual pixels is large compared to the total area of small lakes, the default/random variability in extent of small lakes is higher. In a lake spanning 0.05 km², a single 900-m² pixel going from wet to dry leads to a ~2% change. I wonder how similar the variability in extent would be between small and large lakes if it were evaluated with equal relative pixel sizes (e.g., 5 m for a 0.05 km² lake and 5 km for a 50 km² lake).

b. The outsized role of small lakes in driving carbon emissions may be largely driven by the fact that reservoirs were excluded from the calculations. My guess is that the inclusion of emissions from large reservoirs may change this finding. I suggest that this be mentioned. Moreover, all sections discussing the carbon emission estimates should use the term “natural lakes” rather than just “lakes” (as the latter refers to both lakes and reservoirs by default in the manuscript; P4L70). The reporting of the findings would thus be clearest to readers who may not read the entire piece (i.e., most readers), avoiding that this work be miscited. This is not currently clear in the abstract for example.

Response 4: Thanks for your encouraging comment on our work. As suggested, we used the Tibetan Plateau as a case study to investigate the impact of the size scale on the estimation of relative area changes of lakes. We resampled the water occurrence map as 300m and compared it with the results derived from 30m (see the figure below). It can be seen that the probability-weighted lake area at different spatial resolutions of the occurrence map generally agreed well, with a concentrated distribution along the 1:1 line (subplots a/b), indicating that the impact of resampling on lake area estimation was minor. Furthermore, we calculated the relative area change rate for lakes within the size range of 0.5-1.5 km² at the resolution of 30m and compared it with those between 50-150 km² at the 300m resolution scale (fig. c). As is presented, the relative area change rate for lakes between 50-150 km² concentrated within $\pm 10\%$, while the relative area change rate for lakes between 0.5-1.5 km² span across $-10^3 \sim 10^3$, the range of which was far larger than the former lake group. Therefore, it's clear that small lakes still exhibited higher variations in relative change rate when evaluating under equal relative pixel sizes. In addition, as we mentioned, “small lakes supplied a disproportionately large contribution to global lake expansion, representing 46.2% of the net areal increase from 1980-90s to 2010s.”. We also demonstrated that “the changes in small lakes showed dominant contributions (>50%) in approximately half of the examined inland regions (49.9% of the grid cells from 1980-90s to 2000s, and 50.1% from 2000s to 2010s)”. These contributions were all assessed on the basis of the absolute area (change) that would not be influenced by the size scale issue. Based on the above analysis, we could state that our conclusion about the outsized role of small lakes in driving global lake area changes was robust and reasonable.

As for the estimation of carbon emission section, after a review of Holgerson and Raymond (2016), we discovered that they only excluded the in-situ samples of CH₄/CO₂

fluxes of reservoirs but combined the natural lake area with reservoir area in global upscale (i.e., they used GLOWABO dataset which did not distinguish between natural lakes and reservoirs). This indicated that they actually regarded reservoirs as natural lakes in terms of carbon emission estimation, like several other studies (Raymond et al. 2013, DelSontro et al. 2018, Li et al. 2020). Therefore, we decided to add the reservoir area into the total lake area and update our new results, while we informed readers that the waterbody type (natural lakes vs reservoirs) matters in carbon emission estimates for global lakes in Supplementary Note 3: “Nevertheless, previous studies had revealed that methane emissions in different water body type might be driven by different processes, which would inevitably impact the final estimation results (Deemer et al. 2016, Hayes et al. 2017, Deemer and Holgerson 2021)”. After incorporating the reservoir area into the total lake area, the contribution of small lakes to total emission decreased given the relatively large size of reservoirs. Nevertheless, the roles of small lakes in modulating lacustrine carbon emissions were still evident, considering the small proportion of lake surface area. Please refer to the section “New estimates of lacustrine carbon emissions” for more detailed updated results (lines 171 to 217).

Fig. Comparison of estimated probability-weighted lake area at different spatial resolutions of the occurrence map in Tibetan Plateau, while (a) is presented for the period 1984-1999 and (b) is for 2010-2019. (c) Comparison of the relative area change rate distribution for lakes in the Tibetan Plateau within the size range of 0.5-1.5 km² at the resolution of 30m (red) vs. those between 50-150 km² at the 300m resolution scale (purple). Note that the total area of the histogram equals 1 for both the blue and purple sides. The range of change rates for smaller lakes (0.5-1.5 km²) is larger than that of

larger lakes (50-150 km²) even when evaluating under the equal relative pixel scale (30m vs. 300m).

Point 5: The main text currently contains no discussion on sources of uncertainty, I highly encourage that a substantial paragraph or section be dedicated to the main sources and extents of uncertainty affecting this study. For instance, discussions on the difficulty of disentangling lakes from temporarily inundated floodplains or agricultural fields would be needed.

Response 5: Thanks for this valuable suggestion. We have added a section called “uncertainty and limitation” with substantial content for discussion (lines 777 to 788 and Supplementary Note 3).

“Several uncertainties or limitations should be acknowledged in this study, both during the process of lake mapping along with related change analysis of lake area and carbon emissions. In lake mapping, these could be further categorized into the following major sources: lake definition, auxiliary datasets, U-Net model, and post-processing. The temporal change of probability-weighted lake area among different time periods, otherwise, may be influenced by seasonal lake dynamics. As for the estimation of global carbon emissions as well as their long-term change, the accuracy of our results were closely related to the representativeness of the average emission rates used for global upscaling, the impacts of lake dynamics at shorter timescales, and the quantification of emissions through the different pathway (for CH₄).

First of all, our global lake coverage did not include ocean-connected lakes and those beyond the latitude range of the GSWO dataset (i.e., 60°S-80°N). In addition, as illustrated above, the features of lakes and rivers could be well distinguished by the model in most regions. Nevertheless, there were lakes that had similar shapes as rivers but were usually short in length, such as some lakes in the Canadian Shield and oxbow lakes alongside rivers. This may be responsible for the relatively high omissions of the region type AR. In addition, a variety of reservoirs were actually built upon river channels, which were probably identified as rivers and thus resulted in missed detection. To solve this issue, we replaced the U-Net predictions of the on-river reservoirs (defined by GRWL) with those yielded from the automatic extraction method used in sample preparation.

Besides, the disentanglement of lakes from floodplains was also challenging as the definition of floodplain extent was ambiguous and arduous. Here we used a globally uniform threshold (30% occurrence) to depict the lake/floodplain interface alongside rivers, which may cause bias on a regional scale. Besides, in post-processing procedures, the floodplain buffers were restricted by the presence and accuracy of the GRWL layer,

leading to negligence of floodplains outside the 1km river buffers and those resulting from the absence of the underlying GRWL mask. Equally challenging was the division between natural lakes and agricultural fields. First of all, given the inherent constraint of the GSWO dataset, not all paddy fields were mapped (Pekel et al. 2016). Besides, the size threshold of 0.03 km² helped screen out some small and isolated agricultural fields. In addition, compared to natural lakes, a vast proportion of agricultural fields exhibit regular shapes, grided textures, and low occurrence in GSWO maps, serving as a basic principle for discrimination from natural lakes. However, since the focus of this study was mainly the partition of lentic and lotic water systems, these human-transformed water bodies were temporarily considered as lakes in our GLAKES dataset.

The setting of a pre-defined fixed cutting threshold was the major reason for the relatively high omission errors for small lakes (especially those around 0.03km²). This could be induced by the influence of the negative samples. Specifically, in the process of sample preparation, we applied a filter to screen out all lakes with sizes <0.03km². In fact, the difference between lakes just exceeding the size threshold, and those approximating the thresholds (e.g., 0.031km² and 0.029 km²) were probably minor. Therefore, the setting of a fixed cutting threshold for samples may somewhat “confuse” the model, in a way that some lakes were interpreted as true lakes (because they were marked by our labels) while the others with similar features may be identified by the model as non-lakes (due to the lack of overlaying labels), thus leading to a certain extent of missed detections of small lakes around the size of the cutting threshold (0.03 km²). Under such circumstances, a better solution may be the inclusion of samples with sizes <0.03 km² in model training, followed by a result-oriented determination of the cutting threshold for lake predictions (i.e., finding out a threshold where the accuracy for lakes below the certain size threshold was unacceptable, if possible).

Uncertainty and limitation could also result from the auxiliary dataset, such as the river mask and water occurrence map used for lake mapping. According to the developing procedure of the GSWO map, only waters that were visible from space (i.e., Landsat observation) without any overlaying obstacles were able to be included in the final surface water mapping. This could lead to missed detection of frozen lakes and vegetated wetlands and thus an underestimation of lake coverage in regions like the Canadian Shield and Scandinavia. In addition, it could be observed that inconsistencies between the GSW occurrence map and the GRWL and OSMWL river masks existed in many regions. On the one hand, the coverage of GRWL and OSMWL for many large rivers was inadequate compared to that of the occurrence map. Application of these river masks for river exclusion would lead to a large number of river residuals that required further elimination (especially through manual revision). On the other hand, a small percent of lakes also had the risk of being masked mistakenly due to such

inconsistencies. Nevertheless, the overall accuracy of the utilized river masks was generally satisfactory, and the impact of dataset inconsistency could be reduced by using the area ratio before and after- the river mask, as mentioned above.

The capability of the U-Net models in differentiating large rivers with broad river widths from lakes would be constrained by the scale of the patches that served as the basic unit for feature learning. An enlargement of the patch size could partially solve this problem but would greatly increase the GPU memory as well. The 512 x 512 pixel was the largest scale that was applicable in our study given our maximum GPU RAM of 24 GB. Hence, we utilized auxiliary river masks in label generation as well as the post-processing process for exclusion of the remaining rivers, although this introduced new uncertainty as stated before.

In temporal change analysis, owing to the constraint of the valid observations of Landsat images, this study mainly focused on the changes in lake extent at a decadal scale. However, lake dynamics at shorter timescales could also be evident. Pickens *et al.* discovered that only 23% of the total area of open surface water was permanent without ice cover within 2019, while permanent water covered by seasonal ice/snow constituted 41% of the total area, and the remaining 36% were made up of seasonal waters regardless of ice coverage (Pickens *et al.* 2022). Furthermore, such a seasonal pattern of water/land/ice transition may witness substantial changes during the whole study period owing to the impact of climate change, including an intensifying reduction in ice cover duration and varied changes in wetting/drying trends (Magnuson *et al.* 2000, Sharma *et al.* 2019, Greve *et al.* 2014, Roderick *et al.* 2014, Woolway *et al.* 2020). These changes at a shorter timescale may impose divergent impacts on our decadal change analysis of lake surface area. The impact of seasonal water/land transition along with its trend was minimal as they have already been incorporated into the occurrence map and thus the calculation of probability-weighted lake area each period. However, the negligence of lake ice coverage (in the GSW MWH dataset, ice was flagged as invalid observation) might lead to a conservative extraction of the lake outlines as well as an underestimation of the water occurrence value. Given that the extension of the ice-free season was reported to exhibit an increasing trend (Woolway *et al.* 2020, Wang *et al.* 2022), the underestimation of the probability-weighted lake area might be less severe in the more recent period, indicating that there might be a slight overestimation of the calculated lake area changes over different periods in places covered with ice.

In this study, the method we used had already been applied in a previous study to calculate lacustrine carbon emissions (Holgerson and Raymond 2016). It was calculated by multiplying the average emission rate (flux) of different lake size classes by the total lake area accounting for corresponding size bins. However, the average emission rates

were aggregated from finite in-situ measurements that may not reflect the true global distribution (Rosentreter et al. 2021, DelSontro et al. 2018), which limited the accuracy of our global estimates. In particular, the empirical size-dependent flux values listed in Holgerson and Raymond (2016) were derived from in situ lake samples with certain geographic dependencies (e.g., the majority of sampled lakes were located within the range of 30°N-70°N), which may cause uncertainty when upscaling for global lakes. Besides, the incorporation of other relevant drivers such as the water body type, water depth, water productivity, sediments as well as ecoclimate zone would also enhance the representativeness of the average emission rates for more accurate global estimates (Rasilo et al. 2015, Wik et al. 2016, DelSontro et al. 2018, Deemer and Holgerson 2021). Using the water body type as an example, here we combined the total area of reservoirs and natural lakes to obtain the global estimation results, without more discrete classification, as in several other studies (Raymond et al. 2013, DelSontro et al. 2018, Li et al. 2020). Nevertheless, previous studies have revealed that methane emissions in different water body types might be driven by different processes, which would inevitably impact the final estimation results (Deemer et al. 2016, Hayes et al. 2017, Deemer and Holgerson 2021).

In addition, in calculating the changes over three periods, we kept the average flux values as constant, considering only the long-term carbon emission changes that were brought by the lake area variations over different time periods. Nevertheless, the transfer of carbon gases from the aquatic environment to the atmosphere is a highly dynamic process, which could also be modulated by lake dynamics (such as ice phenology and water/land transitions) at shorter timescales (Chamberlain et al. 2016, Deemer et al. 2016, Holgerson and Raymond 2016, Wik et al. 2016, DelSontro et al. 2018, Keller et al. 2021). It has been reported that CO₂ and CH₄ accumulate under the ice, and subsequently vent a substantial amount to the atmosphere during the spring melt, during which CH₄ oxidation may co-occur, although this is probably not applicable to oligotrophic lakes or completely frozen lakes (Bastviken et al. 2004, Kortelainen et al. 2006, Michmerhuizen et al. 1996, Utsumi et al. 1998). Considering the trend of global warming over the study period, the ice-free seasons for most lakes extended, and some permanently frozen lakes became seasonally ice-covered, leading to a further boost to global carbon emissions (Natchimuthu et al. 2014, Sharma et al. 2019, Wik et al. 2016). Besides, the seasonal drying and wetting of lakes was also an important carbon emission source. Studies have revealed complex relationships between water level and aquatic carbon emissions and identified dry aquatic sediments as significant carbon gas hot spots (Chamberlain et al. 2016, Keller et al. 2021, Marcé et al. 2019, Tangen and Bansal 2019), which we also did not account for. Given the varied wetting/drying trends across different regions globally (Greve et al. 2014, Roderick et al. 2014, Woolway et al.

2020), the overall impact on long-term carbon emission changes is hard to quantify, and more data are required for systematic evaluations.

While our study only paid attention to the diffusive CH₄ flux, CH₄ ebullition was deemed the dominant CH₄ emission pathway (Bastviken et al. 2004). However, the data availability of direct in-situ ebullition rate measurements as well as their high spatiotemporal variability hampers the systematic assessments of CH₄ ebullition on a global scale (Beaulieu, McManus and Nietch 2016, Holgerson and Raymond 2016). Nevertheless, although the accuracy of our global-scale carbon emission estimates would be impacted by the above-mentioned factors, the main objective of this section is to highlight the essential roles of small lakes in driving lacustrine carbon emissions, which remain robust.”

Point 6: Overall, the deep learning model deserves to be explained more clearly and at greater length. Additional explanations should be provided as to how the deep learning model functions and a table of hyperparameters should be provided in the supplementary material. Additional information on why some decisions were taken is needed (see my specific comments further on), which could be complemented by information on the sensitivity of model outputs to these decisions.

Response 6: Thanks for this valuable suggestion. We have added more details of the U-Net model and illustrated how it worked and why it's suitable for such kind of classification problem.

“Deep learning has been widely used in many areas (Krizhevsky et al. 2012, Hinton et al. 2012, Sutskever et al. 2014, LeCun et al. 2015), and are proven to be a powerful and creative tool in detecting features of interest from satellite images (Ma et al. 2017, Weiss et al. 2020, Reichstein et al. 2019). A recent inspiring deep learning application in remote sensing image processing was documented by Brandt et al. (Brandt et al. 2020), who detected tree crowns by combining the U-Net model with submeter high-resolution satellite images. The U-Net model used in Brandt et al. (Brandt et al. 2020) is a typical semantic segmentation technique that performs pixel-wise classification within an image for precise segmentation (Yu et al. 2018, Ronneberger et al. 2015). Compared to the conventional classification tasks, U-Net yields not only the label category of a specific image, but also its corresponding location. Upon the application of U-Net, Brandt et al. (Brandt et al. 2020) managed to map more than 1.8 billion non-forest tree crowns (>3 m²) in the West African Sahara and the Sahel, somewhat overturning the previous stereotype of trees scarcity in these dryland regions. Here, we modified the U-Net model developed by Brandt et al. (Brandt et al. 2020) and transferred its application to global lake mapping. We expect a well-trained U-Net model to perform well when classifying

lakes and rivers using GSWO images, as lakes and rivers are already highly distinguishable in visual examinations (Extended Data Fig. 3).

Building upon a fully convolutional neural network (Long et al. 2015), the U-Net model composes of various and hierarchical convolution layers that are widely used in semantic segmentation field for feature detection (LeCun et al. 2015, Ronneberger et al. 2015, Liu 2018), vital for the extraction and segmentation of lakes from rivers. The convolution layer extracts features from an image in the previous layer and results in a less redundant output image called feature map. Generally, the features learned by convolution layers transition from simple to more abstract ones as the level of the convolution layers increase (Zeiler and Fergus 2014, LeCun et al. 2015, Ribeiro et al. 2018), and these features are determined by the convolution kernels (i.e., an array of weights) that are learned automatically through backpropagation. Except for convolution layers, there are various structures that are also essential in the modified U-Net architecture of Brandt et al. (Brandt et al. 2020), including activation function (enabling nonlinear classification), batch normalization (stabilizing and accelerating the training process), pooling (reducing data dimension and computation complexity), up-convolution (restoring the size of feature maps for precise localization) as well as concatenation (combing the higher-level feature map with a lower-level one to better learn representation). In addition, the U-Net model adopts the overlap-tile strategy, which makes it possible to perform a seamless segmentation for extremely large images without losing information about the divided border regions (Ronneberger et al. 2015). This makes U-Net particularly applicable to our goal of pixel-wise lake classification through the GSWO images at a global scale.

Specifically, the U-Net model comprises two major parts: a contracting path for feature interpretation and a near symmetric expanding path for location identification, leading to a u-shaped architecture that enables pixel-to-pixel classification (Ronneberger et al. 2015). In the contracting path, the input feature map undergoes four repeated blocks for downsampling, consisting of two 3x3 convolution layers (each accomplished with a rectified linear unit (ReLU) activation function), a batch normalization layer and a 2x2 max-pooling layer. Notably, the feature channels double after each downsampling process and then half after each upsampling process in the expanding path. Likewise, the expanding path consists of four comparable blocks for upsampling. The difference is that once a 2x2 up-convolution and batch normalization are conducted on the feature map, a concatenation will be performed with its cropped feature map from the corresponding contracting path, and together they go through two 3x3 convolutions activated by ReLU. Finally, a 1x1 convolution layer is used to produce the final classification map.”

The key hyperparameters that were adjusted in our study are presented in the table below. We have tried different sets of hyperparameters to assess the model's performance, and the set of hyperparameters presented below was determined for the final model. We're sorry that since many of the former results were not saved, we could not provide relevant sensitivity analysis to assess how our model would be impacted by these hyperparameters. Nevertheless, we have already tested the performance of the final model from many aspects (see "Accuracy assessments and comparisons" section and Supplementary Note 2) and have proved its applicability in global lake mapping. This has already met our expectations since the main focus of this study is not the accuracy of specific algorithms, but the broader applications in global lake mapping.

Table S1 | Keep hyper-parameters that were tested and adjusted in the U-Net Model. The set of hyperparameter values applied for the final model is also presented.

Hyperparameters	Setting
Optimizer	Adadelta
Loss function	Tversky: $\alpha = 0.5$, $\beta = 0.5$
Batch size	16
Iteration	Normal model: 750; Floodplain model: 600
Epoch	250
Patch size	512 × 512

Point 7 : An assessment of the spatial distribution of uncertainties would be valuable. In which regions does the model perform better or worse? This could be implemented through spatial cross-validation.

Response 7: Thanks for this valuable suggestion. The spatial distributions of the omission and commission errors of each patch in the test set are presented below. Overall, the omission error was generally higher than the commission errors in most places regardless of the model type, with the median omission error being 15.76% and 14.05% and median commission errors being 1.25% and 1.71% for the Normal Model or Floodplain Model, respectively. Spatially, the high omission errors (>20%) in terms of Normal Model are mainly distributed in Alaska, Siberia, and the Amazon basin. We also revealed that most of these omissions happened in region type 4 (i.e., "lakes alongside river"), the major omission source of the Normal Model. In addition, Alaska and Siberia also contributed the most in terms of the number of high omission patches for the Floodplain Model. High omissions also occurred in several other large river floodplains, such as the Ganges River in India and the LaPlata-Parana River in South America. On the other hand, the commission error of the majority of test patches (> 90% for both

models) was below 10%, while only a few left high commission patches distributed sparsely and irregularly across the globe. The related changes have been made in Supplementary Note 2 and Fig. S2.

Fig. S2 | Validation of the deep-learning algorithm across the globe. Accuracy assessments (omission/commission errors) for each sample region across the globe based on independent test labels, where the Normal Model and the Floodplain Model are evaluated separately.

Point 8: Currently, the labels used for assessing the model performance are only partly independent, as they were created through the same semi-automatic process using the same data source (GSWO). Adding a truly independent validation, based on a different sensor and/or higher resolution imagery (e.g., using Sentinel data as was most recently done by Pickens et al.7), would strengthen the evaluation of the model performance.

Response 8: Thanks for this suggestion. Theoretically, applying an independent test set on the basis of other sources instead of GSWO may be a more proper alternative for the performance assessment of the U-Net models. However, the lake polygons depicted in our GLAKES dataset were a record of lake variations during the past four decades (1984-2019). Except for GSWO, it seemed that there's no such dataset that covered the lake changes during such a long period, such as the Global surface water dynamics dataset (1999-2020) from Pickens et al. (2020), the Global 3-second/1-second Water Body Map dataset (1990-2010) from Yamazaki, Trigg and Ikeshima (2015), the GIEMS-2 dataset (1992-2015, $0.25^\circ \times 0.25^\circ$) from Prigent, Jimenez and Bousquet (2020) and so on. Similarly, the application of sentinel images mentioned above was also constrained by the mismatch of the observation period. Therefore, we maintained our test set for evaluation of the model performance in the absence of a better option.

Point 9: The evaluation of model performance (Extended Data Table 1) is only provided at the pixel level whereas this study also produced a polygon dataset. To better grasp uncertainties, I encourage the authors to provide a polygon-based performance assessment (omission and commission at the lake entity level as well as measures of fit and bias between test polygons and output polygons).

Response 9: Thanks for this suggestion. We have added the polygon-based performance assessment in Table S2, Table S3 and Fig. S1. As shown in Fig. S1, our prediction also agreed well with the label area at the polygon base. Notably, the Percent Bias (PBIAS) here remained the same as the pixel-based assessment results, considering the implication of PBIAS. In addition, we calculated the omission and commission errors for each pair of lake polygons, and the averaged results are presented in TableS2 and Table S3. Overall, we observed generally larger omission errors for both models compared to the pixel-based evaluation, in particular for small lakes where the mean omission errors exceeded 30%. In addition, here, the average omission errors for size group “all” were actually biased by the prevalent occurrence of small lakes and thus also appeared high. On the other hand, the commission errors remained low for both models (< 6% across all size groups), reaffirming the high accuracy of our models in terms of misclassification.

Fig. S1 | Validation of the deep-learning algorithm at the lake-entity level. The predicted area for each lake polygon is compared against the corresponding label area at the lake entity level. Region types are also annotated.

Table S2 | Accuracy assessments of our developed deep-learning algorithm at the lake-entity level for lakes with different size classes.

Error matrix for the GLAKES dataset estimated using independent test labels for the Normal Model and Floodplain Model, listing the accuracy levels derived for different lake size groups at the lake-entity level. The omission and commission values presented below represent the average of the corresponding values of all lake polygons being assessed.

	Size	Omission (%)	Commission (%)
Normal	Small (0.03-1 km ²)	35.0	3.0
	Medium (1-100 km ²)	8.3	1.0
	Large (>100 km ²)	1.2	0
	All	32.6	2.8
Floodplain	Small (0.03-1 km ²)	32.3	5.7
	Medium (1-100 km ²)	10.6	3.4
	Large (>100 km ²)	9.2	0.2
	All	30.4	5.5

Table S3 | A further investigation of the accuracy of our developed deep-learning algorithm in subdivided small lake groups.

Error matrix for the GLAKES dataset estimated using independent test labels for the Normal Model and Floodplain Model, listing the accuracy levels derived for subdivided size groups of small lakes. Both pixel-based and polygon-based results are included for assessments. Likewise, the polygon-based omission and commission values presented below represent the average of the corresponding values of all lake polygons being assessed.

Size	Pixel-based		Polygon-based		
	Omission (%)	Commission (%)	Omission (%)	Commission (%)	
Normal	<0.01 km ²	85.3	76.6	91.4	88.1
	0.01-0.03 km ²	52.6	17.4	54.9	22.3
	0.03-0.05 km ²	50.5	3.6	51.5	3.7
	0.05-0.1 km ²	34.5	3.1	35.0	3.0
	0.1-1 km ²	18.9	2.3	22.2	2.5
Floodplain	<0.01 km ²	81.4	73.0	88.6	86.1
	0.01-0.03 km ²	51.4	21.1	53.7	25.2
	0.03-0.05 km ²	51.2	6.7	52.8	6.8
	0.05-0.1 km ²	31.3	6.4	32.0	6.4
	0.1-1 km ²	17.0	4.7	19.5	4.8

Point 10: I opened and visualized the dataset with no difficulty. It fits the description provided in the manuscript. The documentation is clear. Although it may not be in its final

form, I suggest that the authors also provide a license for the data, and that the database be available as a Shapefile and/or geo-package to enable a greater range of users to access these data.

Response 10: Thanks for this suggestion. The license has been added and the .shp format dataset will be provided upon the publication of this manuscript. We have also added several properties for each lake to describe the different geographic conditions (i.e., glacier-fed/non-glacier-fed, permafrost-fed/non-permafrost-fed, human-regulated/natural formed, and endorheic/exoreic). Hope this can help readers with broader applications of the GLAKES dataset.

Miscellaneous comments and suggested improvements:

Point 11: P2L29 “explicit extents and changes”: this relates back to my previous comment on the meaning of the GLAKES polygons. “explicit” extent is not clear, is the dataset about average or maximum extent?

Response 11: Thanks for this suggestion. The lake polygons our dataset provided represent the maximum extent, while the lake changes throughout different time periods are presented using probability-weighted area. Therefore, the original statement has been replaced by “Here, we map 3.4 million lakes at a global scale, including their explicit maximum extents and probability-weighted area changes over the past four decades.”.

Point 12: P2L30: “Lake area increased across all six continents”, an explicit mention of the dates used in the study here would be useful.

Response 12: Thanks for this comment. The relevant sentence has been rewritten as “From the beginning period (1984-1999) to the end (2010-2019), lake area increased across all six continents analyzed, with a net change of +46,278 km², and 56% of the expansion was attributed to reservoirs.”.

Point 13: P2L32 “global lake areas” and “variabilities”. I believe that both of these should be singular.

Response 13: Corrected as suggested.

Point 14: P2L35: “Our findings illustrate the emerging roles of small lakes in regulating local inland water variabilities and greenhouse gas emissions.” The results suggest that small lakes

don't only regulate local inland water (extent?), but also the global dynamics of surface water extent and greenhouse gas emissions.

Response 14: Thanks for this suggestion. The statement has been reframed as “Our findings illustrate the emerging roles of small lakes in regulating not only local inland water variability, but also the global trends of surface water extent and carbon gas emissions.”

Point 15: To be specific, the authors could refer to carbon emissions rather than greenhouse gas emissions, as N₂O was not studied here (despite its role as a GHG11).

Response 15: Thanks for this suggestion, we have replaced all “greenhouse gas emissions” with “carbon emissions”.

Point 16: P3L39: “...underpin vital ecosystem function and services” for the sake of thoroughness, please provide a citation.

Response 16: Thanks for this suggestion. Two relevant references below have been added:

1. Schallenberg, M. et al. Ecosystem services of lakes. *Ecosystem services in New Zealand: conditions and trends*. Manaaki Whenua Press, Lincoln, 203-225 (2013).
2. Reynaud, A. & Lanzaova, D. A global meta-analysis of the value of ecosystem services provided by lakes. *Ecological Economics* 137, 184-194 (2017).

Point 17: P3L41: Woolway, R. I., Kraemer, B. M., Lenters, J. D., Merchant, C. J., O'Reilly, C. M., & Sharma, S. (2020). Global lake responses to climate change. *Nature Reviews Earth & Environment*, 1(8), 388-403. Would be more adequate (global scale) for this statement than O'Beirne. The following study would also be relevant to this statement: Grant, L., Vanderkelen, I., Gudmundsson, L., Tan, Z., Perroud, M., Stepanenko, V. M., Debolskiy, A. V., Droppers, B., Janssen, A. B. G., Woolway, R. I., Choulga, M., Balsamo, G., Kirillin, G., Schewe, J., Zhao, F., del Valle, I. V., Golub, M., Pierson, D., Marcé, R., ... Thiery, W. (2021). Attribution of global lake systems change to anthropogenic forcing. *Nature Geoscience* 2021, 1–6. <https://doi.org/10.1038/s41561-021-00833-x>

Response 17: Thanks for this detailed suggestion. We replaced O'Beirne et al. (2017) with the two aforementioned references.

Point 18: P3L57: “Estimates of the global extent of lakes are available” would be a more exact description of these datasets.

Response 18: Corrected as suggested.

Point 19: P4L65: “However, these available global assessments...” it is not entirely explicit in this paragraph what “these assessments” refer to. Please clarify.

Response 19: We clarified as “these available global assessments of *ecosystem parameters*.....”

Point 20: P4L71: “We used deep learning to identify lakes smaller than the minimum mapping unit for all previous global lake datasets (0.1 km²).” Verpoorter et al.9 describe mapping lakes down to 0.002 km² (nine Landsat pixels).

Response 20: Thanks for this comment. We’re aware that the GLOWABO dataset has extracted lakes as small as 0.002 km² in size. However, the GLOWABO dataset is not publicly available. We have revised this sentence “smaller than the minimum mapping unit for global lake datasets *that are publicly available* (0.1 km²)”. Revisions have been made accordingly on places where similar descriptions occur.

Point 21: P4L81: “Deep learning makes it possible to detect lakes as small as 0.03 km² (corresponding to approximately 33 Landsat image pixels)...”. This may be due to my misunderstanding of what is involved in the deep learning algorithm, but I am not clear about how deep learning itself enables detecting small lakes. This statement deserves additional explanation/justification.

Response 21: Thanks for this suggestion. The actual procedure is that at first we predefined the cutting threshold of samples as 0.03 km² (33 pixels). Then the lake samples with sizes above this threshold were trained by the models to extract their underlying features and yield predictions. The results would be impacted by the value of the cutting threshold during this process, as we observed a vast amount of tiny lakes with size <0.03 km² unable to be detected by models. Of course, there were indeed some lakes below the threshold that were extracted by our models, which were then further screened out during our post-processing procedure so that our remaining lake polygons were all >0.03 km². In this way, we were able to “extract lakes as small as 0.03 km² by using deep learning algorithm”.

Point 22: Figure 1: This is a nicely done and informative figure. Good job! For panel b, given that a degree square has a substantially different surface area depending on latitude, I recommend that the lake extents be expressed as limnicity (% land area covered by lakes) rather than absolute area (km²).

Response 22: Thanks for this encouraging comment and good suggestion. The grided lake extents were expressed as limnicity as suggested (see below). While expressing as lake area density was better than the absolute area, the overall global lake pattern was generally consistent with the previous result.

Fig. 1 | Spatial distribution of global lakes. Lakes with maximum surface area >0.03 km² were mapped, showing (a) lake count (total number of lakes) and (b) lake area density (total lake area/grid area) per 1°x1° grid cell. The latitudinal and longitudinal lake profiles summarizing (by 1°) the lake count and lake area are shown on the right and bottom of (b). Statistics for small (<1 km²), medium (1-100 km²), and large (>100 km²) lakes are presented within each panel.

Point 23: P6L103: For ease of reading, I suggest simply using the actual years (1980-90s, 2000s, and 2010s) throughout the manuscript rather than P1, P2 and P3.

Response 23: Thanks for this suggestion, we replaced all “P1”, “P2”, “P3” with the actual years they indicated.

Point 24: P10L189: It is not clear from the methods whether this estimate is determined with the total area of the polygons (that would mean the maximum water extent based on my observation of the dataset, and would thus potentially represent an overestimate) or the probability-weighted area over a given period.

Response 24: Thanks for this detailed suggestion. The total CO₂ and CH₄ emissions (Fig. 4a, Extended Data Fig. 9) were estimated from the total area of the lake polygons, while the carbon emission changes over different periods were estimated based on the probability-weighted area (Fig. 4b). The reason why we used polygon area instead of probability-weighted area for comparison with the previous study was analogous to response 2. Please see response 2 for more information.

Point 25: Figure 4. I suggest using the same colour set for small, medium and large lakes here as in panels a and b of Figure 3.

Response 25: Thanks for this detailed suggestion. The color set has been changed as suggested.

Point 26: P12L226: Very interesting finding!

Response 26: Thanks for this encouraging comment.

Point 27: P12L235: The following citations are quite relevant here for discussion:

o Keller, P. S., Marcé, R., Obrador, B. et al. Global carbon budget of reservoirs is overturned by the quantification of drawdown areas. *Nature Geoscience*. 14, 402–408 (2021). <https://doi.org/10.1038/s41561-021-00734-z>

o Johnson, M. S., Matthews, E., Bastviken, D., Deemer, B., Du, J., & Genovese, V. (2021). Spatiotemporal methane emission from global reservoirs. *Journal of Geophysical Research: Biogeosciences*, 126, e2021JG006305. <https://doi.org/10.1029/2021JG006305>

o Bridget R. Deemer, John A. Harrison, Siyue Li, Jake J. Beaulieu, Tonya DelSontro, Nathan Barros, José F. Bezerra-Neto, Stephen M. Powers, Marco A. dos Santos, J. Arie Vonk, Greenhouse Gas Emissions from Reservoir Water Surfaces: A New Global Synthesis, *BioScience*, Volume 66, Issue 11, 1 November 2016, Pages 949–964, <https://doi.org/10.1093/biosci/biw117>

Response 27: Thanks for this suggestion. As we have already incorporated the reservoir area into estimation (see response 4 for more detailed information), we now changed the statement to “Nevertheless, we believe that the changes in CO₂ and CH₄ estimated herein represent conservative values, and the magnitudes of these changes will be higher when increased lacustrine eutrophication and expanded lakes in smaller size are incorporated into the gas exchange calculations (Beaulieu, DelSontro and Downing 2019, DelSontro et al. 2018, Davidson et al. 2018, Holgerson and Raymond 2016, Downing 2010).”. These references you mentioned are all very informative, but are no longer needed after the revision.

Point 28: P12L235: “Our detailed mapping of the dynamics of 3.4 million lakes can potentially be used to better characterize regional-to-global hydrological budgets...” A brief mention could also be made of the possibilities for a more thorough assessment of the causes of surface water extent variations given that the present study is (understandably, given its scope) cursory in its assessment of the effects of climatic and anthropogenic influences.

Response 28: Thanks for the suggestion. We added the following clause to the context: “In addition, our dataset enabled a more thorough evaluation of the causes of the variations of surface water extent that may previously be constrained by the completeness or quality of global lake mapping (especially for small lakes), so as to gain a more comprehensive perception of the impacts of climate change and anthropogenic activities.”.

Point 29: P15L349: Here it is worth pointing out as an assurance to the reader that Pekel et al. demonstrated remarkable continuity in the accuracy of the GSWO among sensors and, consequently, through time, which is paramount to the validity of this analysis.

Response 29: Thanks for this valuable suggestion. The statement has been revised as below: “Extensive validation of the GSWO dataset has been conducted at a global scale, over the whole study period and among all involving Landsat sensors. The results demonstrate the high accuracy of the surface water delineation in the GSWO datasets (1<% false water area detections and <5% missed water area) and, consequently, the ability to afford comparable, continuous, and consistent mapping spatially, temporally and across sensors.”

Point 30: P15L353: How was the lower limit of 33 pixels determined? Was a sensitivity analysis conducted in terms of model performance with higher and lower size limits? How

does performance change towards that lower limit of 0.03 km² (versus for all small lakes together)?

Response 30: Thanks for this comment. In practice, we first predefined the cutting threshold of samples as 0.03 km² according to our visual observations. The main objectives were to exclude small polygon residuals generated during this sample extraction procedure as well as to screen out small isolated agricultural fields. Then the lake samples with sizes above this threshold were trained by the models to extract their underlying features and yield predictions. As we stated above (Response 3), the setting of pre-define cutting threshold for lake samples should probably be responsible for the relatively high omission error for small lakes (especially those around 0.03km²). We also tested the performance of models toward the lower limit of 0.03km² in Response 3: “As seen in the table, the omission and commission errors generally decreased as the lake size increased. The errors were extremely high for lakes with sizes <0.03 km² in both models, with omission error of >50% and commission error of >15%. The accuracy was much higher when considering lakes with sizes >0.1 km² (the bottom size threshold for most global lake datasets), where the corresponding omission and commission values dropped to below 20% and 5%, respectively. Nevertheless, although lakes with the size range of 0.03-0.05 km² still faced high omission issues, the commission error declined suddenly from >15% to ~5%. In practice, we treasured commission error more than omission error to ensure that the detected portion of our GLAKES polygons was generally “true” and thus could be placed with more confidence in further analysis. Therefore, we kept the size threshold as 0.03km² to include more lakes in our dataset without much compromise on misclassification.” Of course, as we stated above (Response 3), a better solution may be the inclusion of samples with sizes <0.03 km² in model training, followed by a result-oriented determination of cutting threshold (i.e., finding out a threshold where the accuracy for lakes below the certain size threshold was unacceptable, if possible), which may be considered in our next version of GLAKES in the future.

Point 31: P17L407: It seems that the reference for this dataset is incorrect. Ref 49 points to Yamazaki, D. et al. A high-accuracy map of global terrain elevations. *Geophysical Research Letters* 44, 5844-5853 (2017). However, this reference does not include any reference to OSMWL. I believe that the authors may be referring to Yamazaki, D., Ikeshima, D., Sosa, J., Bates, P. D., Allen, G. H., & Pavelsky, T. M. (2019). MERIT Hydro: a high-resolution global hydrography map based on latest topography dataset. *Water Resources Research*, 55, 5053–5073. <https://doi.org/10.1029/2019WR024873> And the following dataset: http://hydro.iis.utokyo.ac.jp/~yamadai/OSM_water/index.html.

Response 31: Thanks for this detailed comment. Yes, the OSMWL referred to the indicated dataset http://hydro.iis.utokyo.ac.jp/~yamada/OSM_water/index.html. The original reference has been replaced by the recommended reference in our revised manuscript.

Point 32: P17L409: How were the thresholds of 5% and 30% for non-floodplain and floodplain regions determined?

Response 32: For non-floodplain regions, a threshold of 5% was set to mask land pixels and exclude low-confidence waters that may be caused by classification errors embedded in GSWO. As for lakes in floodplains, the occurrence threshold was set to 30% mainly because that 30% was large than the occurrence of long-term floodwaters that lasted for less than one season (i.e., 25%) each year. Therefore, the application of 30% as an occurrence threshold enabled the extraction of the ‘core’ portions of lakes and segmentation of these portions from the larger-scale floodwaters. Of course, it should be noted that there were no implicit values for the two thresholds. In contrast, 5% and 30% were only our definitions of the thresholds for lakes without and within floodplains, on the basis of visual inspections across the globe.

To make the determination of thresholds more clear, relevant statements were revised below: “We used the GSWO map to mask land pixels and exclude low-confidence water pixels (some of which may be caused by the inherent classification errors of GSWO) with <5% occurrence (i.e., <5% of the Landsat observations were classified as water during the past four decades). Notably, in floodplains, the occurrence threshold was set to 30% instead of 5% to capture the ‘core’ portions of lakes and segment these portions from the larger-scale floodwaters (periodically occurring over a long time but lasting for less than one season (i.e., 25% occurrence) each year).”

Point 33: P18L426: I believe that the correct word would be “compared to” rather than “with” in “floodplains showed distinctive patterns with all other lake regions...”.

Response 33: Thanks for this detailed suggestion. The expression “with” was corrected as “compared to”.

Point 34: P18L428: It makes sense that additional region types were identified, yet their relationship to the normal and floodplain region types (and the multiple uses of the term region or region type) is unclear and there is no reference elsewhere of these “region subtypes” in the manuscript (in the figures or in the performance reporting). How are model uncertainties distributed among those regions?

Response 34: Sorry for the previous vagueness of the relationship between the Normal/Floodplain Model and the five region types. First of all, we added figures exhibiting examples of the 5 categories of sample regions (see below). Note that the region type was only the representativeness of the major hydrological features of the lakes within the region bound, where lakes with distinctive features may also co-exist. In practice, not all five categories of sample regions were included in the Normal Model and Floodplain Model. Specifically, the Normal model consisted of type 1 (“lakes with middle/high occurrence”, HO), 2 (“lakes with low occurrence”, LO), 3 (“large lakes”, LL) and 4 (“lakes alongside river”, AR), since the type 5 (“lakes within floodplains”, WF) was not the target of the Normal Model. On the other hand, the Floodplain Model was made up of types 1, 3, 4 and 5. The reason why types 1, 3 and 4 were included for model interpretation was that the main patterns described by types 1, 3 and 4 were also observable within the regions defined by type 5, while type 2 was excluded because of the relatively high occurrence threshold (i.e., 30%) applied for the Floodplain Model.

In addition, we re-calculated the error matrix listed in Table S4 for each region type individually and specified the region types of each scatter point in Extended Data Fig. 2 to assess the model’s performance across different categories of region (see below).

Overall, the accuracy of the former three categories (HO, LO, and LL) was higher than that of the remaining two categories (AR and WF), probably owing to the relatively intricate hydrological conditions of the last two types of regions. Here the largest omission errors (19.6%) of the Normal Model originated from AR, which probably resulted from the missed detection of oxbow lakes that were hardly distinguishable from rivers. There was additional and comparative omission lying in the Floodplain Model, i.e., WF (12.8%), where the occurrence patterns were extremely complicated, and the exact floodplain extents were hard to depict. It is noteworthy that the deviation of the scatter points representing the region type AR was not evident for either model in Extended Data Fig. 2, although exhibiting large omission errors. This is because their containing lake area was generally small and thus, their scatter points were hidden in the lower-left region of the scatter plot with dense scatter concentration. Similarly, commission errors were relatively low in almost all region types, especially compared to omission errors, confirming the model’s conservative strategy in delineation for all region types analyzed.

Another important step was that the lakes from Normal Model and Floodplain Model were combined to generate our final version of global lake polygons (the “GLAKES” dataset). In practice, given that both the Normal Model and Floodplain Model yielded lake predictions at global coverage, we applied pre-defined river buffer zones to determine whether the extracted lake polygons from the Normal Model or Floodplain Model should be used for our final GLAKES dataset. Specifically, for buffer zones

flagged as “floodplain”, lake polygons (within these buffer zones) extracted from the Floodplain Model were selected as a part of GLAKES dataset, while the corresponding outputs from the Normal Model were discarded. On the contrary, for all remaining areas (including buffer zones flagged as “normal” and areas outside the river buffer zones), lake polygons from the Normal Model were included in our final dataset. The basic principle for the determination of the exact flag (“normal” or “floodplain”) for each buffer zone is to measure the extent of seasonally flooded non-lake and non-river waters within the buffer zone. To begin with, a 1km buffer was applied to each vectorized polygon of global rivers documented in the GRWL Mask V01.01 product (<https://zenodo.org/record/1297434#.YrvEzj5ByUk>). Within each buffer, the area of seasonally flooded non-lake and non-river waters was calculated by summing the area of all GSWO pixels with occurrence <75% (to exclude the permanent and near-permanent water), except for those already being defined as rivers (by GRWL mask) and lakes (by rasterized HydroLAKES polygons). Finally, buffer zones where the ratio of their containing flooded area to the corresponding buffer area exceeded the flooding threshold were flagged as “floodplain”. In this study, the flooding threshold was set as 0.1 through trial and error.

Extended Data Fig. 3 | Image pairs revealing the performance of the deep-learning algorithm in predicting lake extents using the test set. The right panels show the predicted lake extents; the correct classification (Label + Prediction), omission errors (Label only), and commission errors (Prediction only) are color-coded. The left panels are the input images sourced from the GSWO dataset and are independent of the labels used for algorithm training and validation. The lower right annotations represent the abbreviations for the five region types: “lakes with middle/high occurrence” (HO), “lakes with low occurrence” (LO), “large lakes” (LL), “lakes alongside rivers” (AR) and “lakes within floodplains” (WF). For specific accuracy statistics, please refer to Extended Data Fig. 2 and Extended Data Table 1.

Extended Data Fig. 2 | Development and validation of the deep-learning algorithm for predicting lake extents. (b) The total lake count/area within each patch (512×512 pixels) from labels and predictions are compared, where region types are also annotated.

Table S4 | Accuracy assessments of our developed deep-learning algorithm for different region types.

Error matrix for the GLAKES dataset estimated using independent test labels for the Normal Model and Floodplain Model, listing the accuracy levels derived for different region types. Note that only pixel-based results are presented here as the polygon-based results are largely biased by the prevalence of small lake polygons in almost all region types, and thus cannot reflect the true deviations among different region types.

	Type Index	Type Name	Omission (%)	Commission (%)
Normal	1	High Occurrence (HO)	12.4	0.3
	2	Low Occurrence (LO)	8.5	1.6
	3	Large Lakes (LL)	3.3	0.2
	4	Alongside Rivers (AR)	19.6	5.0
	-	All	5.4	0.5
Floodplain	1	High Occurrence (HO)	8.0	1.9
	3	Large Lakes (LL)	4.2	0.3
	4	Alongside Rivers (AR)	12.6	1.9
	5	Within Floodplains (WF)	12.8	2.7
	-	All	9.6	1.9

Point 35: P18L438: were the regions allocated by formal stratified random sampling? What is the size distribution of labels (Extended Data Fig. 1 suggests that they are of different sizes)?

Response 35: Yes, the regions were allocated by the formal stratified random sampling. We have specified it in our revised manuscript. The histogram exhibiting the size distribution of sample regions (here should be sample regions instead of labels according to our new definition, see point 36) is appended in the Extended Data Fig. 2a (see below). Overall, the logarithmic sizes of these regions were approximated to a normal distribution, both in terms of the Normal model and Floodplain Model, while the total number of the former was ~1.7 times of the latter. In addition, the size of most sample regions was at the order of magnitude of $10^2 \sim 10^3 \text{ km}^2$, with a median area of $5.91 \times 10^2 \text{ km}^2$ for the Normal Model and $5.69 \times 10^2 \text{ km}^2$ for the Floodplain Model.

Extended Data Fig. 2 | Development and validation of the deep-learning algorithm for predicting lake extents. (a) Spatial distribution of the sample regions selected for training, validation and testing, along with size range of the sample regions.

Point 36: P19L446: “For each sample region, a variety of patches were randomly generated and used for model training, and the same local normalization method was also utilized for each patch.” I am unclear about the meaning of this sentence. I suggest that patches, regions (including region types and sub-types), and labels be more clearly defined and differentiated. Normalization was also not mentioned beforehand. A workflow diagram could help the readers to grasp this analysis, as it is crucial to the quality of the model.

Response 36: Thanks for this valuable suggestion. We have re-considered the relationship between these terms, and formal definitions are given to them accordingly. In our study, labels indicate the sample polygons produced from an automatic extraction of pre-processed GSWO layer as well as manual revision. Sometimes the corresponding

rasterized images covered from sample polygons are also called labels (e.g., Extended Data Fig. 3). In terms of regions (or sample regions), they refer to the rectangular boundaries (as presented in Extended Data Fig. 2a) covering labels. In addition, all regions are classified into 5 region types based on the dominating lake patterns exhibited within regions. Note that we do not use the term “normal regions” and “floodplain regions” anymore to differentiate them from the 5 categories of sample regions. In practice, we applied a pre-defined river buffer zone to determine whether the extracted lake polygons from the Normal Model or Floodplain Model were used for our final GLAKES dataset (see point 34). Patches are essentially small square areas with a fixed size of 512×512 pixels. A patch is the basic unit for the model to grasp the features of lakes. In general, the size of sample regions is too large for the U-Net model to analyze the lake features within the region boundaries. Instead, a variety of patches were randomly generated within each sample region, and the lake patterns within the patches were extracted and interpreted by the model. We have added the above explanation for some terms at their first occurrence to reduce ambiguity.

In addition, the main content of “Normalization” has been introduced in our revised manuscript (lines 555 to 557 and Supplementary Note 1). Now the sentence “For each sample region, a variety of patches were randomly generated and used for model training, and the same local normalization method was also utilized for each patch” has been rewritten as: “In general, the size of sample regions were too large for the U-Net model to analyze the lake features within the region boundaries. Instead, a variety of patches (with a fixed size of 512×512 pixels) were randomly generated within each sample region, and the lake patterns within the patches were extracted and interpreted by the model. In addition, we applied the same local normalization method from Brandt et al. (2020) for each patch. That is, the occurrence raster within all patches was normalized first with the mean and standard deviation of the corresponding sample region, while local normalization was performed on 40% of the patches, where image values within these patches were changed to form a standard normal distribution. It’s essential as the omission and commission errors of lake classification would be impacted by the proportion of the local normalization patches. Nevertheless, we maintained the probability as 40% as it also suited well for our study”.

The workflow is presented below.

Extended Data Fig. 1 | Flowchart for developing the GLAKES dataset. The complete workflow for the extraction of GLAKES lake polygons can be further divided into the following four modules. 1) Sample preparation: lake samples are generated and allocated to the training, validation and test sets for both Normal Model and Floodplain Model, where different region types representing varied lake features are considered. 2) Model application: the two U-Net models are trained to learn different features of lakes from the GSWO map, and each yield a raw global lake classification map. 3) Floodplain identification: the floodplain buffer zones are determined to combine the outputs from the Floodplain Model and Normal Model. 4) Post-processing: the two global lake classification maps further undergo several post-processing steps and is combined to generate the final GLAKES lake polygons.

Point 37: P19L461: please define MIoU in the manuscript itself.

Response 37: Thanks for this suggestion. We have revised and added the following description of MIoU in our revised manuscript: “In addition, we also introduced the mean intersection over union (MIoU) to assist model evaluation (which was not used in Brandt et al. (2020)). MIoU is a widely used image segmentation performance indicator that fully considers true positives and false negatives (Zhao et al. 2021). Specifically, the IoU of each class was calculated as the area of overlap divided by the area of union between the labels and predictions of that class. Then, the IoU from different classes was averaged to estimate MIoU”.

Point 38: P21L498: It seems from Extended Data Fig. 1 that the model systematically underpredicts label area. I suggest reporting mean percent bias for the normal and floodplain models and addressing this pattern in the text.

Response 38: Thanks for this suggestion. We have replaced the RMSE with Mean Bias Error (MBE) and Percent Bias (PBIAS) in Extended Data Fig. 2. As this comment indicated, our models systematically underpredicted label area, which was consistent with the results in Extended Data Table 1 that our models exhibited relatively high omission errors. Overall, the PBIAS was relatively low on the two models ($|PBIAS| < 10\%$), while the PBIAS in terms of label count was smaller compared to that of label area.

Extended Data Fig. 2 | Development and validation of the deep-learning algorithm for predicting lake extents. (b) The total lake count/area within each patch (512×512 pixels) from labels and predictions are compared, where region types are also annotated.

Point 39: P22L518: The GLWD is a minor component of HydroLAKES. The vast majority of lake polygons from 56°S to 60°N are from the Shuttle Radar Topography Mission (SRTM) Water Body Data (SWBD), and all lakes in Canada (62% of all lakes in the database) are from CanVec. This clause may be corrected or removed.

Response 39: Thanks for this valuable suggestion. The corrected clause is presented as follows: “Based on the Shuttle Radar Topography Mission Water Body Data (for most lakes between 56°S and 60°N), CanVec (for the majority of North American lakes), as well as other lake datasets (see HydroLAKES documentation), the most widely used

global lake database HydroLAKES was developed, along with intensive automated and manual corrections.”.

Point 40: P22L527: “the statistics for lakes smaller than 0.1 km² (if any were available) were represented by values extrapolated from larger lakes” I mentioned it in a previous comment, but I was under the impression that GLOWABO provided statistics for lakes at least down to 0.01 km² if not 0.002 km².

Response 40: Thanks for this comment. There’s some problem with our statement here as the GLOWABO truly provided lake polygons with size < 0.1km². Therefore, this clause has been removed.

Point 41: P22L538: “Moreover, we found a substantial number of missing lakes in eastern Canada and Scandinavia in the HydroLAKES dataset and lake overestimations with varying degrees in other regions (Extended Data Figs. 3b & 4); these errors could be due to uncharacterized seasonal or interannual dynamics and other unsourced uncertainties from the inherited datasets.”

- o In its current form, the first clause of this sentence is not entirely clear. Figure 3b suggests that HydroLAKES overestimates lake density and area compared to GLAKES in eastern Canada and Scandinavia (although the current sentence structure suggests the opposite) but underestimates lake prevalence in several other regions compared to GLAKES (foremost in Siberia and along major river floodplains e.g., Mississippi, Amazon, and Ganges-Brahmaputra).
- o Regarding the second clause of the sentence, I recommend caution in characterizing the discrepancies between HydroLAKES and GLAKES as necessarily errors.
 - In Canada, many lakes that are present in HydroLAKES but absent from GLAKES do exist. Canadian data in HydroLAKES was sourced from CanVec, which itself was built by digitizing topographic maps that, in my experience, are rather reliable, particularly in southern regions. I expect that GLAKES is unable to detect a lot of smaller and shallower lakes because their water surface is frozen/snow-covered for 4-9 months/year and heavily vegetated for the other months of the year (see example maps below, which fall within the frame of Figure 4 and show HydroLAKES outlines and GSWO). Many lakes in this region are transitioning to wetlands, so some may also be dry for part of the year.
 - In Siberia above 60N, the underestimation of lake prevalence by HydroLAKES is real. This underestimation stems from the fact that polygons in this region were generated from the MODerate resolution Imaging Spectroradiometer (MODIS)

MOD44W Collection 5 water mask, which has a resolution of only 250-m. A considerable proportion of surface water bodies between 10 and 25 ha (≤ 4 pixels) were not detected in this region due to the coarse pixel size of the MODIS instrument (each pixel is ~ 6.25 ha in area).

▪ The overall higher number of small lakes in GLAKES compared to HydroLAKES South of 60N may also stem from two other reasons:

- a size discontinuity in the SWBD that was used to produce HydroLAKES in this region. The minimum size threshold used by technicians for digitizing a waterbody was a length of 600 m (and a width of 90 m). The largest lake missing due to this constrain is theoretically a round lake of 570 m in diameter spanning ~ 25 ha, and the proportion of omitted lakes increases with decreasing lake area.

- The inclusion of additional seasonal waterbodies, including some that cannot really qualify as lakes, particularly in river floodplains (e.g., lake_id 69 is $> 2,000$ km² but appears to result from a flood), and of flooded fields for agriculture (lake_id 410210 are human-flooded fields).

o A broader point with these examples is that, while GLAKES may not be always right (and lakes found in HydroLAKES may thus not be errors), is that a strength of GLAKES is its global coherence/consistency (as I pointed out in previous sections), compared to datasets that result from aggregating multiple datasets. This is worth highlighting.

Response 41: Thanks for this valuable comment with detailed explanations, which inspired us a lot about the discrepancies between GLAKES and HydroLAKES dataset. Based on this suggestion and the above-mentioned comments, now the relevant sentences have been rewritten as: “Moreover, we found a substantial number of missing lakes in eastern Canada and Scandinavia in our dataset compared to HydroLAKES, as well as lake overestimations with varying degrees in other regions, such as Siberia and major river floodplains (Extended Data Figs. 4b & 5). These discrepancies could be raised for many reasons, either to be responsible for the inherent limitation of the GLAKES or otherwise HydroLAKES. One example is the inability of GLAKES (or GSWO) to capture lakes that are seasonally ice-covered throughout the year and heavily vegetated in the remaining month, which is typical for some small and shallow lakes in places such as the Canada Shield. The large values of GLAKES could also be partially explained by the inclusion of some agricultural fields (used to be lakes) or accidentally large floodplains. In contrast, for HydroLAKES, the constraint of its composing dataset (e.g., MODIS MOD44W water mask and SRTM Water Body Data) in detecting small lakes may be the possible reason for the lake underestimation in some regions. Overall, both GLAKES and HydroLAKES have their own strengths and limitations in terms of lake coverage, but what distinguishes GLAKES is its global consistency (not mosaic

from different datasets), higher resolution (better characterizes water/land interface), the reflection of multidecadal lake extent (not snapshot on short time period) as well as the inclusion of smaller lakes (<0.1 km²).”

Point 42: P23L553: The area of each pixel is 900 m².

Response 42: Corrected.

Point 43: P23L563: In Pekel et al. 2016, changes in water occurrence between epochs is matched by month (see quote from the original publication below) to avoid artefacts stemming from unequal detection among months and satellite coverage. Is the current computation approach immune from this issue?

"Change in water occurrence intensity between two epochs (16 March 1984 to 31 December 1999, and 1 January 2000 to 10 October 2015) was also produced (Extended Data Fig. 6a). This is derived from homologous pairs of months (that is, the same months contain valid observations in both epochs). The occurrence difference between epochs was computed for each pair and differences between all homologous pairs of months were then averaged to create the surface water occurrence change intensity map. Areas where there are no pairs of homologue months could not be mapped. The averaging of the monthly processing mitigates variations in data distribution over time (that is, both seasonal variation in the distribution of valid observations, temporal depth and frequency of observations through the archive) and provides a consistent estimation of the water occurrence change. This map shows where surface water occurrence increased, decreased or remained invariant between the two epochs."

Response 43: This is a good question. Briefly speaking, our current computation approach would not face the same issue mentioned above, mainly due to the discrepancies in the underlying data used for the calculation of water occurrence. In fact, the method we used for occurrence calculation is generally inherited from Pekel et al. (2016), i.e., through normalizing the number of water presence (N_w) incidences against the number of valid observations (N_{v0}) within a period. However, the meaning of N_w and N_{v0} actually differed in our calculation compared to that of Pekel due to the lack of relevant information. Taking N_{v0} as an example, the N_{v0} in Pekel's calculation represented the actual number of valid observations, which could be >1 within a month. However, such information (i.e., the number of valid observations for each month) was not provided by the author, which forced us to utilize alternative data for the calculation of occurrence. In practice, the N_{v0} was derived by using the GSW MWH data collection, where each pixel was assigned to one of the three values (0: no data, 1: non-water pixel,

and 2: water pixel). Compared to the N_{vo} from Pekel (which could be >1 within a month and thus varied among seasons), the value of N_{vo} obtained in our study would be either 0 or 1 for any month (0 if the corresponding pixel value on the GSW MWH layer this month indicated 0, and 1 otherwise), regardless of the season and year. Therefore, the MWH layer-based calculation actually erased the seasonal variations embedded in the calculation approach of Pekel, which was the reason why our method was exempted from the indicated issue.

Point 44: P24L579: Here and elsewhere in the manuscript, it would be worth caveating that lakes identified with this method is mostly focused on “lakes that recently (probably within the last few decades”) detached from glaciers due to glacial retreat as well as larger supraglacial lakes that are persistent enough to be visible on multi-year mosaics.”, as explained in Shugar et al.13.

Response 44: Thanks for this kind reminder. We have added relevant content to inform readers about this issue, which appeared like this: “We identified glacier-fed lakes as lakes that spatially intersected with the 1-km buffers surrounding the glacier polygons obtained from the RGI 6.0 and IMBIE Rignot datasets, following the same method as Shugar et al. (2020). It should be noted that the main focus of this method is lakes experiencing recent detachment from glaciers within a few decades or large supraglacial lakes that are highly distinguishable on long-term satellite observations.”.

Point 45: P24L583: “the percent area of permafrost within lake polygon boundaries was 10%”?

Response 45: Not exactly. The permafrost data we used here was gridded at 0.1 degrees. “The percent area $>10\%$ ” indicated the $0.1^\circ \times 0.1^\circ$ grids whose inside permafrost coverage was $>10\%$. The permafrost-fed lakes were then identified by applying spatial intersection with those selected grids with permafrost coverage $>10\%$. To avoid confusion, the relevant sentence has been rewritten as below: “Likewise, a spatial intersection approach was applied to identify lakes that received water supply from permafrost (i.e., intersection with selected $0.1^\circ \times 0.1^\circ$ grids whose inside permafrost coverage was $>10\%$, determined by using permafrost distribution data sourced from the National Snow and Ice Data Center) (Extended Data Fig. 6f).”.

Point 46: P24L587: GEODAR authors report including 23,680 dam points (often, multiple dams can be associated with a single reservoir) and 20214 reservoir polygons. How was the intersection conducted to extract 24,514 reservoirs in GLAKES?

Response 46: Thanks for this good question. This may be due to the fact that a single GeoDAR polygon may correspond to multiple GLAKES polygons. The generation of multiple GLAKES polygons within a single GeoDAR polygon may stem from both the discontinuity of the underlying GSWO data and the classification algorithm of our U-Net model. The figure below demonstrates the two situations. As can be seen in the left portion of the figure, the GLAKES polygon was broken down into two pieces (indicated by the orange circle), where the underlying GSWO layer that connected the two GLAKES polygons was too slim (only 4 pixels in one column) to be captured by the U-Net Model. As for the right portion of the figure, an evident gap could be observed in the place marked by the orange circle, which naturally resulted in the segmentation of the original reservoir into two parts.

Fig. Comparison of GLAKES (Black) and GeoDAR (Red) polygons. The underlying layer indicated water occurrence from the GSWO dataset.

Point 47: P25L595: Please provide a table summarizing the equations used.

Response 47: Thanks for this suggestion. We have changed the approach for calculating CH₄ emissions. Now the method for calculating CH₄ emissions is the same as that for CO₂ emissions. Please see response 49 for detailed information.

Point 48: P25L603: What was the data source used to determine air temperature?

Response 48: The air temperature data for each lake was retrieved on the basis of the ERA5-Land monthly averaged air temperature at 2m. Nevertheless, as we mention in response 49, we have changed the approach for calculating CH₄ emissions. Now the method for calculating CH₄ emissions is the same as that for CO₂ emissions. Therefore, the air temperature data is no longer needed anymore. Please see response 49 for detailed information.

Point 49: P25L610: Does this imply that lakes were not classified among the logarithmic size classes for computing CH₄ emissions? Please clarify.

Response 49: Thanks for this comment. In our previous submission, the CH₄ emissions were not computed by directly multiplying the size-dependent mean flux estimates from Holgerson and Raymond (2016) with the total lake area of each logarithmic size class. We actually utilized multiple equations from Holgerson and Raymond (2016) to estimate flux and carbon emission for each lake, and sum them all to obtain the global lake emissions. This process, however, we thought might have some issues. As reviewer 1 said, the occurrence of negative carbon fluxes might reflect that the empirical equations used in Holgerson and Raymond (2016) in applying all lakes were problematic. This could both result from the fact that these lakes might be more closely related to water productivity (or other factors) instead of lake size, or the empirical equations in calculating carbon emissions had large uncertainties for lakes outside the geographic range (i.e., 30°N-70°N) of the in-situ samples used to construct the empirical equations.

Reviewer 1 recommended the use of the average flux within binned logarithmic size classes. Although this comment was mainly for CO₂ emissions, we thought it was also applicable to CH₄ emissions, though uncertainty still existed (see Supplementary Note 3). Overall, we kept the method for calculating the CH₄ emissions the same as that of calculating CO₂ emissions and updated relevant results. Considering only the emissions from lakes > 0.1km² (which were the common size range for lakes in GALKES and GLOWABO dataset), we could clearly find out that our former version of the global CH₄ estimate was evidently overestimated compared to that of Holgerson and Raymond (2016), given that the lake area used in our study was smaller than that of Holgerson and Raymond (2016), but we eventually yielded a far larger emission value (5.3 vs 2.0 Tg C yr⁻¹), even though the equations used to calculate the CH₄ emissions were from their study. In contrast, the magnitude of global CO₂ emission and updated global CH₄ emission values were reasonable compared to that of Holgerson and Raymond (2016),

the discrepancy between which mainly resulted from the deviations in the lake area used for calculations.

Point 50: P26L628: By combining, do you mean that you computed the product of the probability-weighted lake areas and lake-specific CO₂/CH₄ areal fluxes, then summed across lakes?

Response 50: Thanks for this comment. As we stated in response 49, the global CO₂ emissions and updated CH₄ emissions were computed by directly multiplying the size-dependent mean flux values from Holgerson and Raymond (2016) with the total lake area of each logarithmic size class, followed by summing across size classes.

Point 51: Extended Data Fig. 1. Label area unit should be km², not km³.

Response 51: Thanks for this detailed comment. The label area unit has been changed accordingly.

Point 52: Extended Data Fig. 4. In light of my previous comment on possible issues with GLAKES in Canada, I recommend altering the language of the title of this figure to a more neutral tone about comparing.

Response 52: Thanks for this suggestion. We revised the full description as below: “Extended Data Fig. 5 | Examples showing the discrepancies of lake extents between our GLAKES dataset and the HydroLAKES dataset. (a) Inconsistent delineation of lake extents in eastern Canada, where many regions within the lake boundaries indicated by HydroLAKES show low (or zero) water occurrence, resulting in no detection in GLAKES dataset. (b) Divergent lakes mapping in northern Russia, where many water bodies with high water occurrence (>90%) are not included in HydroLAKES. In particular, a substantial number of lakes with surface area >0.03 km² and <0.1 km² (i.e., the lower limit for HydroLAKES) are mapped in GLAKES datasets. The background images are obtained from the GSWO dataset.”.

Point 53: Extended Data Fig. 8. Please indicate the meaning of the box plot components in the legend.

Response 53: Thanks for this comment. The class “Small” indicates that only small lakes within each 1°×1° grid cell were used to calculate the relative area changes, while “All” means all lakes within each 1°×1° grid cell were applied for aggregation.

Point 54: Extended Data Figure 9. A few suggestions below the comparison to be more informative:

- o For the different estimates to be more comparable, I recommend either writing the minimum size of lakes included in each study or maybe showing these as 2D plots showing the total lake area/number determined in each study on one axis and the estimated emission on the other axis. Alternatively, different bar plots could be coloured depending on which baseline lake dataset was used.
- o I am not an expert in biogeochemistry, but do all these studies include the same types of fluxes (diffusive and/or ebullitive? See Deemer et al.11)?
- o Finally, the error bars represent different intervals (e.g., Monte-Carlo approach for Holgerson et al. to produce a 95% interval, vs. a min-max fluxes for this study) so this should be explained in the caption.xi

Response 54: Thanks for this valuable comment. First of all, we colored the bar plots according to the baseline lake dataset used for emission calculations and specified the minimum lake unit as suggested. This's a good suggestion, since the discrepancies in carbon emissions from different studies are not only originated from the specific approach for evaluation, but also highly dependent on the underlying lake area dataset used for upscaling. It's helpful for variables control for better comparison.

As for the flux type, we only calculated the diffusive CH₄ emissions, as the data availability of direct in-situ ebullition rate measurements, as well as their highly spatiotemporal variability, hampered the systematic assessments of CH₄ ebullition at a global scale (see Supplementary Note 3).

Finally, the explicit meaning of the error bar for each study has also been added in the caption (see figure below).

Extended Data Fig. 9 | Comparisons of global lacustrine carbon emissions from Holgerson et al., 2016 with other previous estimates (Bastviken et al. 2004, Cole et al. 2007, Duarte et al. 2008, Marotta et al. 2009, Tranvik et al. 2009, Bastviken et al. 2011, Raymond et al. 2013, Holgerson and Raymond 2016, DelSontro et al. 2018, Li et al. 2020), and the estimated results by applying Holgerson’s method in our GLAKES lake dataset (this study). (a) CO₂. (b) CH₄. Note: 1) Estimations using different lake area datasets (Meybeck 1995, Jacob 2002, Lehner and Döll 2004, Sobek, Tranvik and Cole 2005, Downing et al. 2006, Verpoorter et al. 2014) are marked by different colors. 2) The hatched and empty areas represent the statistics derived for lakes with surface area below and above 0.1 km², where 0.1 km² can be used to define the common size range without extrapolation for all lake datasets listed above. The filled areas denote results that are incapable of subdivision since the original studies did not provide relevant binned size information. 3) The minimum lake size threshold for Holgerson *et al.*, 2016 is 0.0001 km², for Li *et al.*, 2020 is 0.002 km², for Bastviken *et al.*, 2004 & Cole *et al.*, 2007 & Duarte *et al.*, 2008 is unknown, for this study is 0.03 km² and for the others is 0.001 km². 4) The error bars, if any, represent the lower and upper bounds of the estimations, although with different implications. Cole *et al.*, 2007 & Marrotta *et al.*, 2009: min-max; Raymond *et al.*, 2013 & DelSontro *et al.*, 2018: 95% confidence interval; Holgerson *et al.*, 2016 & this study: 25–75th percentiles; The others: unknown. 5) The emissions from both natural lakes and reservoirs are included in most studies except for Duarte *et al.*, 2008, Marotta *et al.*, 2009, Tranvik *et al.*, 2009 and Bastviken *et al.*, 2004, where reservoirs are excluded.

Reference

- Natural Resources Canada. CanVec Hydrography: Waterbody Features. Version 12.0. Data. Available at <ftp://ftp2.cits.rncan.gc.ca/pub/canvec/>. Accessed May 2013.
- Bastviken, D., J. Cole, M. Pace & L. Tranvik (2004) Methane emissions from lakes: Dependence of lake characteristics, two regional assessments, and a global estimate. *Global biogeochemical cycles*, 18.
- Bastviken, D., L. J. Tranvik, J. A. Downing, P. M. Crill & A. Enrich-Prast (2011) Freshwater methane emissions offset the continental carbon sink. *Science*, 331, 50-50.
- Beaulieu, J. J., T. DelSontro & J. A. Downing (2019) Eutrophication will increase methane emissions from lakes and impoundments during the 21st century. *Nature communications*, 10, 1-5.
- Beaulieu, J. J., M. G. McManus & C. T. Nietch (2016) Estimates of reservoir methane emissions based on a spatially balanced probabilistic - survey. *Limnology and Oceanography*, 61, S27-S40.
- Borges, A. V., L. Deirmendjian, S. Bouillon, W. Okello, T. Lambert, F. A. Roland, V. F. Razanamahandry, N. R. G. Voarintsoa, F. Darchambeau & I. A. Kimirei (2022) Greenhouse gas emissions from African lakes are no longer a blind spot. *Science Advances*, 8, eabi8716.

- Brandt, M., C. J. Tucker, A. Kariryaa, K. Rasmussen, C. Abel, J. Small, J. Chave, L. V. Rasmussen, P. Hiernaux & A. A. Diouf (2020) An unexpectedly large count of trees in the West African Sahara and Sahel. *Nature*, 587, 78-82.
- Chamberlain, S. D., N. Gomez - Casanovas, M. T. Walter, E. H. Boughton, C. J. Bernacchi, E. H. DeLucia, P. M. Groffman, E. W. Keel & J. P. Sparks (2016) Influence of transient flooding on methane fluxes from subtropical pastures. *Journal of Geophysical Research: Biogeosciences*, 121, 965-977.
- Cole, J. J., Y. T. Prairie, N. F. Caraco, W. H. McDowell, L. J. Tranvik, R. G. Striegl, C. M. Duarte, P. Kortelainen, J. A. Downing & J. J. Middelburg (2007) Plumbing the global carbon cycle: integrating inland waters into the terrestrial carbon budget. *Ecosystems*, 10, 172-185.
- Davidson, T. A., J. Audet, E. Jeppesen, F. Landkildehus, T. L. Lauridsen, M. Søndergaard & J. Syväranta (2018) Synergy between nutrients and warming enhances methane ebullition from experimental lakes. *Nature Climate Change*, 8, 156-160.
- Deemer, B. & M. A. Holgerson (2021) Drivers of methane flux differ between lakes and reservoirs, complicating global upscaling efforts. *Journal of Geophysical Research: Biogeosciences*, 126, e2019JG005600.
- Deemer, B. R., J. A. Harrison, S. Li, J. J. Beaulieu, T. DelSontro, N. Barros, J. F. Bezerra-Neto, S. M. Powers, M. A. Dos Santos & J. A. Vonk (2016) Greenhouse gas emissions from reservoir water surfaces: a new global synthesis. *BioScience*, 66, 949-964.
- DelSontro, T., J. J. Beaulieu & J. A. Downing (2018) Greenhouse gas emissions from lakes and impoundments: Upscaling in the face of global change. *Limnology and Oceanography Letters*, 3, 64-75.
- Downing, J. A. (2010) Emerging global role of small lakes and ponds: little things mean a lot. *Limnetica*, 29, 0009-24.
- Downing, J. A., Y. Prairie, J. Cole, C. Duarte, L. Tranvik, R. G. Striegl, W. McDowell, P. Kortelainen, N. Caraco & J. Melack (2006) The global abundance and size distribution of lakes, ponds, and impoundments. *Limnology and Oceanography*, 51, 2388-2397.
- Duarte, C. M., Y. T. Prairie, C. Montes, J. J. Cole, R. Striegl, J. Melack & J. A. Downing (2008) CO₂ emissions from saline lakes: A global estimate of a surprisingly large flux. *Journal of Geophysical Research: Biogeosciences*, 113.
- Greve, P., B. Orlowsky, B. Mueller, J. Sheffield, M. Reichstein & S. I. Seneviratne (2014) Global assessment of trends in wetting and drying over land. *Nature geoscience*, 7, 716-721.
- Hayes, N. M., B. R. Deemer, J. R. Corman, N. R. Razavi & K. E. Strock (2017) Key differences between lakes and reservoirs modify climate signals: A case for a new conceptual model. *Limnology and Oceanography Letters*, 2, 47-62.
- Hinton, G., L. Deng, D. Yu, G. E. Dahl, A.-r. Mohamed, N. Jaitly, A. Senior, V. Vanhoucke, P. Nguyen & T. N. Sainath (2012) Deep neural networks for acoustic modeling in speech recognition: The shared views of four research groups. *IEEE Signal processing magazine*, 29, 82-97.
- Holgerson, M. A. & P. A. Raymond (2016) Large contribution to inland water CO₂ and CH₄ emissions from very small ponds. *Nature Geoscience*, 9, 222-226.
- Jacob, K. 2002. *Limnology: Inland Water Ecosystems 2nd Edition*. Prentice Hall, Upper Saddle River.
- Keller, P. S., R. Marcé, B. Obrador & M. Koschorreck (2021) Global carbon budget of reservoirs is overturned by the quantification of drawdown areas. *Nature Geoscience*, 14, 402-408.

- Kortelainen, P., M. Rantakari, J. T. Huttunen, T. Mattsson, J. Alm, S. Juutinen, T. Larmola, J. Silvola & P. J. Martikainen (2006) Sediment respiration and lake trophic state are important predictors of large CO₂ evasion from small boreal lakes. *Global Change Biology*, 12, 1554-1567.
- Krizhevsky, A., I. Sutskever & G. E. Hinton (2012) Imagenet classification with deep convolutional neural networks. *Advances in neural information processing systems*, 25.
- LeCun, Y., Y. Bengio & G. Hinton (2015) Deep learning. *nature*, 521, 436-444.
- Lehner, B. & P. Döll (2004) Development and validation of a global database of lakes, reservoirs and wetlands. *Journal of hydrology*, 296, 1-22.
- Li, M., C. Peng, Q. Zhu, X. Zhou, G. Yang, X. Song & K. Zhang (2020) The significant contribution of lake depth in regulating global lake diffusive methane emissions. *Water research*, 172, 115465.
- Liu, Y. H. 2018. Feature extraction and image recognition with convolutional neural networks. In *Journal of Physics: Conference Series*, 062032. IOP Publishing.
- Long, J., E. Shelhamer & T. Darrell. 2015. Fully convolutional networks for semantic segmentation. In *Proceedings of the IEEE conference on computer vision and pattern recognition*, 3431-3440.
- Ma, L., M. Li, X. Ma, L. Cheng, P. Du & Y. Liu (2017) A review of supervised object-based land-cover image classification. *ISPRS Journal of Photogrammetry and Remote Sensing*, 130, 277-293.
- Magnuson, J. J., D. M. Robertson, B. J. Benson, R. H. Wynne, D. M. Livingstone, T. Arai, R. A. Assel, R. G. Barry, V. Card & E. Kuusisto (2000) Historical trends in lake and river ice cover in the Northern Hemisphere. *Science*, 289, 1743-1746.
- Marcé, R., B. Obrador, L. Gómez-Gener, N. Catalán, M. Koschorreck, M. I. Arce, G. Singer & D. von Schiller (2019) Emissions from dry inland waters are a blind spot in the global carbon cycle. *Earth-science reviews*, 188, 240-248.
- Marotta, H., C. M. Duarte, S. Sobek & A. Enrich - Prast (2009) Large CO₂ disequilibria in tropical lakes. *Global biogeochemical cycles*, 23.
- Meybeck, M. 1995. Global distribution of lakes. In *Physics and chemistry of lakes*, 1-35. Springer.
- Michmerhuizen, C. M., R. G. Striegl & M. E. McDonald (1996) Potential methane emission from north - temperate lakes following ice melt. *Limnology and Oceanography*, 41, 985-991.
- Natchimuthu, S., B. Panneer Selvam & D. Bastviken (2014) Influence of weather variables on methane and carbon dioxide flux from a shallow pond. *Biogeochemistry*, 119, 403-413.
- Pekel, J.-F., A. Cottam, N. Gorelick & A. S. Belward (2016) High-resolution mapping of global surface water and its long-term changes. *Nature*, 540, 418-422.
- Pickens, A. H., M. C. Hansen, M. Hancher, S. V. Stehman, A. Tyukavina, P. Potapov, B. Marroquin & Z. Sherani (2020) Mapping and sampling to characterize global inland water dynamics from 1999 to 2018 with full Landsat time-series. *Remote Sensing of Environment*, 243, 111792.
- Pickens, A. H., M. C. Hansen, S. V. Stehman, A. Tyukavina, P. Potapov, V. Zalles & J. Higgins (2022) Global seasonal dynamics of inland open water and ice. *Remote Sensing of Environment*, 272, 112963.
- Prigent, C., C. Jimenez & P. Bousquet (2020) Satellite - derived global surface water extent and dynamics over the last 25 years (GIEMS - 2). *Journal of Geophysical Research: Atmospheres*, 125, e2019JD030711.
- Ran, L., D. E. Butman, T. J. Battin, X. Yang, M. Tian, C. Duvert, J. Hartmann, N. Geeraert & S. Liu (2021) Substantial decrease in CO₂ emissions from Chinese inland waters due to global change. *Nature communications*, 12, 1-9.

- Rasilo, T., Y. T. Prairie & P. A. Del Giorgio (2015) Large - scale patterns in summer diffusive CH₄ fluxes across boreal lakes, and contribution to diffusive C emissions. *Global Change Biology*, 21, 1124-1139.
- Raymond, P. A., J. Hartmann, R. Lauerwald, S. Sobek, C. McDonald, M. Hoover, D. Butman, R. Striegl, E. Mayorga & C. Humborg (2013) Global carbon dioxide emissions from inland waters. *Nature*, 503, 355-359.
- Reichstein, M., G. Camps-Valls, B. Stevens, M. Jung, J. Denzler & N. Carvalhais (2019) Deep learning and process understanding for data-driven Earth system science. *Nature*, 566, 195-204.
- Ribeiro, M., A. E. Lazzaretti & H. S. Lopes (2018) A study of deep convolutional auto-encoders for anomaly detection in videos. *Pattern Recognition Letters*, 105, 13-22.
- Roderick, M., F. Sun, W. H. Lim & G. Farquhar (2014) A general framework for understanding the response of the water cycle to global warming over land and ocean. *Hydrology and Earth System Sciences*, 18, 1575-1589.
- Ronneberger, O., P. Fischer & T. Brox. 2015. U-net: Convolutional networks for biomedical image segmentation. In *International Conference on Medical image computing and computer-assisted intervention*, 234-241. Springer.
- Rosentreter, J. A., A. V. Borges, B. R. Deemer, M. A. Holgerson, S. Liu, C. Song, J. Melack, P. A. Raymond, C. M. Duarte & G. H. Allen (2021) Half of global methane emissions come from highly variable aquatic ecosystem sources. *Nature Geoscience*, 14, 225-230.
- Sánchez-Carrillo, S., J. Alcocer, M. Vargas-Sánchez, I. Soria-Reinoso, E. M. Rivera-Herrera, D. Cortés-Guzmán, D. Cuevas-Lara, A. P. Guzmán-Arias, M. Merino-Ibarra & L. A. Oseguera (2022) Greenhouse gas emissions from Mexican inland waters: first estimation and uncertainty using an upscaling approach. *Inland Waters*, 12, 294-310.
- Sharma, S., K. Blagrove, J. J. Magnuson, C. M. O'Reilly, S. Oliver, R. D. Batt, M. R. Magee, D. Straile, G. A. Weyhenmeyer & L. Winslow (2019) Widespread loss of lake ice around the Northern Hemisphere in a warming world. *Nature Climate Change*, 9, 227-231.
- Shugar, D. H., A. Burr, U. K. Haritashya, J. S. Kargel, C. S. Watson, M. C. Kennedy, A. R. Bevington, R. A. Betts, S. Harrison & K. Strattman (2020) Rapid worldwide growth of glacial lakes since 1990. *Nature Climate Change*, 10, 939-945.
- Sobek, S., L. J. Tranvik & J. J. Cole (2005) Temperature independence of carbon dioxide supersaturation in global lakes. *Global Biogeochemical Cycles*, 19.
- Sutskever, I., O. Vinyals & Q. V. Le (2014) Sequence to sequence learning with neural networks. *Advances in neural information processing systems*, 27.
- Tangen, B. A. & S. Bansal (2019) Hydrologic lag effects on wetland greenhouse gas fluxes. *Atmosphere*, 10, 269.
- Tootchi, A., A. Jost & A. Ducharne (2019) Multi-source global wetland maps combining surface water imagery and groundwater constraints. *Earth System Science Data*, 11, 189-220.
- Tranvik, L. J., J. A. Downing, J. B. Cotner, S. A. Loiselle, R. G. Striegl, T. J. Ballatore, P. Dillon, K. Finlay, K. Fortino & L. B. Knoll (2009) Lakes and reservoirs as regulators of carbon cycling and climate. *Limnology and oceanography*, 54, 2298-2314.
- Utsumi, M., Y. Nojiri, T. Nakamura, T. Nozawa, A. Otsuki, N. Takamura, M. Watanabe & H. Seki (1998) Dynamics of dissolved methane and methane oxidation in dimictic Lake Nojiri during winter. *Limnology and Oceanography*, 43, 10-17.

- Verpoorter, C., T. Kutser, D. A. Seekell & L. J. Tranvik (2014) A global inventory of lakes based on high - resolution satellite imagery. *Geophysical Research Letters*, 41, 6396-6402.
- Wang, X., L. Feng, W. Qi, X. Cai, Y. Zheng, L. Gibson, J. Tang, X. p. Song, J. Liu & C. Zheng (2022) Continuous loss of global lake ice across two centuries revealed by satellite observations and numerical modeling. *Geophysical Research Letters*, e2022GL099022.
- Weiss, M., F. Jacob & G. Duveiller (2020) Remote sensing for agricultural applications: A meta-review. *Remote Sensing of Environment*, 236, 111402.
- Wik, M., R. K. Varner, K. W. Anthony, S. MacIntyre & D. Bastviken (2016) Climate-sensitive northern lakes and ponds are critical components of methane release. *Nature Geoscience*, 9, 99-105.
- Woolway, R. I., B. M. Kraemer, J. D. Lenters, C. J. Merchant, C. M. O'Reilly & S. Sharma (2020) Global lake responses to climate change. *Nature Reviews Earth & Environment*, 1, 388-403.
- Wulder, M. A., J. C. White, T. R. Loveland, C. E. Woodcock, A. S. Belward, W. B. Cohen, E. A. Fosnight, J. Shaw, J. G. Masek & D. P. Roy (2016) The global Landsat archive: Status, consolidation, and direction. *Remote Sensing of Environment*, 185, 271-283.
- Yamazaki, D., M. A. Trigg & D. Ikeshima (2015) Development of a global~ 90 m water body map using multi-temporal Landsat images. *Remote Sensing of Environment*, 171, 337-351.
- Yu, H., Z. Yang, L. Tan, Y. Wang, W. Sun, M. Sun & Y. Tang (2018) Methods and datasets on semantic segmentation: A review. *Neurocomputing*, 304, 82-103.
- Zeiler, M. D. & R. Fergus. 2014. Visualizing and understanding convolutional networks. In *European conference on computer vision*, 818-833. Springer.
- Zhao, S., G. Hao, Y. Zhang & S. Wang (2021) A Real-Time Semantic Segmentation Method of Sheep Carcass Images Based on ICNet. *Journal of Robotics*, 2021.
- Zheng, Y., S. Wu, S. Xiao, K. Yu, X. Fang, L. Xia, J. Wang, S. Liu, C. Freeman & J. Zou (2022) Global methane and nitrous oxide emissions from inland waters and estuaries. *Global Change Biology*.

Reviewer #1 (Remarks to the Author):

I was impressed by the thorough responses to the comments from the reviewers. I am fully satisfied by this revision and recommend publication.

Reviewer #2 (Remarks to the Author):

The revised manuscript provides substantial additional context and clarifications that enhance and explain the findings. The authors should be commended for this effort in providing this dataset along with details on so many nuanced aspects of its creation.

I was able to access the .gdb version of dataset at the google drive link and open it using open source software, however it appeared that the .shp format product listed in the readme file may have been missing. I assume the authors will also make this file available upon publication.

Reviewer #3 (Remarks to the Author):

The authors thoroughly addressed the (many) comments from the reviewers. I reiterate the importance of this work and recommend its publication, provided that my last remaining comments (minor, below) are addressed and that a last round of copy editing is conducted. Best regards,
Mathis Loïc Messager

- Several references are missing a title in the methods section: # 67, 69, 70, 71
- In general, a last copy pass would be appropriate as some sentences are slightly difficult to understand. For instance:
 - o L385: "involving Landsat sensors". "all Landsat sensors involved" would be more appropriate.
 - o "the U-Net model composes of various and hierarchical convolution layers". Composes should be replaced with "is composed of"
 - o L452 "For seasonal water bodies, we tend to address those located around rivers and meanwhile span a large scale (such as floodplains), while keeping the small ephemeral water bodies as lakes." The meaning of "We tend to address" is not clear here.
 - o L458 "and are thus beyond the capture of the GSWO images" is also difficult to understand.
 - o "lakes smaller than 0.1 km² that is publicly available without statistical extrapolation" should be written "lakes smaller than 0.1 km² without statistical extrapolation that is publicly available"
 - o "L82-85: "accessible in public" should be "publicly accessible"
 - o "such seasonal patterns of water/land/ice transition may witness substantial changes" — "undergo" substantial changes would be correct
 - o "including intensifying reductions in ice cover duration and varied changes in wetting/drying trends" is unclear, please reword
- L162-167: Despite rewording, I think that "small lake-dominated regions" remains confusing. "Regions where lake variability is dominated by small lakes" would be more adequate.
- The response to Point 1 from Reviewer 3 should be included in the manuscript's Methods, particularly that on manual corrections.
- A brief sentence mentioning response to Point 4 from Reviewer 3 would be informative.
- Point 21 to Reviewer 3: I still don't think that deep learning inherently makes it possible to detect smaller lakes. The resolution of the underlying dataset is really what enables detecting smaller lakes. Deep learning in this case seems to only enable distinguishing lakes from rivers. I think that this needs to be made clearer.
- Extended Data Fig. 1: replace "vecter" with "vector"
- Point 53 to Reviewer 3: sorry for being unclear. Please indicate the meaning of the boxplot edges and whiskers.
- I forgot to mention, but somewhere in the manuscript, a mention should be made of the the global lake area, climate, and population dataset (GLCP; <https://www.nature.com/articles/s41597-020-0517-4>). It provides lake surface area from from 1995 to 2015 for all HydroLAKES polygons based on the GSW MWH. Same limitations as HydroLAKES in terms of lake size limit, consistency,

and they computed area based on a buffer around lake polygons, which means that they may have captured water coverage from other lakes, wetlands and rivers or missed greater spatial extents.

Dear editor and reviewers:

Thanks again for your letter and for the reviewers' comments on our revised manuscript in Round I (manuscript number: NCOMMS-21-48587A). We're very happy and appreciated to receive the encouragement and recommendation for publication. We have addressed the remaining comments and made revisions accordingly, hoping that the revised version is qualified for formal publication.

Response to the reviewer 1:

General comments:

Point: I was impressed by the thorough responses to the comments from the reviewers. I am fully satisfied by this revision and recommend publication.

Response: Thanks for your encouraging comments and recommendation.

Response to the reviewer 2:

General comments:

Point: The revised manuscript provides substantial additional context and clarifications that enhance and explain the findings. The authors should be commended for this effort in providing this dataset along with details on so many nuanced aspects of its creation.

I was able to access the .gdb version of dataset at the google drive link and open it using open source software, however it appeared that the .shp format product listed in the readme file may have been missing. I assume the authors will also make this file available upon publication.

Response: Thanks for your encouraging comments and recommendation. The “.shp” version of dataset has been added and all related data are currently available at <https://doi.org/10.5281/zenodo.7016548>.

Response to the reviewer 3:

General Comments:

Point: The authors thoroughly addressed the (many) comments from the reviewers. I reiterate the importance of this work and recommend its publication, provided that my last remaining comments (minor, below) are addressed and that a last round of copy editing is conducted.

Response: Thanks for your encouraging comments and recommendation. We have read the remaining comments carefully and made revisions accordingly.

Specific comments:

Point 1: Several references are missing a title in the methods section: # 67, 69, 70, 71.

Response 1: Thanks for this detailed suggestion. We have reviewed the *Reference* section and corrected such type of issue.

Point 2: In general, a last copy pass would be appropriate as some sentences are slightly difficult to understand. For instance:

- L385: “involving Landsat sensors”. “all Landsat sensors involved” would be more appropriate.
- “the U-Net model composes of various and hierarchical convolution layers”. Composes should be replaced with “is composed of”.
- L452 “For seasonal water bodies, we tend to address those located around rivers and meanwhile span a large scale (such as floodplains), while keeping the small ephemeral water bodies as lakes.” The meaning of “We tend to address” is not clear here.
- L458 “and are thus beyond the capture of the GSWO images” is also difficult to understand.
- “lakes smaller than 0.1 km² that is publicly available without statistical extrapolation” should be written “lakes smaller than 0.1 km² without statistical extrapolation that is publicly available”.
- “L82-85: “accessible in public” should be “publicly accessible”.
- “such seasonal patterns of water/land/ice transition may witness substantial changes” — “undergo” substantial changes would be correct.
- “including intensifying reductions in ice cover duration and varied changes in wetting/drying trends” is unclear, please reword.

Response 2: Thanks for this detailed suggestion. All suggested expressions (such as “all Landsat sensors involved”) have been accepted to replace the original one (such as “involving Landsat sensors”).

The sentence “For seasonal water bodies, we tend to address those located around rivers and meanwhile span a large scale (such as floodplains), while keeping the small ephemeral water bodies as lakes.” has been changed as “For seasonal water bodies, we

implemented more careful examinations to those located around rivers and meanwhile span a large scale (such as floodplains), while keeping the small ephemeral water bodies as lakes.”

The sentence “and are thus beyond the capture of the GSWO images” has been changed as “and thus cannot be captured by the GSWO images”.

The sentence “including intensifying reductions in ice cover duration and varied changes in wetting/drying trends” has been changed as “Furthermore, such seasonal patterns of water/land/ice transition may undergo substantial changes during the whole study period owing to the impact of climate change, where the ice cover duration experienced intensifying reductions and the wetting/drying trends varied in different regions”.

Point 3: L162-167: Despite rewording, I think that “small lake-dominated regions” remains confusing. “Regions where lake variability is dominated by small lakes” would be more adequate.

Response 3: Thanks for this suggestion. The full sentence has been corrected as “The changes in small lakes showed dominant contributions (>50%) in approximately half of the examined inland regions (49.9% of the grid cells from the 1980-90s to the 2000s, and 50.1% from 2000s to 2010s) (Fig. 3c), and regions where lake variability is dominated by small lakes were spread across the entire globe in both low-populated regions and areas with high chances of human disturbance (Supplementary Fig. 9).” .

Point 4: The response to Point 1 from Reviewer 3 should be included in the manuscript’s Methods, particularly that on manual corrections.

Response 4: Thanks for this comment. We have added more necessary details of the manual corrections in the revised manuscript, as indicated in lines 373-381: “Finally, extensive visual examinations were performed from one sample region to another, where some lakes in hydrologically complex regions were given more attention, such as those in the large river basins and floodplain regions. Manual corrections were performed mainly on the following two situations: (1) river residuals resulting from the absent coverage of the corresponding river masks and (2) river-connected lakes that required further division from river channels. Of all sample polygons, Case 1 polygons frequently occurred, which could take up ~10% of the total lake samples and thus require careful inspection. On the contrary, the percentage of Case 2 polygons was minor (far less than 1%).”

Point 5: A brief sentence mentioning response to Point 4 from Reviewer 3 would be informative.

Response 5: Thanks for this suggestion. We have summarized the response to point 4 from reviewer 3 and added it to the revised manuscript accordingly (lines 142 to 150). Now the full context becomes “The median values of the relative areal changes in small lakes were +2.9% from the 1980-90s to the 2000s and +0.6% from the 2000s to the 2010s; these changes were significantly greater than those derived for medium and large lakes (matched-pair t-test, $P < 0.05$) (Fig. 3a). Such difference in temporal variability between small lakes and median/large lakes was also evident even when evaluating under equal relative pixel sizes, as revealed through a case experiment in Tibetan Plateau, which showed that lakes within the size range of 0.5-1.5 km² at the resolution of 30 m were found to exhibit a far larger range of the relative areal changes compared to lakes between 50-150 km² at the 300 m resolution (Supplementary Fig. 7).”

Point 6: Point 21 to Reviewer 3: I still don't think that deep learning inherently makes it possible to detect smaller lakes. The resolution of the underlying dataset is really what enables detecting smaller lakes. Deep learning in this case seems to only enable distinguishing lakes from rivers. I think that this needs to be made clearer.

Response 6: Thanks for this valuable suggestion. We're agree with your opinion and now the sentence has been rephrased as “The GSWO dataset provides the probability of water presence, which was established using 30-m resolution Landsat satellite observations between 1984 and 2019. Deep learning allows for the disentanglement of lakes from rivers in the GSWO images, and the integration of high-resolution remote sensing images and deep learning makes it possible to detect lakes as small as 0.03 km² ...” (lines 84-91).

Point 7: Extended Data Fig. 1: replace “vecter” with “vector”.

Response 7: Corrected as suggested.

Point 8: Point 53 to Reviewer 3: sorry for being unclear. Please indicate the meaning of the boxplot edges and whiskers.

Response 8: Thanks for this suggestion. The boxplot edges indicate the first (Q_1) and third quartile (Q_3) of the data, while the length of whiskers is 1.5 times the IQR (Interquartile range, defined as $Q_3 - Q_1$). We have clarified this in the caption of the Supplementary Fig. 9.

Point 9: I forgot to mention, but somewhere in the manuscript, a mention should be made of the global lake area, climate, and population dataset (GLCP). It provides lake surface area from 1995 to 2015 for all HydroLAKES polygons based on the GSW MWH. Same limitations as HydroLAKES in terms of lake size limit, consistency, and they computed area based on a buffer around lake polygons, which means that they may have captured water coverage from other lakes, wetlands and rivers or missed greater spatial extents.

Response 9: Thanks for this suggestion. We have added the relevant clauses in the *Accuracy assessments and comparisons with previous global lake datasets* section (lines 572 to 586). Now the full context are: “Overall, both GLAKES and HydroLAKES have their own strengths and limitations in terms of lake coverage, but what distinguishes GLAKES is its global consistency (not mosaic from different datasets), higher resolution (better characterizes water/land interface), the reflection of multidecadal lake extent (not snapshot on short time period) as well as the inclusion of smaller lakes (<0.1 km²). This is significant for the long-term monitoring of the lake surface water area dynamics. As a comparison, the Global Lake area, Climate, and Population (GLCP) dataset provides annual time series lake surface area records from 1995 to 2015 for all HydroLAKES polygons (Meyer et al. 2020). Nevertheless, GLCP faces the same limitations as HydroLAKES in terms of the lake size limit and spatial consistency. Besides, since the HydroLAKES polygons did not represent the maximum water extent, a fixed buffer zone around lakes was generated in GLCP for area estimation, which might result in fallaciously inclusion of water coverage that did not belong to the target lakes or missed detection of water area due to the insufficient coverage of the buffer outlines.”.

References

Meyer, M. F., S. G. Labou, A. N. Cramer, M. R. Brousil & B. T. Luff (2020) The global lake area, climate, and population dataset. *Scientific data*, 7, 1-12.